# Tyrosine kinase targeting uncovers oncogenic pathway plasticity in Tasmanian devil transmissible cancers

Anna Schönbichler [1], Anna Orlova[1], Carmen Kreindl[1], Lukas Endler [2], Richard Wilson [3], Lindsay Kosack[4], Anna Hofmann[2], Csilla Viczenczova[2,4], Jocelyn Darby[5], Fettah Erdogan [6], Amanda L Patchett[5], Anna Koren[4], Stefan Kubicek [4], Mathias Müller [1], Andrew S Flies [5], Andreas Bergthaler [2,4]✉ & Richard Moriggl [1,7]✉

## Abstract

**Two transmissible cancers, Devil Facial Tumour 1 (DFT1) and Devil Facial Tumour 2 (DFT2), have caused a significant decline in the Tasmanian devil population. DFT1 is driven by ERBB, while DFT2 is driven by PDGFRA. We show that DFT cancer cells exhibit distinct kinase phosphorylation profiles that dictate their responses to tyrosine kinase inhibitors. Upon long-term treatment, both DFT cell lines develop resistance, with DFT1 cells rapidly evading ERBB inhibition without major copy number alterations or significant changes in phosphorylation, suggesting signalling plasticity and engagement of alternative oncogenic drivers. In contrast, DFT2 cells exhibit a slowed development of resistance to imatinib, a selective kinase inhibitor with known activity against PDGFRs. Moreover, DFT2 cell resistance is accompanied by copy number alterations and an activation of ERBB and JAK/STAT signalling with *MHCI* downregulation, resembling DFT1 signalling. Dual targeting of ERBB and PDGFR shows synergistic effects in DFT1 and may prevent resistance emergence. These findings provide critical insight into the adaptive capacity of transmissible cancers and inform conservation strategies. Moreover, they highlight broader principles of kinase-driven resistance relevant to human cancers with high pathway plasticity.**

**Keywords** Devil Facial Tumour Disease; Phosphoproteomics; Transmissible Cancer; Tyrosine Kinase Inhibitor Resistance; Combination Therapy
**Subject Categories** Cancer; Immunology; Signal Transduction

## Introduction

Cancer is typically understood as a disease that remains confined to the individual. Yet in dogs (*Canis lupus familiaris*), Tasmanian devils (*Sarcophilus harrisii*), and certain bivalve species, cancer cells defy this dogma and have evolved into transmissible malignancies (Pearse and Swift, 2006; Pye et al, 2016; Murgia et al, 2006; Murchison et al, 2014; Metzger et al, 2015; Schönbichler and Bergthaler, 2023). These unique cancer cells not only survive horizontal transmission between hosts but also evade immune detection to establish tumours in new individuals, effectively metastasizing beyond the boundaries of a single organism.

In Tasmanian devils, two distinct transmissible cancer clones, called Devil Facial Tumour 1 (DFT1) and Devil Facial Tumour 2 (DFT2), have contributed to a devastating population decline of the species. DFT1, originating over 25 years ago from a Schwann cell of a female, has spread across more than 90% of Tasmania. It is aided by the downregulation of MHC class I (MHCI), which contributes to immune evasion of host animals (Lazenby et al, 2018; Cunningham et al, 2021; Murchison et al, 2010; Siddle et al, 2010, 2013). DFT2, discovered in 2014 and confined to southern Tasmania, also arose from a Schwann cell but in a male devil and is therefore genetically distinct (Pye et al, 2016; Patchett et al, 2020; Stammnitz et al, 2023). Unlike DFT1, DFT2 cells express MHCI and carry a Y chromosome, which may enhance immune recognition. However, observations of MHCI loss and Y chromosome deletions in DFT2 tumours in nature raise concerns about its potential to rapidly adapt new immune evasion strategies for wider transmission and its impact on devil populations (Stammnitz et al, 2023; Caldwell et al, 2018).

Efforts to immunize Tasmanian devils against DFT1 by restoring MHCI surface expression have had limited success, generating immune responses but failing to prevent transmission (Tovar et al, 2017; Pye et al, 2018, 2021). Additionally, DFT1

[1]Animal Breeding and Genetics, University of Veterinary Medicine Vienna, Vienna 1210, Austria. [2]Institute of Hygiene and Applied Immunology, Center for Pathophysiology, Infectiology and Immunology, Medical University of Vienna, Vienna 1090, Austria. [3]Central Science Laboratory, College of Science and Engineering, University of Tasmania, Sandy Bay, TAS 7005, Australia. [4]CeMM Research Center for Molecular Medicine of the Austrian Academy of Sciences, Vienna 1090, Austria. [5]Menzies Institute for Medical Research, University of Tasmania, Hobart, TAS 7000, Australia. [6]Department of Chemical & Physical Sciences, University of Toronto Mississauga, Mississauga, ON L5L 1C6, Canada. [7]Department of Biosciences & Medical Biology, Paris Lodron Universität Salzburg, Salzburg 5020, Austria. ✉E-mail: andreas.bergthaler@meduniwien.ac.at; richard.moriggl@plus.ac.at

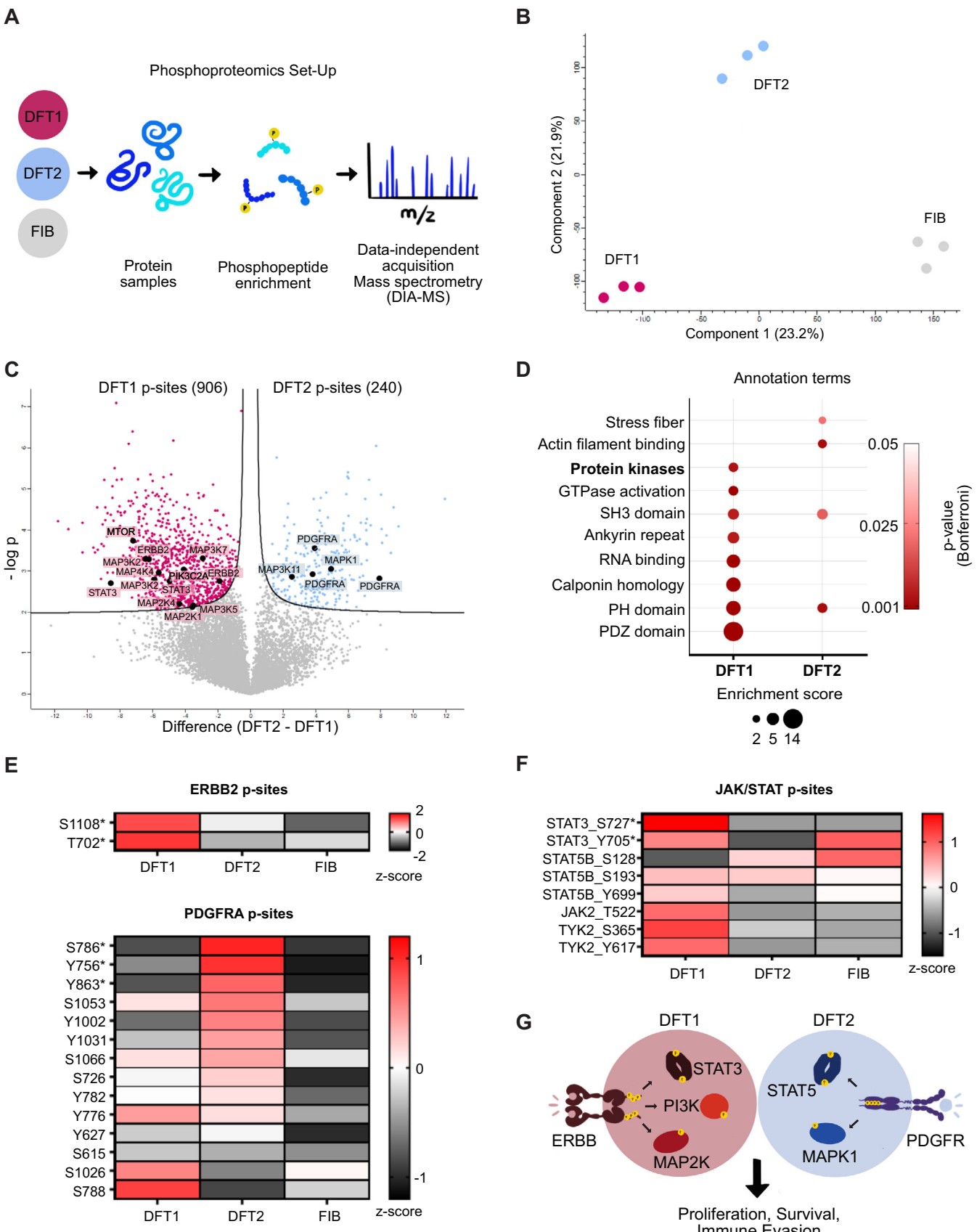

**Figure 1.  DFT1 and DFT2 exhibit unique kinase phosphorylation signatures.**

(A) Experimental design for phosphoproteomic analysis using data-independent acquisition mass spectrometry (DIA-MS). (B) Principal component analysis (PCA) of phosphoproteomics data showing separation between DFT1 (pink), DFT2 (blue), and healthy fibroblast (FIB, grey) samples. (C) Volcano plot comparing DFT1 and DFT2 phosphoproteomics. Significant proteins of interest are highlighted in black. Two-sample $t$ test with permutation-based FDR correction (FDR < 0.05, $s_0 = 0.1$) was used to identify significantly altered phosphopeptides. Data represent $n = 3$ independent replicates for DFT1 and $n = 3$ independent replicates for DFT2. (D) Bubble matrix plot of enriched annotation terms in significant phosphopeptide sets for DFT1 and DFT2. Terms were identified via DAVID Bioinformatics and enrichment was assessed using DAVID's modified Fisher's exact test (EASE score), with Bonferroni-corrected $P$ values < 0.05 and enrichment scores >2. (E, F) Heatmaps of relative abundance (z-scored log2 intensity values) of p-peptides mapped to ERBB2, PDGFRA and JAK/STAT pathway members. Stars indicate significant differences between DFT1 and DFT2 ($t$ test, FDR < 0.05, s0 = 0.1). (G) Illustration of phosphorylation differences between DFT1 and DFT2.

tumours demonstrated phenotypic plasticity post-vaccination, suggesting that Schwann cell plasticity may act as an immune evasion mechanism (Patchett et al, 2021). Pharmacological inhibition of oncogenic drivers can effectively target conserved signalling pathways across species, potentially limiting their adaptive capacity. While deploying small molecule inhibitors in field settings is challenging, these studies provide crucial insights into DFT1 and DFT2 biology and may guide future vaccine development.

Both DFT1 and DFT2 are driven by receptor tyrosine kinases (RTKs), making tyrosine kinase inhibition a promising therapeutic strategy with demonstrated efficacy in both human and veterinary oncology (London, 2009; Stammnitz et al, 2018; Kosack et al, 2019). In DFT1, copy number gains of *ERBB3*, activation of ERBB2/ERBB3, and downstream signalling through the STAT3 pathway promote tumour growth while suppressing *MHCI* expression. Notably, the ERBB inhibitor sapitinib (AZD-8931) has been shown to selectively kill DFT1 cells in a mouse xenograft model and restore *MHCI* expression in vitro (Stammnitz et al, 2018; Kosack et al, 2019). In DFT2, copy number amplifications suggest *PDGFRA* as a key driver. The absence of mutations in other well-known oncogenes and in vitro vulnerability further supports PDGFRA as a candidate target (Stammnitz et al, 2023, 2018). However, DFT2 has not yet been targeted pharmacologically in an in vivo model. The differences between DFT1 and DFT2 driver genes may result in fundamental distinctions in core signalling pathways, with unexplored phosphorylation dynamics contributing to their differential tyrosine kinase inhibitor (TKI) vulnerabilities. These distinct phosphorylation patterns could provide insights into disease progression and establishment, as altered phosphorylation is implicated in many diseases (Lahiry et al, 2010). Additionally, they may uncover new therapeutic targets and offer clues about the broader impact of pathway inhibition.

In this study, we compared the phosphoproteomes of DFT1 and DFT2 to identify key phosphorylated cancer pathways and explore differences in their signalling landscapes. We show that tyrosine kinase inhibition reduced DFT2 tumour growth in a xenograft mouse model. However, prolonged TK inhibition induced resistance in DFT1 and DFT2 cell lines. DFT1 cells displayed a rapid adaptation to sapitinib, targeting ERBB tyrosine kinases, with minimal accompanying genomic or phosphoproteomic changes. In contrast, DFT2 cells showed a more gradual adaptation to the selective kinase inhibitor imatinib, involving genomic alterations and signalling shifts, including STAT3 activation, reminiscent of features observed in DFT1. Notably, combination therapies

exhibited synergistic effects in DFT1 cells, suggesting a potential strategy to delay reduced drug sensitivity. DFT2 cells were less responsive to such combinations, indicating differential adaptability in the two cell lines.

# Results

## DFT1 and DFT2 exhibit unique kinase phosphorylation signatures

To gain a comprehensive view of phosphorylation events across signalling pathways in DFT1 and DFT2, we employed data-independent acquisition mass spectrometry (DIA-MS) using DFT cell lines and fibroblasts as a non-cancer reference (Fig. 1A).

Phosphopeptide-enriched sample analysis identified 21,218 phosphosites (p-peptides), of which 10,867 were used for statistical analysis. Principal component analysis (PCA) confirmed distinct separation between DFT1, DFT2, and fibroblast p-peptide samples, showing that both transmissible cancers exhibit distinct phosphorylation signatures (Fig. 1B). Differential abundance analysis using $t$ test (FDR < 0.05, s0 = 0.1) identified 906 p-peptides significantly enriched in DFT1 and 240 significantly enriched in DFT2 when comparing both tumour types (Fig. 1C). Functional annotation of these p-peptides revealed that the key differences between DFT1 and DFT2 lie in the phosphorylation of cytoskeletal proteins, including stress fibers, actin filaments, and ankyrin repeats, as well as kinase signalling pathways (Fig. 1D). Using the InterPro term IPR011009, to classify kinases within the significant DFT1 and DFT2 p-peptides signatures, 31 kinases were found to be more significantly phosphorylated in DFT1, compared to only 8 in DFT2. Notably, two ERBB2 p-peptides were identified as part of the DFT1 signature, while three PDGFRA p-peptides were included in the DFT2 signature, supporting the role of these receptors as divergent drivers in these cancers (Table 1) (Kosack et al, 2019; Stammnitz et al, 2023, 2018). Closer examination of all p-peptides mapped to these proteins of interest revealed that both p-peptides associated with ERBB2 were more abundant in DFT1, whereas the majority of the 14 identified p-peptides mapping to PDGFRA were elevated in DFT2 (Fig. 1E). To explore the downstream effects of these differences in receptor phosphorylation, we examined the phosphorylation of key pathways shared by both ERBB and PDGFR signalling, including JAK/STAT3/5, PI3K-AKT-mTOR, and MAPK-ERK pathways. Our analysis revealed distinct phosphorylation patterns across all three pathways in DFT1 and DFT2, despite both tumours utilizing these same pathways. Notably, p-peptides

**Table 1.** (Related to Fig. 1): Differentially phosphorylated peptides in DFT1 and DFT2 mapped to kinases.

| Tumour | Gene | Protein description | Protein name | PTM* multiplicity | PTM* site (AA) | PTM* site location | PTM* flanking region |
|---|---|---|---|---|---|---|---|
| DFT1 | AAK1 | AP2 associated kinase 1 | A0A7N4PE76_SARHA | 2 | S | 546 | LGSLTPPSSPKAQRA |
| DFT1 | CDK7 | Cyclin-dependent kinase 7 | A0A7N4PWF9_SARHA | 1 | T | 170 | GSPNRAYTHQVVTRW |
| DFT1 | CDKL5 | Cyclin dependent kinase like 5 | G3VW94_SARHA | 1 | S | 746 | GSFYRVPSPRPDNSF |
| DFT1 | CLK3 | CDC like kinase 3 | G3WQI1_SARHA | 1 | S | 68 | RYRERRDSDNYRFEE |
|  |  |  |  | 1 | S | 135 | SSKRSSRSVEDDKEG |
|  |  |  |  | 1 | S | 9 | HHCKRYRSPEQDSYL |
|  |  |  |  | 1 | S | 79 | RFEERSPSFGEDYYS |
| DFT1 | DCLK1 | Doublecortin like kinase 1 | G3WAA2_SARHA | 1 | S | 352 | RSSQHGGSSTSLAST |
|  |  |  |  | 1 | S | 32 | SRVNGLPSPTHSAHC |
| DFT1 | ERBB2** | Receptor protein-tyrosine kinase | G3WQQ2_SARHA | 1 | S | 1108 | GLPPRDLSPLQRYSE |
|  |  |  |  | 1 | T | 702 | TELVEPLTPSGALPN |
| DFT1 | HIPK1 | Homeodomain interacting protein kinase 1 | A0A7N4NZT4_SARHA | 1 | T | 1197 | TGYPLSPTKISQYSY |
| DFT1 | LMTK2 | Protein kinase domain-containing protein | A0A7N4NWP6_SARHA | 1 | S | 1469 | LQTSKYFSPPPPSRS |
| DFT1 | MAP2K1** | Mitogen-activated protein kinase kinase 1 | A0A7N4P620_SARHA | 1 | S | 196 | LIDSMANSFVGTRSY |
| DFT1 | MAP2K4** | Mitogen-activated protein kinase kinase 4 | A0A7N4PLS5_SARHA | 1 | S | 271 | ISGQLVDSIAKTRDA |
| DFT1 | MAP3K2** | Mitogen-activated protein kinase kinase 2 | G3VVK8_SARHA | 2 | T | 159 | LPVIGPTTRDRSSPP |
|  |  |  |  | 2 | T | 158 | RLPVIGPTTRDRSSP |
| DFT1 | MAP3K5** | Mitogen-activated protein kinase kinase | A0A7N4NIK2_SARHA | 1 | S | 1010 | EDHSAPPSPEEKDSG |
| DFT1 | MAP3K7** | Mitogen-activated protein kinase kinase 7 | G3WZI8_SARHA | 1 | S | 452 | SVTGTEPSQVSSRSS |
| DFT1 | MAP4K4** | Mitogen-activated protein kinase kinase kinase 4 | G3VHS5_SARHA | 1 | T | 992 | ETQSASNTLQKHKSS |
| DFT1 | MARK1 | non-specific serine/threonine protein kinase | A0A7N4V6S6_SARHA | 1 | T | 252 | TIGNKLDTFCGSPPY |
| DFT1 | MARK3 | Non-specific serine/threonine protein kinase | A0A7N4V4B0_SARHA | 1 | S | 400 | LNNSTGQSPHHKVQR |
| DFT1 | MTOR** | Serine/threonine-protein kinase mTOR | A0A7N4PQE2_SARHA | 1 | S | 1246 | PMKKLHVSTINLQKA |
| DFT1 | NUAK1 | NUAK family kinase 1 | A0A7N4V1L9_SARHA | 1 | S | 472 | RTGAPLKSPVETEGA |
| DFT1 | PIK3C2A** | Phosphatidylinositol-4-phosphate 3-kinase catalytic subunit type 2 alpha | G3VD73_SARHA | 1 | T | 1557 | DANPLSPTSGQVGGA |
| DFT1 | PRKD1 | Serine/threonine-protein kinase | A0A7N4P9B8_SARHA | 2 | S | 205 | GVRRRRLSNVSLTGL |
|  |  |  |  | 1 | S | 405 | DDSNRMISPSTSNNI |
| DFT1 | PRKG1 | cGMP-dependent protein kinase | A0A7N4PZK6_SARHA | 1 | Y | 555 | TFCGTPEYVAPEIIL |
| DFT1 | RAF1 | Non-specific serine/threonine protein kinase | A0A7N4NGZ1_SARHA | 1 | S | 528 | QVEQPTGSILWMAPE |
|  |  |  |  | 1 | S | 377 | MLSTRIGSGSFGTVY |
| DFT1 | RPS6KA3 | Ribosomal protein S6 kinase | G3W972_SARHA | 1 | S | 724 | SALNRNQSPVLEPVG |
| DFT1 | RPS6KC1 | Non-specific serine/threonine protein kinase | A0A7N4PG14_SARHA | 1 | S | 620 | SLSRSKNSPMAFFRI |
| DFT1 | STK10 | Non-specific serine/threonine protein kinase | G3WHY4_SARHA | 1 | S | 461 | KINKGSRSRPTSSIL |
|  |  |  |  | 1 | S | 963 | AECPNPNSPNKPAKF |
|  |  |  |  | 1 | T | 185 | VSAKNLKTLQKRDSF |
| DFT1 | TAOK3 | TAO kinase 3 | A0A7N4NSY2_SARHA | 1 | S | 373 | FATIKSASLVTRQIH |
| DFT1 | TEX14 | Testis expressed 14, intercellular bridge forming factor | A0A7N4NSD8_SARHA | 1 | S | 587 | CQPDSPQSPSALPEK |

**Table 1.** (continued)

| Tumour | Gene | Protein description | Protein name | PTM* multiplicity | PTM* site (AA) | PTM* site location | PTM* flanking region |
|---|---|---|---|---|---|---|---|
| DFT1 | TNIK | TRAF2 and NCK interacting kinase | G3WC40_SARHA | 1 | S | 684 | PQRTTSISPALARKN |
|  |  |  |  | 1 | S | 773 | ANTKSEGSPVLPHEP |
|  |  |  |  | 1 | S | 711 | GSQPIRASNPDLRRT |
|  |  |  |  | 1 | T | 585 | SGVQPARTPPMLRPV |
|  |  |  |  | 1 | S | 692 | PALARKNSPGNGNAL |
| DFT1 | ULK1 | Non-specific serine/threonine protein kinase | A0A7N4P1Y3_SARHA | 1 | S | 518 | PGQSRVPSPQGTEMC |
| DFT1 | VRK1 | VRK serine/threonine kinase 1 | A0A7N4P3D5_SARHA | 1 | S | 376 | EMEESIESGAEDMEI |
| DFT1 | VRK2 | VRK serine/threonine kinase 2 | G3W0L6_SARHA | 1 | S | 373 | NKSKEERSAESHVIW |
| DFT1 | WNK1 | Non-specific serine/threonine protein kinase | A0A7N4PI27_SARHA | 1 | S | 2467 | NMDDGSGSPHSTHHL |
|  |  |  |  | 1 | S | 179 | AASNHVGSKEEPAAA |
| DFT1 | WNK4 | Non-specific serine/threonine protein kinase | G3WI41_SARHA | 1 | S | 1074 | LQGETRLSPITEEGK |
| DFT2 | CDK13 | Cyclin dependent kinase 13 | A0A7N4NWZ2_SARHA | 1 | S | 433 | RSRHSSISPSTLTLK |
| DFT2 | CDK14 | Cyclin dependent kinase 14 | A0A7N4P5N5_SARHA | 1 | S | 95 | TQVKRVHSENNACIN |
| DFT2 | LOC100919463 | Protein kinase domain-containing protein | G3WC64_SARHA | 1 | S | 221 | SDPERKLSLVGSAFW |
| DFT2 | MAP3K11** | Mitogen-activated protein kinase kinase kinase | G3VMW0_SARHA | 1 | S | 728 | ETRGGAVSPPPGAFR |
| DFT2 | MAPK1** | Mitogen-activated protein kinase | A0A7N4P2U5_SARHA | 1 | Y | 175 | HTGFLTEYVATRWYR |
| DFT2 | PDGFRA** | Platelet-derived growth factor receptor alpha | G3WJ82_SARHA | 1 | S | 786 | PASYKKKSVSDPEVK |
|  |  |  |  | 1 | Y | 863 | DIMHDSNYVSKGSTF |
|  |  |  |  | 1 | Y | 756 | KQADTTQYVPMLERK |
| DFT2 | PI4KB | Phosphatidylinositol 4-kinase beta | G3WMQ8_SARHA | 1 | S | 506 | GDIRRRLSEQLAHTP |
| DFT2 | PKN1 | protein kinase C | A0A7N4NVX6_SARHA | 1 | S | 940 | PREARPLSAAEQEAF |

Phosphorylation sites with significant differential abundance between DFT1 and DFT2 cell lines are shown, including site location and flanking sequences. Peptides were filtered from all differentially phosphorylated peptides (FDR < 0.05, $s_0 = 0.1$; see Fig. 1C and Source Data) and annotated using the InterPro term IPR011009 (Protein kinase-like domain) via DAVID Bioinformatics. *All post-translational modifications (PTMs) listed in this table are phosphorylation events. **Peptides of interest are highlighted in Fig. 1C.

mapping to the JAK/STAT pathway were more abundant in DFT1, consistent with a previously established role in this tumour type (Kosack et al, 2019). STAT3 exhibited markedly higher levels of phosphorylation at S727 and Y705. The abundance of STAT5B p-peptides varied across the analysed cell types (Fig. 1F). Differences were also observed in the phosphorylation of the PI3K-AKT-mTOR and MAPK-ERK pathways, which share common downstream effects, including the phosphorylation of STAT3 and STAT5B at key serine and tyrosine residues (Fig. EV1) (Johnson et al, 2023).

Gene expression and protein production profiling, combined with staining of recent tumour biopsies, expanded our understanding of the signalling differences between DFT1 and DFT2 (Fig. EV2; Appendix Fig. S1). Surprisingly, both mRNA and protein levels of ERBB3 were not only elevated in DFT1, where it has been shown to have a detrimental role in cancer cell signalling (Kosack et al, 2019), but also in DFT2. Western blot analysis confirmed that ERBB3 is also phosphorylated in both tumour types, suggesting its activation despite not being detected in our

phosphoproteomics analysis. Importantly, PDGFRA was markedly upregulated in DFT2, while PDGFRB expression was more prevalent in DFT1, aligning with previously reported gene duplication events (Stammnitz et al, 2023, 2018). Increased levels of both total STAT3 and its phosphorylated forms (Y705 and S727) were confirmed in DFT1 cell lines compared to DFT2. In contrast, STAT5 expression was elevated in DFT2 at both the mRNA and protein levels, consistent with observations in the biopsies (Fig. EV2A–D; Appendix Fig. S1). In DFT2 cells, STAT5 phosphorylated at S128 and S193 may act as a downstream mediator of PDGFRA signalling, whereas in DFT1 cells, STAT3 functions downstream of ERBB signalling.

In conclusion, our findings reveal key differences in core cancer signalling between DFT1 and DFT2. Notably, both tumours express ERBB and PDGFR proteins, yet exhibit differential phosphorylation of ERBB2 and PDGFRA. These upstream variations likely drive the distinct kinase regulation observed within common downstream cancer pathways and suggest unique therapeutic vulnerabilities for each tumour type (Fig. 1G).

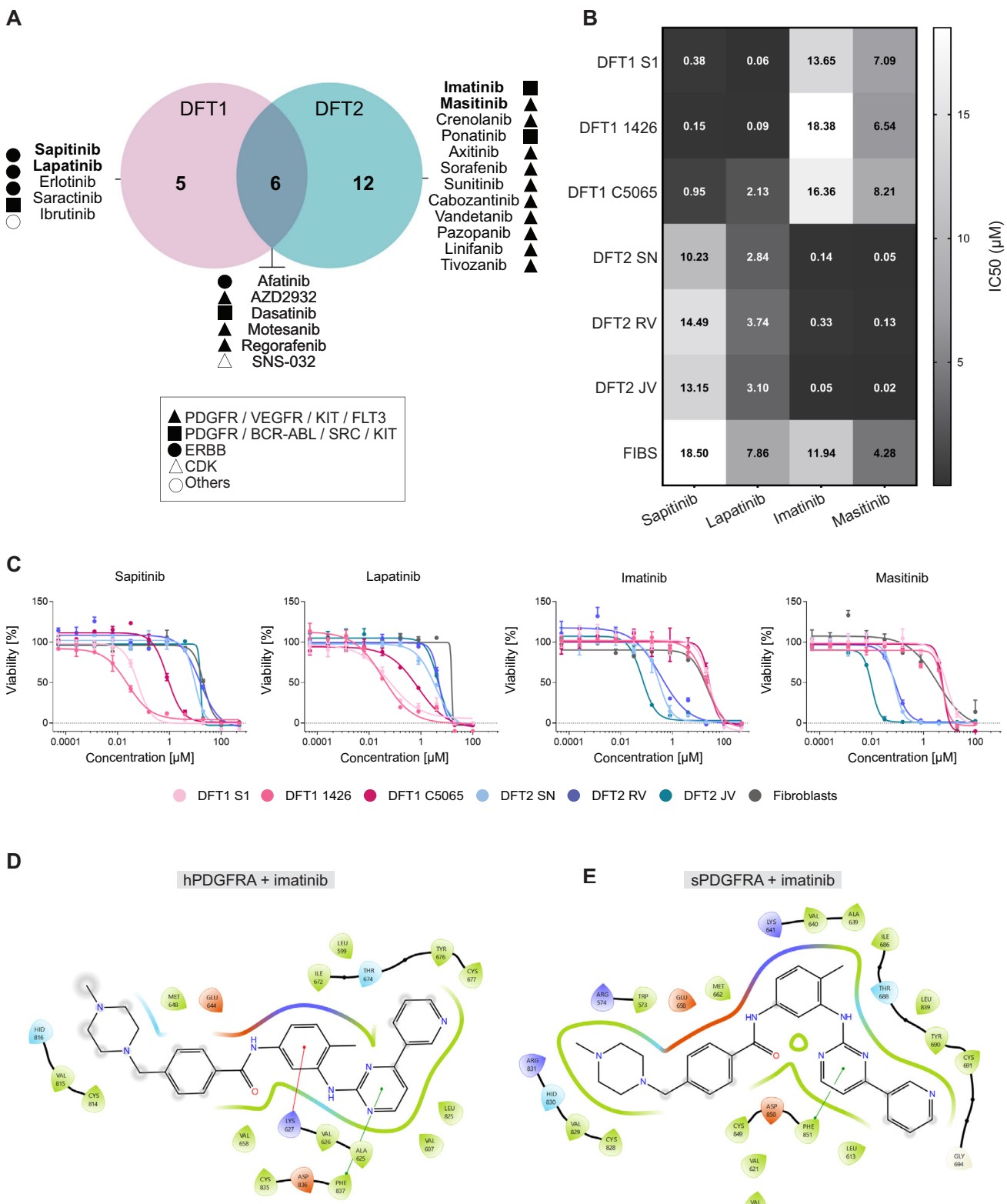

◄ **Figure 2. Distinct kinase activation drives TKI responses in DFT1 and DFT2.**

(A) Venn diagram showing hits from a focused tyrosine kinase inhibitor (TKI) drug screen that met selection criteria in DFT1 and/or DFT2 cell lines compared to healthy fibroblasts. Target specificity of inhibitors is indicated using category-specific icons shown in the figure legend. For details, refer to Dataset EV1. (B) Heatmap showing mean $IC_{50}$ values (μM) for four selected drug hits, tested in triplicate across six DFT cell lines and one healthy fibroblast control. Values represent means from three independent experiments. (C) Representative dose–response curves for sapitinib, imatinib, lapatinib, and masitinib across six DFT cell lines and the fibroblast control. Data are presented as normalised percent of control (POC), shown as mean ± SD. (D, E) Residues of human PDGFRA (hPDGFRA, D) and Tasmanian devil PDGFRA (sPDGFRA, E) that interact with imatinib at the catalytic site of the kinase domain. Structural data and electron density maps for the hPDGFRA kinase domain bound to imatinib were obtained from the RCSB Protein Data Bank (PDB ID: 4RIX).

## Distinct kinase activation is associated with TKI responses in DFT1 and DFT2

Many TKIs have already been tested in DFT1 in vitro, while a smaller subset was evaluated in DFT2 previously (Kosack et al, 2019; Stammnitz et al, 2018). Building on new insights from our phosphoproteomics data, we conducted a focused TKI screen to systematically compare kinase inhibitor responses between DFT1 and DFT2. We screened a curated panel of 48 TKIs across three cell lines per tumour type, with fibroblasts serving as controls (Appendix Fig. S2; Dataset EV1). Inhibitors were selected based on clinical relevance in human oncology and prior screening data. This approach allowed us to both validate reported hits and to identify novel responses, contributing to a broader understanding of kinase signalling and dependencies in the cell lines.

The drug screen revealed tumour-type–biased vulnerabilities to TKIs in DFT1 and DFT2 (Fig. 2A). In line with the phosphorylation profiles described above, DFT1 cell lines were selectively sensitive to ERBB inhibitors such as sapitinib and lapatinib, which showed no significant activity in DFT2 cell lines or fibroblasts. Conversely, DFT2 cells responded to inhibitors such as imatinib and masitinib, which are known to target multiple kinases including PDGFR, VEGFR, ABL, SRC and KIT (Fig. 2B,C). The observed sensitivity of DFT2 cells to these kinase inhibitors may be mediated through PDGFRA, which is markedly upregulated, phosphorylated, and genomically amplified in these cells. Some compounds, such as afatinib and several multi-kinase inhibitors, showed activity in both DFT1 and DFT2. To maintain mechanistic focus on compounds with selective efficacy, we chose sapitinib as the representative core cancer pathway inhibitor in DFT1, supported by prior efficacy data (Kosack et al, 2019). We chose imatinib for DFT2, based on clinical applicability and substantial $IC_{50}$ shifts relative to DFT1 and fibroblasts.

To assess whether both TKIs, developed for human cancers, retain binding efficacy in Tasmanian devils, we performed structure-based modelling. Proteins are prefixed with "s" to indicate *Sarcophilus harrisii* (Tasmanian devil) origin and "h" to indicate the corresponding human proteins. The Tasmanian devil PDGFRA (sPDGFRA) and human PDGFRA (hPDGFRA) kinase domains displayed high amino acid sequence similarity ( > 97%) with a very low RMSD (0.122). Homology modelling of sPDGFRA into the electron density map of imatinib-bound hPDGFRA showed that key polar contacts, such as THR-674, ASP-637, GLU-644, and the backbone of CYS-677, are conserved, supporting the premise that imatinib binds Tasmanian devil PDGFRA in a manner analogous to its human counterpart (Figs. 2D,E and EV3A,B). Similarly, structural alignment of human and

Tasmanian devil ERBB3 kinase domains revealed a high degree of similarity (RMSD = 0.53), and docking of sapitinib into the ATP-binding pocket showed conserved binding poses and energetics (−10.5 and −10.3 kcal/mol, respectively). Key interactions, including those with LEU-790, LEU-841, THR-787, GLN-788, and CYS-740 (adenine-mimetic site), and ARG-838, ASN-839, and GLY-716 (phosphate-mimetic group), were preserved in the Tasmanian devil ERBB3 model (Fig. EV3C–G). These findings reinforce that kinase domain structures and drug–target interactions are conserved across species, supporting the use of these inhibitors in functional studies of DFT1 and DFT2.

## DFT2 tumours respond to tyrosine kinase inhibition in xenograft mouse models

Aiming to validate the tumour-reducing effects of TKI on DFT tumours in an in vivo setting, we established DFT2 xenograft mouse models in immunodeficient NOD scid gamma (NSG) mice, which are mice lacking functional T, B, and NK cells, thus enabling the engraftment of foreign tumour cells. DFT2 xenograft tumours exhibited rapid growth, high vascularity, and a pulpy consistency, in stark contrast to the firm nodules and slower proliferation observed in DFT1 xenograft tumours (Kosack et al, 2019) (Fig. 3A). This aligns with the known higher proliferation rate of DFT2 (Stammnitz et al, 2023).

We employed oral imatinib treatment to investigate whether DFT2 xenografted mice can be effectively treated through kinase inhibition (Fig. 3B). Imatinib treatment resulted in a reduction in both tumour volume and weight in DFT2 xenografts after 8 days (Fig. 3C). Furthermore, imatinib-treated tumours exhibited not only reduced size but also a change in texture and colour (Fig. 3D). IHC staining revealed altered tissue structure with reduced cell proliferation (Ki67) and vascularization (VWF) in the treated tumours. Dispersed cleaved Caspase-3 (CC3)-positive cells in treated tumours suggest apoptosis due to treatment. In contrast, in untreated tumours, a high density of CC3-positive cells were observed in the tumour core, likely resulting from rapid growth outpacing vascular support (Fig. 3E–I).

These findings demonstrate that DFT2 cells can be effectively targeted by imatinib in xenograft mouse models.

## DFT2 acquires more genomic disruptions than DFT1 to gain TKI resistance

To assess resistance evolution and signalling plasticity under long-term tyrosine kinase inhibitor (TKI) treatment, we generated sapitinib-resistant DFT1 and imatinib-resistant DFT2

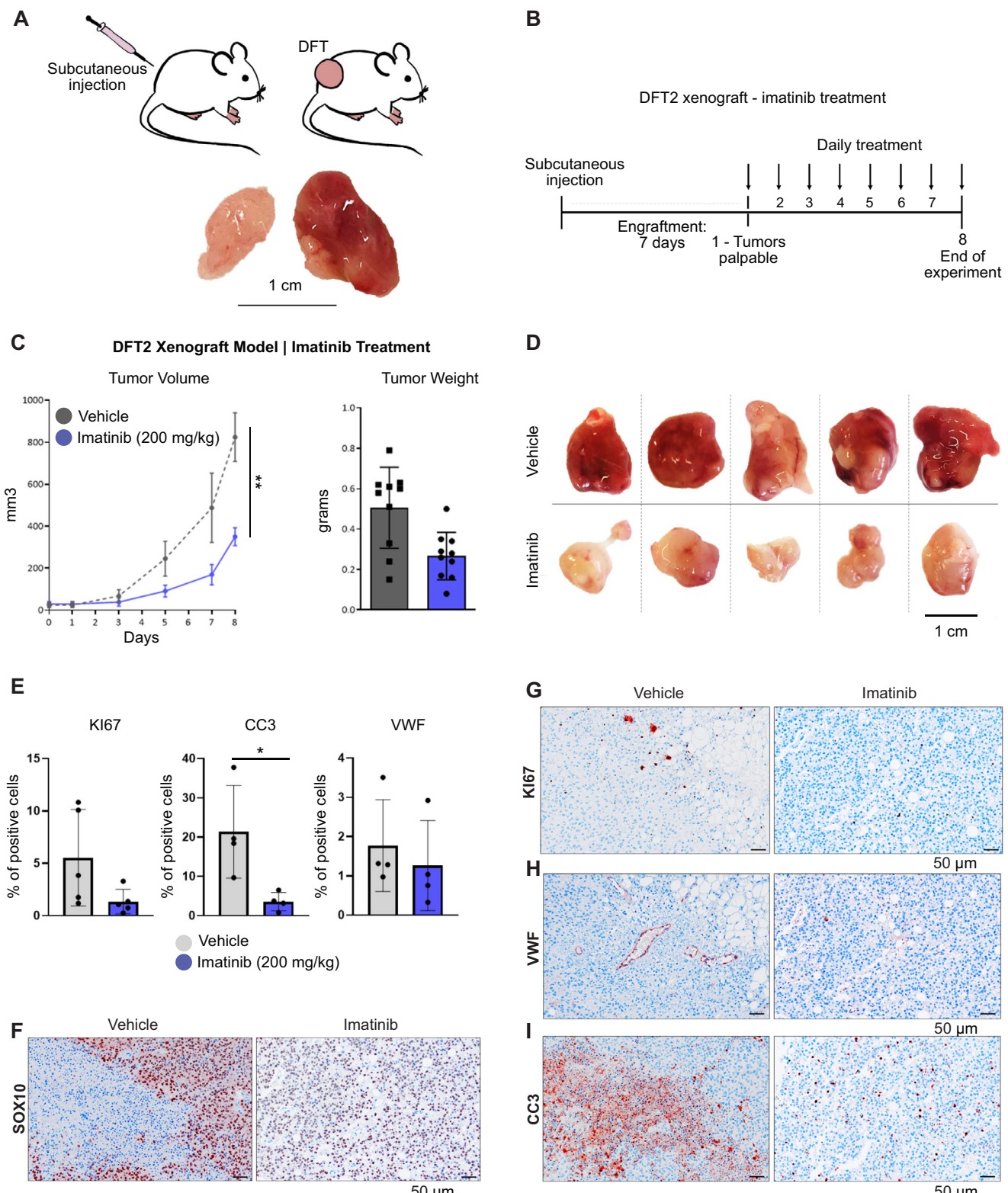

cell lines (Fig. EV4A). Two independent resistant lines per tumour type (R1, R2) were established via gradual dose escalation over 6 months. DFT1 cells developed resistance more rapidly than DFT2 (Fig. 4A). Drug response profiling showed that

resistant DFT1 cells exhibited cross-resistance to the ERBB inhibitor lapatinib but remained sensitive to selective kinase inhibitors imatinib and masitinib. Imatinib-resistant DFT2 cells showed reduced sensitivity to both imatinib and masitinib,

**Figure 3. DFT2 tumours respond to tyrosine kinase inhibition in xenograft mouse models.**

(A) Illustration of xenograft model establishment. DFT1 or DFT2 cells were injected subcutaneously into NSG mice, leading to bilateral tumour formation. Representative images of untreated DFT1 and DFT2 xenograft tumours are shown. Scale bar: 1 cm. (B) Imatinib treatment set-up: 7 days post-DFT2 transplantation, mice were treated with vehicle or 200 mg/kg imatinib daily for 8 days ($n = 10$ per group). (C) Tumour volume and weight in DFT2-transplanted mice treated with imatinib or vehicle. Data shown are from one of two representative experiments (mean ± SEM, two-way ANOVA with Bonferroni's test). Statistical significance was set at $P < 0.05$ and is indicated as follows: $P < 0.0332$ (*), $P < 0.0021$ (**), $P < 0.0002$ (***), and $P < 0.0001$ (****). Exact $P$ value = 0.0013. (D) Representative images of DFT2 tumours from vehicle- and imatinib-treated mice at the end of the experiment. Scale bar: 1 cm. (E) Quantification of Ki67, cleaved caspase-3 (CC3), and von Willebrand Factor (VWF) in DFT2 tumour tissue. Data are shown as mean ± SD, with error bars representing the standard deviation. Statistical analysis was performed using an unpaired $t$ test. Statistical significance was set at $P < 0.05$ and is indicated as follows: $P < 0.0332$ (*), $P < 0.0021$ (**), $P < 0.0002$ (***), and $P < 0.0001$ (****). For CC3, the exact $P$ value was $P = 0.0253$. (F–I) H&E and IHC staining for SOX10, Ki67, von Willebrand factor (VWF) and cleaved caspase 3 (CC3) in DFT2 tumour tissue. Images show contiguous sections; Scale bars: 50 µm. Source data are available online for this figure.

consistent with broad resistance to PDGFR-targeting drugs (Appendix Fig. S3A,B).

Whole-genome sequencing (WGS) revealed that DFT2-resistant lines had ~60% more significantly altered SNVs and twice as many gained or lost variants compared to DFT1 (Table 2; Datasets EV2 and 3). A higher proportion of novel SNVs also went to fixation in DFT2, indicating stronger selection under drug pressure (Fig. EV4B).

Copy number analysis revealed no major changes in resistant DFT1 lines, aside from two regions on chromosome 1 that gained an additional copy (approximately 10 Mb at position 265 Mb and 9 Mb at position 68 Mb) (Fig. 4B; Dataset EV4).

In contrast, resistant DFT2 lines displayed widespread copy number alterations and evidence of ploidy shifts (Fig. 4C; Dataset EV4). SNV allele frequency spectra and B allele frequency (BAF) profiles revealed peaks consistent with tetraploidy on chromosomes 1 and 4 (¼, ½, ¾) and triploidy on chromosome 6 (⅓, ⅔) (Fig. 4D; Appendix Fig. S3C). Control-FREEC modelling confirmed tetraploidy as the best-fitting ploidy model, supported independently by nQuack analysis (Gaynor et al, 2024) using 10,000 randomly selected SNVs per chromosome (Dataset EV5). Both resistant lines also showed loss on a part of chromosome 1 and a focal gain on chromosome 2 (~20 Mb at position 550 Mb) (Dataset EV4). Together, these data support a tetraploid genome in resistant DFT2 cells, in contrast to the parental line, for which we found little indication of tetraploidy. Accordingly, copy number variation (CNV) calling for the parental DFT2 line was performed using a diploid model.

Several genes of interest showed notable estimated copy number alterations associated with resistance in DFT2. *ERBB3*, which exists in three copies in DFT1 (Stammnitz et al, 2018), was estimated to be present at four and five copies in resistant DFT2 cell lines R1 and R2, respectively (Fig. 4E). *EGFR*, located on chromosome 1, maintained an estimated three copies in both resistant DFT2 lines (Fig. EV4C). *ERBB2* and *ERBB4* were each estimated to have four copies in resistant DFT2 cells, with *ERBB4* further amplified to five copies in resistant cell line R2 (Fig. EV4D,E). These patterns together point to a potential role of ERBB signalling in DFT2 resistance.

Interestingly, *PDGFRA* was estimated to exist in nine to ten copies in resistant DFT2 cells, despite surrounding regions on chromosome 6 showing copy number loss following the presumed duplication (Fig. 4F). In contrast, *PDGFRB* showed no further copy number change in DFT2; however, in DFT1, an already amplified region was further increased in resistant DFT1 cells (Fig. EV4F).

Strikingly, both independently derived resistant lines appear to have followed similar mutation patterns, including an inferred whole-genome duplication in DFT2-resistant cells. This may reflect either an intrinsic propensity for genome doubling under drug pressure or the expansion of pre-existing tetraploid subpopulations with a selective advantage during imatinib exposure. Taken together, these inferred genomic changes point to distinct routes to resistance in DFT1 and DFT2, shaped by differing levels of genomic plasticity.

## Imatinib resistance drives oncogenic pathway switching in DFT2 cells

To further characterise changes on expression and post-translational levels, we analysed both resistant and parental cell lines using in vitro assays and phosphoproteomics via DIA-MS (Fig. 5).

Although gene expression analysis showed a downregulation of *ERBB3* in resistant DFT1 cell lines, *PDGFRA/B* and *STAT3/STAT5* gene and protein expression levels remained relatively stable, suggesting that STAT activation is maintained through alternative signalling pathways, including upstream PDGFR signalling (Figs. 5A, and EV5A,B). In DFT2 resistant cell lines, decreased *PDGFRA, PDGFRB* and *ERBB3* expression was accompanied by a striking 20-fold increase in *EGFR* expression (Fig. 5A). While total STAT3 levels were unchanged, phosphorylation, especially at Y705, as also observed in DFT1, was increased, suggesting enhanced transcriptional activity via STAT3 parallel dimer formation and DNA binding action (Fig. EV5A,C).

Global changes in the respective phosphoproteomes revealed that while DFT1 cells exhibited minimal variation, resistant DFT2 cells diverged from their parental counterparts (Fig. 5B). Consistent with these findings, differential abundance analysis using $t$ test (FDR < 0.05, s0 = 0.1) comparing parental and resistant DFT1 phosphoproteomes identified no p-peptides significantly different between parental and resistant cells (Fig. 5C,E). Comparing parental and resistant DFT2 cells (FDR < 0.05, s0 = 0.1) revealed more extensive changes, with 627 p-peptides enriched in resistant cells compared to 48 in parental cells (Fig. 5D, Source Data). P-peptides enriched in resistant DFT2 cells were associated with SH3 and PDZ domains, as well as protein kinases, most notably showing significant enrichment of the ERBB signalling pathway, with marked phosphorylation changes across multiple pathway components (Fig. EV5D,E; Table 3). Notably, many phosphorylation changes in downstream JAK/STAT proteins were observed,

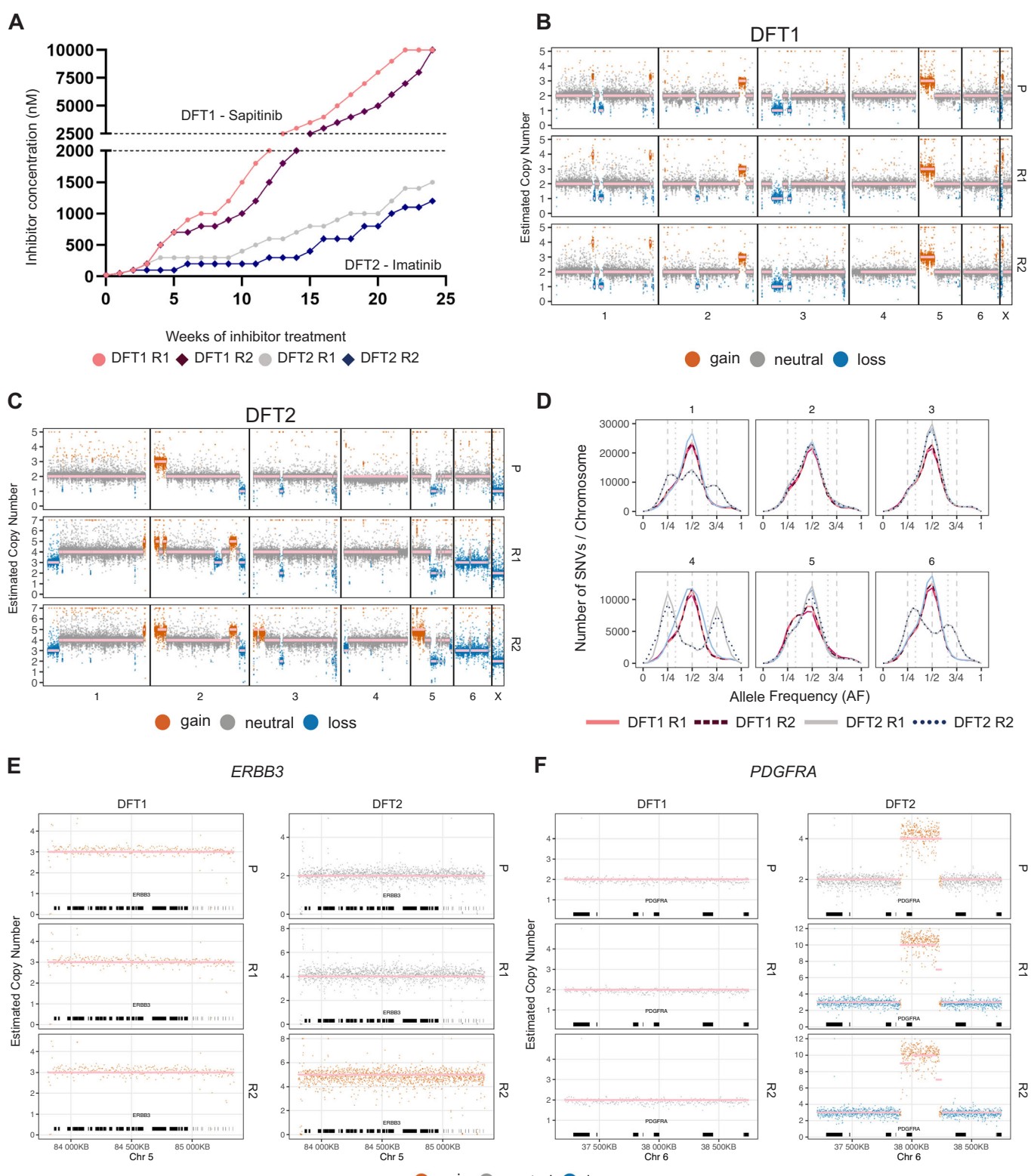

including increased levels of STAT3 (Y705) p-peptides. STAT5 phosphorylation showed a decrease at S128, accompanied by increases at S193 and Y699. Additionally, p-peptides mapping to JAK2 and TYK2 were more abundant (Figs. 5F and EV5A). These changes coincided with the downregulation of immune recognition genes, including *MHCI, B2M* and *TAP1* in resistant DFT2 cells (Fig. 5G). Strikingly, these molecular changes mirror core cancer signalling of DFT1 cells, where sustained STAT3 activation has been linked to MHCI downregulation and immune evasion (Kosack et al, 2019).

Figure 4.   DFT2 acquires more genomic disruptions than DFT1 to gain TKI resistance.

(A) Resistance development over time. Two independent DFT1 cell lines were cultured with gradually increasing concentrations of sapitinib, and DFT2 cell lines with increasing concentrations of imatinib. Cells were maintained as bulk populations without clonal selection. DFT1 cells acquired resistance at 10 µM sapitinib, while DFT2 cells reached resistance at 1 µM imatinib. (B–F) Whole genome sequencing analysis of resistant DFT1 and DFT2 cell lines compared to their parental lines. (D) Allele frequency spectra of SNVs detected on each chromosome in parental and resistant cell lines. The dotted horizontal lines indicate allele frequencies expected in triploid (0.33 and 0.66) and tetraploid (0.25, 0.5 and 0.75) cases. The frequency distributions of SNVs on chromosomes 1, 4 and 6 in the two resistant DFT2 cell lines point to substantial ploidy alterations. (B, C) Copy number (CN) as estimated by Control-FREEC across chromosomes 1–6 and X for DFT1 (B) and DFT2 (C) parental cell lines and their resistant derivatives (R1 and R2). Each dot represents the copy number ratio, that is the normalised read count within a 50 kbp genomic window, calculated as the read count divided by the genome-wide median, multiplied by the ploidy (ploidy 2: DFT1 P/R1/R2 and DFT2 P; ploidy 4: DFT2 R1 and R2). CN gains (red) and losses (blue) were inferred from the median signal across adjacent windows by Control-FREEC. The pink line represents estimated copy number inferred by Control-FREEC. (E, F) Copy number profiles for *ERBB3* (E) and *PDGFRA* (F). Normalised CN ratios were estimated by Control-FREEC using 5 kbp or 1 kbp genomic windows for DFT1 and DFT2, respectively. Genes are indicated as black bars. Source data are available online for this figure.

Overall, DFT1 cells rapidly developed resistance to sapitinib and ERBB inhibition, characterised by no significant phosphorylation changes. DFT2 cells show a slower progression of resistance to imatinib, accompanied by fundamental signalling changes that mirror the core cancer signalling pathways of DFT1.

## Sapitinib and imatinib combination treatment synergizes in DFT1

Given the observed plasticity and adaptability in the resistant DFT cell lines, we explored the potential of combination therapy by simultaneously targeting distinct core cancer pathways with sapitinib and imatinib.

Combination treatment demonstrated synergistic effects in DFT1 and DFT2 cell lines after 72 h of treatment, as indicated by most synergistic area (MSA) scores around 10, calculated using the Bliss and Loewe models (Fig. 6A). These scores provide insights into drug interactions, identifying combinations that either enhance or inhibit each other's effects (Ianevski et al, 2022). Notably, the strongest synergy was observed in the DFT1 C5065 cell line, which also displayed the highest PDGFRB protein expression, suggesting that the efficacy of the combination treatment correlates with PDGFRB expression levels (Fig. 6B). Consistent with this, the combination therapy significantly reduced IC$_{50}$ values in DFT1 cells compared to either agent alone (Fig. 6C). In DFT2 this combination did not lower IC$_{50}$ values compared to single-agent treatments after 72 h (Fig. 6D). Gene expression analysis following 24-hour treatments indicated compensatory upregulation of *ERBB3* in response to inhibitor treatments in DFT1 cells (Fig. 6E). However, the combination treatment resulted in a more pronounced downregulation of key genes downstream of ERBB signalling, including *PDGFRB* and *STAT3*, compared to single-agent treatments. These findings suggest that dual inhibition of ERBB and PDGFR may effectively disrupt compensatory mechanisms in DFT1 cells, potentially reducing resistance development (Fig. 6E). In contrast, combination therapy did not reduce expression of the target genes in DFT2 cells after 24 h (Fig. 6F). These limited synergistic effects, together with differences in ZIP synergy score (Appendix Fig. S4), suggest that DFT2 lacks rapid alternative signalling pathways to PDGFRA, consistent with the slower development of resistance observed previously.

Overall, these findings emphasize that the distinct signalling pathways in DFT1 and DFT2 cancers correlate with differences in inhibitor vulnerabilities and subsequent resistance mechanisms. Combination therapies may help prevent these resistance mechanisms in at least DFT1.

## Discussion

The emergence of two transmissible cancers, DFT1 and DFT2, in Tasmanian devils poses a critical conservation challenge, threatening the survival of this endangered species and raising broader concerns about the potential for transmissible cancers to arise in other species (Cunningham et al, 2021). Understanding cancer signalling is crucial for developing effective vaccines and interventions, yet key signalling pathways have not been explored in detail. Here, we characterise differences in kinase-driven signalling pathways between DFT1 and DFT2 by analysing the cancer phosphoproteome, revealing distinct vulnerabilities and mechanisms of resistance in each tumour.

Although DFT1 and DFT2 share a Schwann cell origin (Murchison et al, 2010; Patchett et al, 2020), they display distinct patterns in receptor tyrosine kinase (RTK) signalling, possibly shaped by their tumour-specific adaptations. In line with its likely origin from a myelinating Schwann cell, DFT1 relies heavily on ERBB signalling and shows strong activation of downstream STAT3, contributing to immune evasion via MHCI downregulation (Murchison et al, 2010; Kosack et al, 2019; Lyons et al, 2005; Boerboom et al, 2017; Siddle et al, 2013). In contrast, DFT2 appears to originate from a more dedifferentiated, mesenchymal-like Schwann cell and is primarily driven by PDGFR signalling (Farahani and Xaymardan, 2015; Jessen and Mirsky, 2016; Patchett et al, 2020; Stammnitz et al, 2018). Regulation of key downstream signalling pathways shared by ERBB and PDGFR, such as JAK/STAT3/5, PI3K-AKT-mTOR, and MAPK/ERK (Citri and Yarden, 2006; Zou et al, 2022), appear to differ between DFT1 and DFT2. Notably, DFT2 showed minimal activation of the JAK/STAT pathway while maintaining MHC class I (MHCI) expression. Interestingly, findings link JAK/STAT signalling to immune evasion in cancer (Hu et al, 2021). These differences emphasize the tumours' distinct strategies for survival and immune evasion and offer crucial insights for designing immunotherapeutic approaches.

Our analysis highlighted the PI3K-AKT-mTOR pathway as critical for both cancers, although their activation mechanisms differ. Gene fusions related to this pathway have been identified in both tumours, shedding light on their evolutionary plasticity. In DFT1, the PIK3R2-CMKLR1 fusion may enhance immune evasion and cellular plasticity, while in DFT2, the PIK3R1-ADAMTS6 fusion likely influences PI3K signalling, tumour proliferation and survival (Laffranchi et al, 2024; Mead, 2022; Stammnitz et al, 2023). These distinct gene fusions and phosphorylation patterns highlight

**Table 2. Single nucleotide variant (SNV) and InDel numbers and changes between parental and resistant DFT1 and DFT2 cell lines.**

| | SNV | | | | InDel | | |
|---|---|---|---|---|---|---|---|
| | Total | Found in Stammnitz et al, 2023* | Loss/gain | Significant change | Total | Loss/gain | Significant change |
| **DFT1** | 922,360 | 0.93 | 10,373 | 169,717 | 482,774 | 20,860 | 67,655 |
| DFT1 P | 916,924 | 0.71 | | | 480,420 | | |
| DFT1 R1 | 919,466 | 0.71 | 9102 | 104,421 | 481,680 | 13,949 | 42,546 |
| DFT1 R2 | 919,738 | 0.75 | 8763 | 104,730 | 481,917 | 13,377 | 41,590 |
| **DFT2** | 1,017,229 | 0.93 | 20,701 | 273,298 | 543,914 | 28,885 | 122,281 |
| DFT2 P | 1,007,301 | 0.75 | | | 538,382 | | |
| DFT2 R1 | 1,008,305 | 0.75 | 18,772 | 192,310 | 541,116 | 20,073 | 87,367 |
| DFT2 R2 | 1,008,551 | 0.75 | 18,604 | 193,444 | 541,039 | 19,772 | 86,838 |

Total numbers of SNVs and InDels after filtering and number of variants in the resistant cell lines showing either significant gains and losses (FDR-corrected binomial test, $P < 0.1$) or frequency changes against the parental lines (FDR corrected Fisher's exact test, $P < 0.05$).
*Fraction of SNVs identified in either the DFT1 or DFT2 samples from Stammnitz et al, 2023, that were also detected in the corresponding DFT1 or DFT2 samples in this study. Only SNVs from the Stammnitz et al study with an allele frequency of at least 25% and present in more than 50% of DFT1 or DFT2 samples were included, resulting in 522.012 SNVs for DFT1 and 609.974 SNVs for DFT2.

how each cancer adapts to selective pressures within its environment.

The distinct phosphorylation profiles of DFT1 and DFT2 correspond to divergent drug vulnerabilities and highlight the need for tailored therapeutic strategies. To target DFT1 and ERBB signalling, sapitinib was selected for further investigation, based on prior efficacy and specificity (Kosack et al, 2019). Imatinib was used as a representative DFT2 specific inhibitor that targets PDGFRs, due to clinical relevance and specificity in our drug screen validations. However, this poses a broader pharmacological challenge: the lack of strict specificity among many kinase inhibitors, including imatinib (Anastassiadis et al, 2011; Klaeger et al, 2017). Such promiscuity may explain overlapping hits in our screen including afatinib, a broad-spectrum ERBB inhibitor that showed activity in both DFT1 and DFT2, unlike more selective ERBB compounds. Similarly, a few PDGFR-targeting inhibitors also showed effects in DFT1, despite PDGFR dependency being more pronounced in DFT2. This complicates mechanistic interpretation and shows the need for orthogonal functional approaches. While PDGFRA is a strong candidate driver, given its phosphorylation, expression, and genomic amplification (Stammnitz et al, 2018), off-target effects of imatinib cannot be excluded as the cause for inhibition success in DFT2 cells. Knockdown or knockout experiments will be crucial to confirm target dependencies. Nevertheless, the tumour-specific drug responses observed here reveal distinct signalling and are our rationale for differential treatment strategies.

Beyond target specificity, other barriers may limit clinical application of TKIs. Systemic toxicity is a known limitation for many small molecule inhibitors, especially broad-spectrum TKI (Fabbro et al, 2015; Zhong et al, 2021). Species-specific toxicity also remains a concern in Tasmanian devils, although our protein modelling suggests strong conservation of kinase binding sites, supporting potential cross-species efficacy. A further challenge is tumour plasticity. Schwann cells, the presumed origin of both DFT1 and DFT2, naturally transition between myelinating and repair phenotypes (Boerboom et al, 2017; Jessen and Mirsky, 2016), a process exploited by cancers to evade targeted therapies via epithelial-to-mesenchymal-like transitions (Clements et al, 2017;

Jain et al, 2023). The phenotypic shifts observed in dedifferentiated DFT1 cells after vaccination highlight this inherent adaptability (Patchett et al, 2021). To explore this, we generated resistant cell line models of both tumours.

Both employed DFT1 and DFT2 cell lines acquired resistance to TK inhibition. In DFT1, resistance to sapitinib emerged rapidly without major copy number alterations or phosphoproteomic shifts, suggesting a reliance on pre-existing signalling plasticity rather than large-scale genomic reprogramming. One plausible mechanism involves reliance on alternative kinases such as PDGFRB, which is expressed in DFT1 and may bypass ERBB inhibition. Supporting this, previous work has identified PDGFRB amplifications on extrachromosomal double minutes and enhanced responses to PDGF ligands in DFT1 (Stammnitz et al, 2018; Maskell et al, 2025). We consider it possible that SNVs in tyrosine kinase encoding or related genes also contribute to resistance. Although we did not systematically analyse target mutations here, our whole-genome sequencing data provide a foundation for future studies to address this question.

By contrast, DFT2 resistance to imatinib developed more slowly and was accompanied by widespread changes on all investigated levels. DFT2 is also known to exhibit higher mutation rates in nature compared to DFT1 (Stammnitz et al, 2018), potentially indicating an intrinsically greater need for adaptive genomic remodelling. Our data suggest that the resistant DFT2 lines underwent whole-genome duplication, maybe as a stress-induced response to drug pressure or via selection of pre-existing tetraploid subpopulations. This interpretation remains to be confirmed through orthogonal methods, such as in situ hybridization or flow cytometry-based ploidy assessment. Interestingly, tetraploid DFT2 tumours have also been reported in the wild (Stammnitz et al, 2023), hinting at a broader relevance of genome doubling in DFT2 evolution.

Intriguingly, several features of imatinib-resistant DFT2 cells resemble those of DFT1, including increased ERBB pathway activity and increased JAK/STAT signalling. Downregulation of MHC-related genes further mirrors immune evasion strategies well-documented in DFT1 (Siddle et al, 2013; Kosack et al, 2019). Given that DFT2 tumours in nature show decreasing MHCI

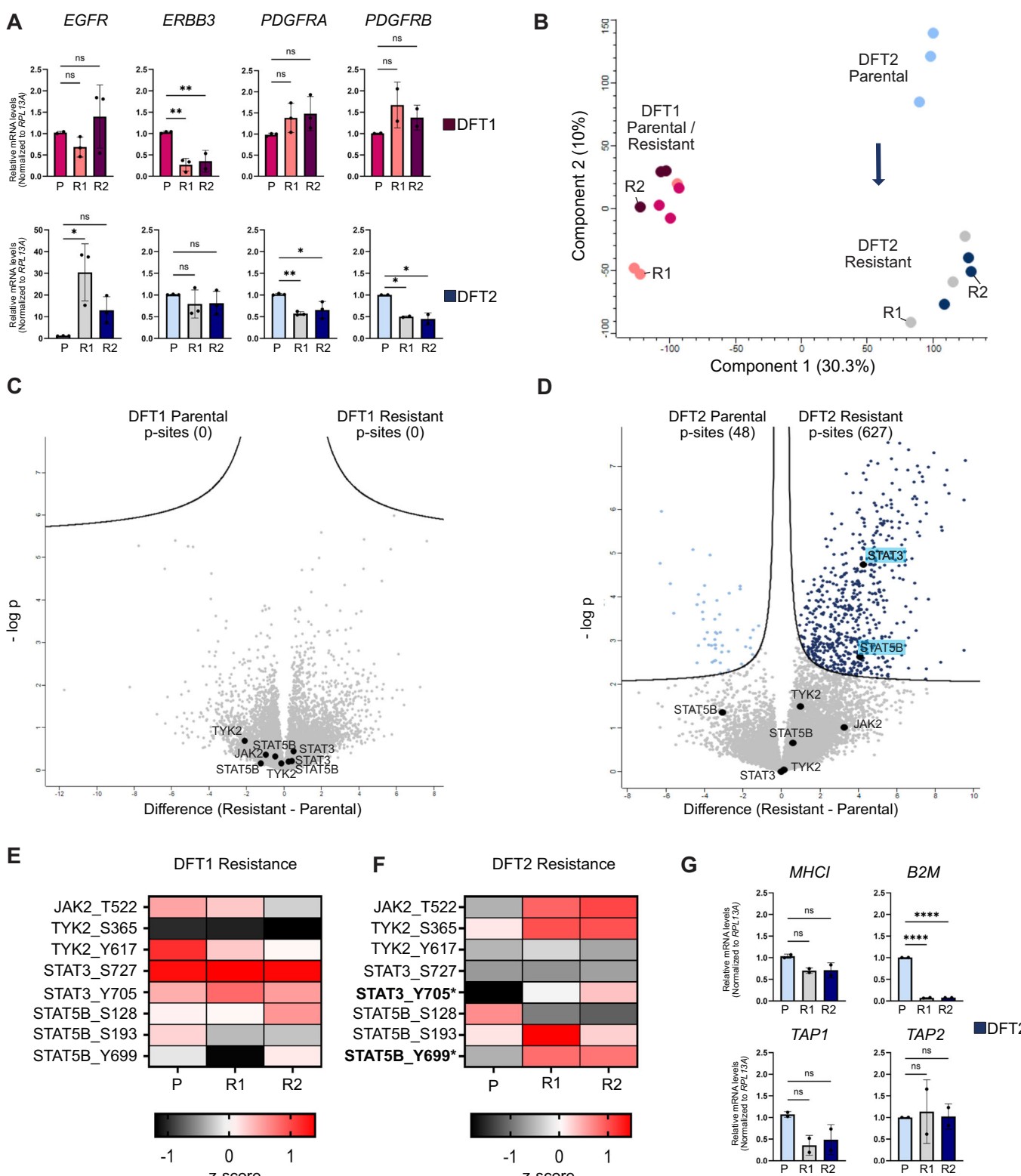

expression (Caldwell et al, 2018), these findings may indicate convergence towards a shared phenotype that supports both oncogenic signalling and immune escape. Although DFT2 remains less prevalent in the wild, its apparent plasticity under selection could enhance its long-term transmissibility (Patchett et al, 2020).

Remarkably, both independently generated resistant DFT2 lines exhibit similar CNV patterns and recurrent mutations,

**Figure 5. Imatinib resistance drives oncogenic pathway switching in DFT2 cells.**

(A) Gene expression of *EGFR*, *ERBB3*, *PDGFRA*, and *PDGFRB* in parental (P) and resistant (R1, R2) DFT1 (top) and DFT2 (bottom) cell lines, quantified by real-time PCR and normalised to *RPL13A*. Data are shown relative to the respective parental control and presented as mean ± SD ($n = 3$; $n = 2$ for *PDGFRB*). Statistical analysis was performed using one-way ANOVA with Bonferroni's multiple comparisons test, comparing each group to the parental (P) group. Statistical significance was set at $P < 0.05$ and is indicated as follows: $P < 0.0332$ (*), $P < 0.0021$ (**), $P < 0.0002$ (***), and $P < 0.0001$ (****). Exact P values: ERBB3 (DFT1): P vs. R1, $P = 0.003$; P vs. R2, $P = 0.0082$. EGFR (DFT2): P vs. R1, $P = 0.0105$. PDGFRA (DFT2): P vs. R1, $P = 0.007$; P vs. R2, $P = 0.0182$. PDGFRB (DFT2): P vs. R1, $P = 0.0172$; P vs. R2, $P = 0.0131$. (B) Principal component analysis (PCA) of phosphoproteomics data showing separation between parental and resistant DFT1 (R1, R2) and parental and resistant DFT2 (R1, R2) cells. The arrow highlights the marked gene expression changes between parental and resistant DFT2 cells, in contrast to minimal changes observed between parental and resistant DFT1 cells. (C, D) Volcano plots comparing parental with (C) sapitinib-resistant DFT1 and (D) imatinib-resistant DFT2 phosphoproteomes (R1 and R2). P-peptides of the JAK/STAT pathway are highlighted in black. Two-sample *t* test with permutation-based FDR correction (FDR < 0.05, $s_0 = 0.1$) was used to identify significantly altered phosphopeptides. Data represent $n = 3$ independent replicates for parental cells and $n = 3$ independent replicates each for resistant lines R1 and R2 (combined for analysis, total $n = 6$). (E, F) Heatmaps of the relative abundance (z-scored log2 intensity values) of JAK/STAT p-peptides in DFT1 parental and sapitinib-resistant (E) and DFT2 parental and imatinib-resistant (F) cell lines. *Significant differences in phosphorylation between parental and resistant cells (P value < 0.05). (G) Gene expression of *MHCI*, *B2M*, *TAP1* and *TAP2* in parental (P) and resistant (R1, R2) DFT2 cell lines, quantified by real-time PCR and normalised to *RPL13A*. Expression levels are shown relative to the respective parental control and presented as mean ± SD. Data represent two independent experiments, each performed with three technical replicates per condition; results from both experiments were pooled for analysis. Statistical analysis was performed using one-way ANOVA with Bonferroni's multiple comparisons test, comparing each group to the parental (P) group. Statistical significance was set at $P < 0.05$ and is indicated as follows: $P < 0.0332$ (*), $P < 0.0021$ (**), $P < 0.0002$ (***), and $P < 0.0001$ (****). *B2M* expression: P vs. R1/R2, $P < 0.0001$. Source data are available online for this figure.

suggesting convergent evolution under drug pressure. These observations raise the possibility that tumour plasticity in these transmissible Schwann cell–derived cancers may be at least partially genetically preconditioned. A parallel can be drawn to prostate cancers, where concurrent loss of the tumour suppressors TP53 and RB1 has been associated with increased lineage plasticity and resistance to therapy (Nyquist et al, 2020). In our study, DFT1 may harbour pre-existing mutations or genomic configurations that facilitate rapid signalling pathway switching without requiring extensive genomic rearrangement. In contrast, resistance in DFT2 appears to involve larger-scale genomic alterations, which may serve as a substrate for adaptive flexibility. These differences suggest that while both tumour types can evolve under pharmacological pressure, the underlying mechanisms and timeframes are shaped by their intrinsic genetic architecture and evolutionary histories.

The adaptive strategies of DFT1 and DFT2 suggest that combination therapies targeting both ERBB and PDGFR pathways may suppress resistance and reduce toxicity by allowing lower individual drug doses. Dual targeting has shown promise in other cancers such as renal cell carcinoma (Schöffski et al, 2006; Subbiah et al, 2020). In our study, ERBB–PDGFR inhibitor combinations exhibited synergy in DFT1, the more widespread and ecologically concerning tumour type (Cunningham et al, 2021; Lazenby et al, 2018). These findings reinforce the importance of integrated targeting strategies in future translational efforts.

Conservation efforts of Tasmanian devils remain an urgent priority (Cunningham et al, 2021; Stammnitz et al, 2023). Our study offers crucial insights into the fundamental molecular wiring of transmissible cancers by exploiting the vulnerabilities and escape mechanisms of DFT using pharmacological targeting of tyrosine kinase inhibitors. This may prove instrumental for novel combined therapeutic strategies, possibly in conjunction with immunotherapy and/or vaccination.

This study relies on in vitro models, which cannot fully recapitulate transmissible tumours in nature. Phosphoproteomic profiling, resistant cell line generation, and whole-genome sequencing were performed using a single representative cell line per tumour type, which may limit the capture of tumour heterogeneity.

Moreover, resistance mechanisms identified in vitro may not fully reflect in vivo dynamics, including immune and stromal interactions. Nevertheless, the remarkable copy-number stability of transmissible cancers and the strong concordance of our genomic and transcriptomic data with previous studies (Stammnitz et al, 2023, 2018) support their value as mechanistic models. Finally, the extent of genomic remodelling observed raises the possibility that structural variants (SVs) and fusion genes, known contributors to DFT evolution, may also underlie resistance. Although not explored here, future analysis of SVs will be key to fully uncovering resistance mechanisms in transmissible cancers.

## Methods

**Reagents and tools table**

| Reagent/resource | Reference or source | Identifier or catalog number |
|---|---|---|
| **Experimental models** | | |
| Cell line DFT1 strain 1 | Deakin et al, PLoS Genet 2012 | 06/2887 |
| Cell line DFT1 1426 | Siddle et al, 2013 | 86 T |
| Cell line DFT1 C5065 | Siddle et al, 2013 | 87 T |
| Cell line DFT2 Red Velvet (RV) | Pye et al, 2016 | 202T2 |
| Cell line DFT2 Snug (SN) | Pye et al, 2016 | 203T3 |
| Cell line Jarvis (JV) | Pye et al, 2016 | 338 T |
| Cell line fibroblasts | Murchison et al, Cell 2012 | 91H |
| DFT1 biopsy | NRE, Tasmania, AUS | T588 |
| DFT1 biopsy | NRE, Tasmania, AUS | T328 |

| Reagent/resource | Reference or source | Identifier or catalog number |
| --- | --- | --- |
| DFT1 biopsy | NRE, Tasmania, AUS | T603 |
| DFT2 biopsy | NRE, Tasmania, AUS | T607 |
| DFT2 biopsy | NRE, Tasmania, AUS | T594 |
| DFT2 biopsy | NRE, Tasmania, AUS | T609 |
| Peripheral nerve biopsy | NRE, Tasmania, AUS | Nerve 1 |
| Peripheral nerve biopsy | NRE, Tasmania, AUS | TD269 |
| NOD scid gamma (NSG) mice | University of Veterinary Medicine Vienna | – |
| **Antibodies** | | |
| Anti-PDGFR alpha | Cell Signaling Technology | 5241 |
| Anti-PDGFR alpha | Abcam | ab124392 |
| Anti-PDGFR alpha | Cell Signaling Technology | 3174 |
| Anti-PDGFR beta | Cell Signaling Technology | 8044 |
| Anti-PDGFR beta | Cell Signaling Technology | 4564 |
| Anti-HER2/ErbB2 | Cell Signaling Technology | 4290 |
| Anti-ERBB3 | Cell Signaling Technology | 12708 |
| Anti-ERBB3 | Cell Signaling Technology | 4791 |
| Anti-STAT3 | BD Biosciences | 610189 |
| Anti-STAT3 | Cell Signaling Technology | 9139 |
| Anti-phospho-STAT3 (Tyr705) | Cell Signaling Technology | 9131 |
| Anti-phospho-STAT3 (Ser727) | Cell Signaling Technology | 9134 |
| Anti-STAT5 | BD Biosciences | 610191 |
| Anti-periaxin | Sigma-Aldrich | HPA001868 |
| Anti-SOX10 | Abcam | ab180862 |
| Anti-Ki-67 | Cell Signaling Technology | 12202 |
| Anti-vWF | Invitrogen Antibodies | PA5-16634 |
| Anti-cleaved caspase-3 (Asp175) | Cell Signaling Technology | 9661 |

| Reagent/resource | Reference or source | Identifier or catalog number |
| --- | --- | --- |
| Anti-HSC70 | Santa Cruz Biotechnology | sc-7298 |
| **Oligonucleotides and other sequence-based reagents** | | |
| CTTCCTAAAAACCATCCAGGAGG | Kosack et al, 2019 | EGFR_forward |
| TTGGAGAGCACAGAAAGTGCAT | | EGFR_reverse |
| TCTGGGGAACCCAAGTGTG | This study | ERBB2_forward |
| GTACAGGTGGCTCAGTGTGT | | ERBB2_reverse |
| TCCAGGGAGGATGTCCAGG | Kosack et al, 2019 | ERBB3_forward |
| TTCCAGGTTTCCCATCACCA | | ERBB3_reverse |
| CGGAGTCACGTAGAGATAAGTTTTCC | Kosack et al, 2019 | PDGFRA_forward |
| TGGAAGCTGGCAAAATGTGA | | PDGFRA_reverse |
| TGTGCCAGATTCCTCTGTGG | Kosack et al, 2019 | PDGFRB_forward |
| TCTCATGCAACGTTACCACCA | | PDGFRB_reverse |
| GGAAAAGCAAGACTGGGACTATGC | Kosack et al, 2019 | STAT1_forward |
| GCGGCTATAGTGCTCATCCAA | | STAT1_reverse |
| AGGACTGGGCATATGCTGCC | Kosack et al, 2019 | STAT3_forward |
| TGCTTGATTCTTCGCAGGTTGT | | STAT3_reverse |
| AGTTATGTGTGAATCTGCCAC | This study | STAT5B_forward |
| GACAAACTCTGGGACCACCT | | STAT5B_reverse |
| CCGTGGGCTACGTGGACGATCAGC | Siddle et al, 2013 | MHCI_forward |
| GTCGTAGGCGAACTGAAG | | MHCI_reverse |
| TGTGCATCCTTCCCTACCTGGAGG | Siddle et al, 2013 | B2M_forward |
| CATTGTTGAAAGACAGATCGGACCGC | | B2M_reverse |
| ACAGACTGGATCCTGCAGGATGAAG | Siddle et al, 2013 | TAP1_forward |
| GAGACGTGATAGCACCTGTTTGG | | TAP1_reverse |
| TGTGGGCTAAGGCAGATTCTGG | Siddle et al, 2013 | TAP2_forward |
| ATTCCCAGGAGGAGCTAAGCG | | TAP2_reverse |
| CCCCACAAGACCAAGCGAGGC | Siddle et al, 2013 | RPL13A_forward |
| ACAGCCTGGTATTTCCAGCCAACC | | RPL13A_reverse |
| **Chemicals, enzymes and other reagents** | | |
| Sapitinib | MedChem Express | HY-13050 |
| Imatinib | MedChem Express | HY-15463 |
| **Software and tools** | | |
| BBDuk (BBTools suite) | Bushnell et al, 2017 | v38.90 |
| BWA-MEM2 | Li and Durbin, 2009; Vasimuddin et al, 2019 | v2.2.1 |
| SAMtools | Danecek et al, 2021 | v1.19 |
| GATK | McKenna et al, 2010 | v4.6.2.0 |
| GATK Mutect2 | Benjamin et al, 2019 | v4.6.2.0 |
| BCFtools | Danecek et al, 2021 | v1.21 |

| Reagent/resource | Reference or source | Identifier or catalog number |
|---|---|---|
| Ensembl Variant Effect Predictor | McLaren et al, 2016 | v113 |
| pysam | Virtanen et al, 2020 | v0.22.1 |
| SciPy | Virtanen et al, 2020 | v1.15.2 |
| R | R Core Team, 2024 | v4.4.0 |
| Control-FREEC | Zhao et al, 2013 | |
| nQuack | Gaynor et al, 2024 | v. 1.0.2 |
| Spectronaut | Biognosys Inc | v18 |
| Perseus | Tyanova and Cox, 2018 | v2.0.11 |
| SynergyFinder | Ianevski et al, 2022 | V3.0 |
| GraphPad Prism | GraphPad Software | v8.4.3 |
| **Other** | | |
| Illumina NovaSeq X | Illumina | |
| Monarch Genomic DNA Purification Kit | New England Biolabs | T3010S |
| NEBNext Multiplex Library Prep Kit | New England Biolabs | E7395 |
| Ultimate 3000 RSLC and Q-Exactive HF Orbitrap | Thermo Fisher Scientific | |

## Cell culture and tissue biopsies

Devil Facial Tumour (DFT) cell lines were established from primary cell cultures derived from fine needle aspirates obtained from wild Tasmanian devils (Appendix Table S1). The following DFT cell lines were cultured: DFT1 S1, DFT1 1426, DFT1 C5065, DFT2 SN, DFT2 RV, and DFT2 JV. These were maintained in RPMI 1640 medium (Gibco™). Fibroblasts were cultured in Advanced DMEM (Gibco™). Both media were supplemented with 10% fetal bovine serum (FBS, Biowest), 1% penicillin-streptomycin (Biowest), 1% L-Glutamine (Gibco™). DFT2 cell lines additionally received 50 mM 2-mercaptoethanol (Gibco™) and were cultured in 10% AminoMax™-II Complete Medium (Gibco™) for one passage after thawing. All cells were incubated at 37 °C in a humidified atmosphere with 5% $CO_2$. To subculture, DFT1 cells were detached using phosphate-buffered saline (PBS, Biowest), DFT2 cells were harvested with PBS containing 1 mM EDTA (Biochemica, AppliChem), and fibroblasts were detached using 0.05% Trypsin-EDTA (Biowest). All inhibitors were dissolved in dimethyl sulfoxide (DMSO, Sigma-Aldrich). Primary biopsies from Tasmanian devils were obtained through the Department of Natural Resources and Environment (NRE), Tasmanian Government, Australia. Biopsies were either stored frozen on dry ice or embedded in paraffin blocks for later processing (Appendix Table S2). All animal procedures were conducted in accordance with a Standard Operating Procedure of the NRE Tasmania Save the Tasmanian Devil Program, and in compliance with guidelines established by the University of Tasmania Animal Ethics Committee.

## Generation of drug-resistant cell lines

To generate drug-resistant cell lines, DFT1 C5065 cells were exposed to increasing concentrations of sapitinib (HY-13050, MedChem Express) and DFT2 SN cells to increasing concentrations of imatinib (HY-15463, MedChem Express) over a period of 6 months (Fig. 4B). Two independent resistant cell lines (R1/R2) were generated from each parental cell line. Cells were seeded in 25 cm² cell culture flasks (Sarstedt or Greiner Bio-One) and allowed to adhere overnight. The cells were then treated with the compounds and the medium was replaced weekly to maintain continuous drug exposure. Following initial drug treatment, cells were allowed to regain confluency before being exposed to escalating drug concentrations in a stepwise manner. The concentration was incrementally increased over time to the final resistance concentrations of 10 µM sapitinib (DFT1) and 1 µM imatinib (DFT2). Cells were then continuously cultured at the final drug concentration, never longer than 3 weeks. To validate drug resistance, the cell viability of resistant clones was assessed in the presence of sapitinib and imatinib using the CellTiter-Blue® reagent (Promega), and the half-maximal inhibitory concentration ($IC_{50}$) was determined and compared to parental cell lines. Prior to experimental use, cells were expanded without drug exposure for at least two passages to ensure the removal of any drug carryover effects.

## Whole-genome sequencing (WGS)

### *Sequencing and library preparation*
Genomic DNA was isolated from the following cell lines: DFT1 parental (P), DFT1 resistant 1 (R1), DFT1 resistant 2 (R2), DFT2 parental (P), DFT2 resistant 1 (R1), and DFT2 resistant 2 (R2) using the Monarch Genomic DNA Purification Kit (New England Biolabs, cat. no. T3010S). Sequencing libraries were prepared using the NEBNext Multiplex Library Prep Kit with Enzymatic Shearing and Unique Dual Index UMI Adaptors (New England Biolabs, cat. no. E7395). Libraries were sequenced as paired-end 150 bp reads on an Illumina NovaSeq X platform using a 10B flow cell, with three libraries sequenced per run. Due to suboptimal fragment sizes (~110 bp) observed in the first sequencing run, all libraries underwent additional size selection and were resequenced. Reads from both runs were merged for downstream variant calling and copy number variation analysis. All steps of library preparation, sequencing, and demultiplexing were performed by the Next Generation Sequencing Facility at Vienna BioCenter Core Facilities (VBCF), Austria.

### *Read preprocessing and alignment*
Demultiplexed reads were quality and adapter trimmed using BBDuk from the BBTools suite (v38.90) (Bushnell et al, 2017) using the parameters: ktrim=r k = 20 mink=11 hdist=1 tbo qtrim=r trimq=10 minlen=20. Trimmed reads were aligned to the Tasmanian devil reference genome (mSarHar1.11; GCA_902635505.1) (Stammnitz et al, 2023) using BWA-MEM2 (v2.2.1) (Li and Durbin, 2009; Vasimuddin et al, 2019). Alignments

**Table 3. (Related to Fig. 5): Differentially phosphorylated peptides linked to kinases in the resistant DFT2 signature.**

| Gene | Protein description | Protein name | PTM* multiplicity | PTM* site (AA) | PTM* site location | PTM* Flanking region |
|---|---|---|---|---|---|---|
| MAST2 | Non-specific serine/threonine protein kinase | A0A7N4NRI8_SARHA | 1 | S | 101 | RNQSLGQSAPSLTAG |
| | | | 1 | S | 225 | LDSPRNFSPNAPAPH |
| | | | 1 | S | 944 | LGVRRRCSGLLDAPR |
| | | | 1 | S | 913 | DRGWVIGSPEILRKR |
| | | | 1 | S | 1037 | RARHRLLSGEAGDKR |
| SLK | Non-specific serine/threonine protein kinase | G3WYI9_SARHA | 1 | S | 340 | IPANKRASSDLSIAS |
| PRKG1 | cGMP-dependent protein kinase | A0A7N4PZK6_SARHA | 1 | Y | 555 | TFCGTPEYVAPEIIL |
| PDGFRA | Platelet-derived growth factor receptor alpha | G3WJ82_SARHA | 1 | S | 615 | VLGRILGSGAFGKVV |
| | | | 1 | Y | 627 | KVVEGTAYGLSRSQP |
| | | | 1 | Y | 1002 | RVECDNTYIGVTYKN |
| | | | 1 | Y | 863 | DIMHDSNYVSKGSTF |
| | | | 1 | Y | 940 | DHATSEVYEIMVKCW |
| | | | 1 | Y | 756 | KQADTTQYVPMLERK |
| | | | 1 | Y | 776 | SDIQRSMYDRPASYK |
| | | | 1 | Y | 782 | MYDRPASYKKKSVSD |
| | | | 1 | S | 726 | LDIFGMNSADESTRS |
| MINK1 | Misshapen like kinase 1 | A0A7N4NK65_SARHA | 1 | S | 914 | QYEVRKGSVVNVNPT |
| MELK | Maternal embryonic leucine zipper kinase | G3W9C9_SARHA | 1 | T | 370 | DVESKPLTPILCRAA |
| | | | 1 | S | 478 | LNQAHRDSPQKRKGT |
| LATS2 | Large tumour suppressor kinase 2 | G3VLS0_SARHA | 1 | S | 405 | STLSRRDSLQNPSLE |
| SIK3 | Non-specific serine/threonine protein kinase | G3WH08_SARHA | 1 | S | 625 | FSPVRRFSDGAASIQ |
| | | | 1 | S | 550 | GPLGRRASDGGANIQ |
| TNIK | TRAF2 and NCK interacting kinase | G3WC40_SARHA | 1 | S | 711 | GSQPIRASNPDLRRT |
| | | | 1 | S | 1056 | LNEARKISVVNVNPT |
| SIK2 | Non-specific serine/threonine protein kinase | A0A7N4NUA8_SARHA | 1 | S | 674 | PQLSRRQSLETQYLQ |
| SRC | Tyrosine-protein kinase | A0A7N4PRI3_SARHA | 1 | Y | 433 | RLIEDNEYTARQGAK |
| STK39 | Non-specific serine/threonine protein kinase | G3WL88_SARHA | 1 | S | 365 | VRRVPGSSGHLHKTE |
| TTK | TTK protein kinase | A0A7N4NGQ9_SARHA | 1 | S | 327 | VPVNLINSPGGPVEK |
| CAMKK2 | Calcium/calmodulin dependent protein kinase kinase 2 | A0A7N4P1C0_SARHA | 1 | S | 107 | ERSQAAPSPGGSGGV |
| PAK1 | Non-specific serine/threonine protein kinase | G3VVJ3_SARHA | 1 | T | 211 | VIDPIPVTPTRDVAT |
| RPS6KC1 | Non-specific serine/threonine protein kinase | A0A7N4PG14_SARHA | 1 | S | 620 | SLSRSKNSPMAFFRI |
| MAP3K20 | Mitogen-activated protein kinase kinase kinase 20 | A0A7N4PGP5_SARHA | 1 | S | 629 | QITPINHSRSSSPTQ |
| ABL1 | Tyrosine-protein kinase | A0A7N4PXL7_SARHA | 1 | S | 737 | KRFLRSCSASCVPHG |
| | | | 1 | S | 16 | LGDQRRPSLPALHFI |
| | | | 1 | T | 832 | GSSPPSLTPKLVRKQ |
| ABL2 | Tyrosine-protein kinase | A0A7N4NRP8_SARHA | 1 | S | 584 | SAQPPSGSPALPRKQ |
| RIPK1 | Receptor interacting serine/threonine kinase 1 | G3WNI7_SARHA | 1 | S | 345 | EVVKRMQSLQIDSVP |
| | | | 1 | S | 443 | EERRRRVSHDPFTVA |

**Table 3.** (continued)

| Gene | Protein description | Protein name | PTM* multiplicity | PTM* site (AA) | PTM* site location | PTM* Flanking region |
|---|---|---|---|---|---|---|
| MARK3 | Non-specific serine/threonine protein kinase | A0A7N4V4B0_SARHA | 1 | S | 469 | GKGLAPASPMLGNAN |
| | | | 1 | S | 419 | SQKQRRYSDHAGPSI |
| MARK2 | Microtubule affinity regulating kinase 2 | A0A7N4PNF7_SARHA | 1 | S | 440 | STAKVPASPLPGLER |
| | | | 1 | S | 394 | NPKQRRFSDQAGPAI |
| MARK1 | Non-specific serine/threonine protein kinase | A0A7N4V6S6_SARHA | 1 | S | 460 | NQKQRRFSDHVGPSI |
| MAP4K4 | Mitogen-activated protein kinase kinase kinase kinase 4 | G3VHS5_SARHA | 1 | S | 1046 | QDPTRKGSVVNVNPT |
| LYN | Tyrosine-protein kinase | G3WDZ1_SARHA | 1 | Y | 399 | RIIEDNEYTAREGAK |
| CDK17 | Cyclin dependent kinase 17 | G3WMX5_SARHA | 1 | S | 146 | EDLNKRLSLPADIRI |
| | | | 1 | S | 137 | NRIHRRISMEDLNKR |
| | | | 1 | S | 9 | KKFKRRLSLTLRGSQ |
| | | | 1 | S | 180 | SRRSRRASLSEIGFG |
| | | | 1 | S | 165 | LEKLQINSPPFDQPM |
| CDK18 | Cyclin dependent kinase 18 | G3WR55_SARHA | 1 | S | 102 | EDVSKRLSLPMDIRL |
| MAP3K1 | Mitogen-activated protein kinase kinase kinase 1 | G3WBL4_SARHA | 1 | S | 1033 | IKEPDKLSPVFTQAR |
| MAPK14 | Mitogen-activated protein kinase 14 | A0A7N4NGT3_SARHA | 1 | Y | 182 | TDDEMTGYVATRWYR |
| DCLK1 | Doublecortin like kinase 1 | G3WAA2_SARHA | 1 | S | 32 | SRVNGLPSPTHSAHC |
| TAOK3 | TAO kinase 3 | A0A7N4NSY2_SARHA | 1 | S | 373 | FATIKSASLVTRQIH |
| WNK1 | Non-specific serine/threonine protein kinase | A0A7N4PI27_SARHA | 1 | S | 2854 | SNLQKSISNPPGSNL |
| CDK14 | Cyclin dependent kinase 14 | A0A7N4P5N5_SARHA | 1 | S | 24 | KKLRRTLSESFSRIA |
| CDK16 | Protein kinase domain-containing protein | A0A7N4P4W0_SARHA | 1 | S | 254 | SRRLRRVSLSEIGFG |

Phosphorylation sites with significant differential abundance between DFT2 parental and resistant cell lines (R1 and R2) are shown, including site location and flanking sequences. Peptides were filtered from all differentially phosphorylated peptides (FDR < 0.05, $s_0$ = 0.1; see Fig. 5D and Source Data) and annotated using the InterPro term IPR011009 (Protein kinase-like domain) via DAVID Bioinformatics.
*All post-translational modifications (PTMs) listed in this table are phosphorylation events

were sorted and PCR duplicates marked using SAMtools (v1.19) (Danecek et al, 2021).

### Reference genome and annotation

All analyses were performed using the mSarHar1.11 genome with gene annotations from Ensembl release 114.111 (Dyer et al, 2025).

As *ERBB3* was not annotated in the reference genome used in this study (Stammnitz et al, 2023), we retrieved its cDNA (ENSSHAT00000006813.1) from an older annotation and used BLAST to locate its likely position in the updated genome. The cDNA aligned to chr5:84,558,652–84,597,822 (reverse strand), overlapping annotated genes ENSSHAG00000026735, ENSSHAG00000031432, and ENSSHAG00000020608, suggesting this is the corresponding locus.

### Base quality score recalibration (BQSR)

BQSR was performed using GATK (v4.6.2.0) (McKenna et al, 2010) with known variant sites from Stammnitz et al, 2023. Due to chromosome size limitations, BQSR was trained on chromosomes 4, 5, and 6, and the resulting model was applied across all chromosomes. BAM files from both sequencing runs were merged post-BQSR.

### Somatic variant calling

Short variants (SNVs and Indels) were called using GATK Mutect2 (v4.6.2.0) (Benjamin et al, 2019), followed by filtering with GATK FilterMutectCalls, excluding variants failing the strand_bias, strict_strand, position, duplicate, or map_qual filters.

VCF file manipulations were performed using BCFtools (v1.21) (Danecek et al, 2021) and variants were annotated with the Ensembl Variant Effect Predictor (VEP, v113) (McLaren et al, 2016).

### Allele frequency analysis

A custom Python script utilizing pysam (v0.22.1) and SciPy (v1.15.2) (Virtanen et al, 2020) was used to calculate allele frequencies (AF) from the AD and DP fields in the VCF, and perform Fisher's exact tests comparing reference and alternative read counts between parental and resistant lines. Where alleles were lost, binomial tests were applied to estimate the probability of observing zero supporting reads under the expected frequency and depth.

All *P* values were corrected for multiple testing using false discovery rate (FDR) control via the p.adjust() function in R (v4.4.0) (R Core Team, 2024).

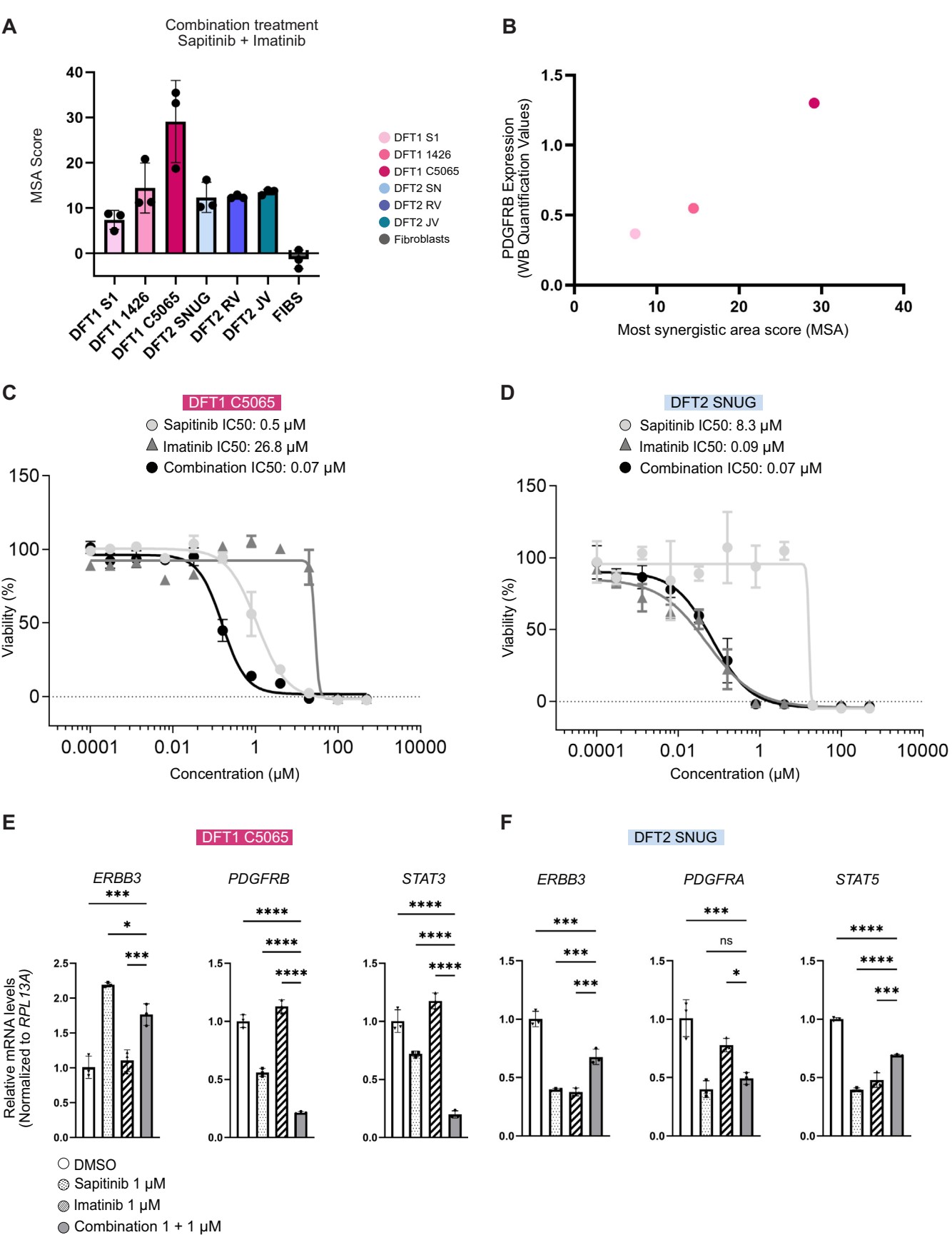

**Figure 6.  Sapitinib and imatinib combination treatment synergizes in DFT1.**

(A) Bar plot showing Most Synergistic Area (MSA) scores for sapitinib and imatinib combination treatment across three DFT1 cell lines, three DFT2 cell lines, and one healthy fibroblast control, tested in triplicate. Error bars represent mean ± SD. Synergy was assessed using SynergyFinder 3.0, with MSA > 10 indicating synergy. (B) Correlation between PDGFRB expression levels and MSA scores in DFT1. Quantification was performed from three independent Western blots (Source Data) using ImageJ. (C, D) Representative dose–response curves for sapitinib, imatinib, and their combination in DFT1 C5065 (C) and DFT2 SNUG (D). Data are presented as normalised percent of control (POC), shown as mean ± SD. $IC_{50}$ (µM) values represent means from three independent experiments. (E, F) Gene expression of *ERBB3*, *PDGFRB*, and *STAT3* (in DFT1 C5056, E) and ERBB3, PDGFRA and STAT5 in DFT2 SNUG (F) following treatment with DMSO, 1 µM sapitinib, 1 µM imatinib, or their combination (1 µM + 1 µM). Gene expression was quantified by real-time PCR, normalised to *RPL13A*, and shown relative to DMSO-treated controls. Data are presented as mean ± SD (one-way ANOVA, Dunnett's multiple comparisons test, one experiment representative of 2). Statistical significance was set at $P < 0.05$ and is indicated as follows: $P < 0.0332$ (*), $P < 0.0021$ (**), $P < 0.0002$ (***), and $P < 0.0001$ (****). Exact $P$ values were as follows: (E) ERBB3 – Combi vs. DMSO: $P = 0.0004$; Combi vs. Sapitinib (1 µM): $P = 0.0125$; Combi vs. Imatinib (1 µM): $P = 0.0009$. PDGFRB—all comparisons: $P < 0.0001$. STAT3—all comparisons: $P < 0.0001$. (F) ERBB3—Combi vs. DMSO: $P = 0.0001$; Combi vs. Sapitinib (1 µM): $P = 0.0003$; Combi vs. Imatinib (1 µM): $P = 0.0002$. PDGFRA—Combi vs. DMSO: $P = 0.0004$; Combi vs. Sapitinib (1 µM): ns; Combi vs. Imatinib (1 µM): $P = 0.0158$. STAT5—Combi vs. DMSO: $P < 0.0001$; Combi vs. Sapitinib (1 µM): $P < 0.0001$; Combi vs. Imatinib (1 µM): $P = 0.0001$. Source data are available online for this figure.

## Copy number variation (CNV) analysis

CNVs were analysed using Control-FREEC (Zhao et al, 2013) with a window size of 50 kb, using default GC-content correction and normalisation. BAF (B-allele frequency) profiles were calculated using SNVs identified as heterozygous (allele frequency 25–75%) in ≥50% of DFT1 or DFT2 samples sequenced in Stammnitz et al, 2023 (DFT1: 331,197 SNVs; DFT2: 389,350 SNVs). All samples were treated as female (XX) to avoid CNV artefacts from the Y chromosome. CNV segmentation and genotype status prediction were performed within Control-FREEC. Plots were generated in R using ggplot2 and patchwork. For testing ploidy, the R package nQuack (Gaynor et al, 2024) was used with 10,000 randomly chosen SNVs for each chromosome and a normal mixture model with uniform distributions and using the model type fixed_3 (variable mixture proportion and variance).

## Phosphoproteomics

For phosphoproteomic analysis, we used three independent replicates ($n = 3$), defined as separately cultured and harvested cell populations collected at different time points.

### Protein sample clean-up and digestion

Protein samples (200 µg) were first precipitated with 4 volumes of chilled ($-20$ °C) acetone for 2 h then resuspended in lysis buffer (5% (w/v) SDS in 100 mM ammonium bicarbonate) for on-column digestion using suspension trapping. Samples were sequentially reduced using 5 mM DTT for 15 min at 55 °C, alkylated using 20 mM iodoacetamide for 10 min in a light-proof container and acidified with phosphoric acid to a final concentration of 1% v/v. Samples were then diluted with 350 µl wash buffer (100 mM Tris-HCl, pH 7.5 in 90% (v/v) methanol) and transferred to S-trap™ micro columns (Protifi). Samples were washed five times using 400 µl wash buffer prior to addition of 10 µg trypsin/LysC (Promega) in 50 mM ammonium bicarbonate. Digests were incubated overnight at 37 °C in a Thermomixer and peptides were eluted according to manufacturer's (Protifi) guidelines, then dried in a SpeedVac. Peptides were resuspended in 300 µl of trifluoroacetic acid (0.1% v/v), desalted using Pierce™ peptide spin columns (Thermo Scientific), eluted in 400 µl of 50% acetonitrile containing 0.1% trifluoroacetic acid.

## Phosphopeptide enrichment and analysis by data-independent acquisition mass spectrometry (DIA-MS)

Phosphopeptides were enriched using Zirconium functionalized magnetic microparticles (MagReSyn® Zr-IMAC HP) according to manufacturer's guidelines (ReSyn Biosciences). Eluted peptides were dried in a SpeedVac, resuspended in 20 µl trifluoroacetic acid (0.1% v/v) and desalted using C18 ZipTips (Merck). Dried peptides were resuspended in 10 µl HPLC loading buffer (2% acetonitrile containing 0.05% trifluoroacetic acid) of which 3 µl was injected. Peptides were analysis by DIA-MS using an Ultimate 3000 RSLC and Q-Exactive HF Orbitrap mass spectrometer equipped with a nanospray Flex ion source (Thermo Fisher Scientific). Peptides were separated using a trap-elute configuration with a 20 mm × 75 µm C18 trapping column and 250 mm × 75 µm C18 analytical column over a 1-h segmented gradient using parameters for nanoLC and DIA-MS as previously published (Balotf et al, 2022).

### Data processing and quantification

Raw MS files were processed using Spectronaut software (v18; Biognosys) using a targeted, library-based approach. A spectral library was generated using the Pulsar search engine to search the *Sarcophilus harrisii* UniProt reference proteome (ID UP000007648) comprising 19,184 entries. Default search settings were used, with the exception that for phosphosite identification the PTM localization filter was activated (minimum threshold 0.75) and phospho (S/T/Y) was included as an additional variable modification. Relative quantitation of phosphopeptides between samples was then achieved by targeted re-extraction of DIA-MS2 spectra. Data were exported from Spectronaut as a PTM site report for the phosphopeptide dataset. Further data filtering and statistical analysis and visualization were performed in Perseus software v2.0.11 (Tyanova and Cox, 2018). The threshold values for statistical significance were FDR < 0.05 and s0 = 0.1, unless stated otherwise.

## Mouse xenograft studies

NOD scid gamma (NSG) mice were housed under pathogen-free conditions at the University of Veterinary Medicine, Vienna. Mice were maintained on a 12-h light/dark cycle with access to standard food and water *ad libitum*. All animal experiments were conducted

with the approval of the institutional ethics and animal welfare committee of the University of Veterinary Medicine, Vienna, and the national authority, in accordance with §§ 26ff of the Animal Experiments Act, TVG 2012 (BMWFW-68.205/0130-WF/V/3b/2016). The experimental design and the number of mice assigned to each treatment group were informed by prior experience with similar models, ensuring sufficient statistical power to detect significant differences. Mice were matched based on initial tumour size and randomized to receive either the inhibitor or the vehicle control (ddH2O) treatment. No animals were excluded from the analysis. For tumour implantation, $1 \times 10^6$ DFT1 S1 or DFT2 RV cells (Appendix Table S1) suspended in 100 µL PBS were injected subcutaneously into both flanks using a 27 G needle. Tumour growth was monitored every other day using Vernier callipers, and tumour volumes were calculated using the formula: tumour volume = (length × width$^2$)/2. Treatment began when tumours were palpable in 75% of mice and experiments were concluded when the tumour volume in the control group reached 1 cm³. At the end of the study, tumours were resected for further analysis, including tumour weight measurement, immunohistochemistry, and real-time PCR as described.

## Tyrosine kinase drug screen

A library of 48 kinase inhibitors used for drug screening was curated in-house. Selection criteria included: (i) TKIs that are FDA-approved or currently in clinical trials for the treatment of human cancers and (ii) compounds that had previously been tested in DFT1 cells, but had not yet been evaluated in DFT2 cells, or had shown inconclusive results (Dataset EV1). Inhibitors were transferred to 384-well plates using an Echo acoustic liquid handler (Backman Coulter; line 489). In all, 50 nL of each inhibitor in DMSO was added to each well. The following number of cells was seeded: 5000 DFT1 cells/well, 2000 DFT2 cells/well, and 2500 fibroblasts/well. Cells were added to each well to a final volume of 50 µL/well using a dispenser (Thermo Fisher Scientific), and the plates were incubated at 37 °C. Cell viability was assessed after 72 h using the CellTiter-Glo Luminescent Cell Viability Assay (Promega) in a multimode plate reader (Revvity, line 494). The screening was performed in five batches. In the first batch, 21 tyrosine kinase inhibitors were tested in a 6-dose response, with triplicate wells for DFT1 strain 1, DFT2 RV, DFT2 SN, and fibroblasts. In subsequent batches, inhibitors were tested in a 7-dose response across triplicate wells for DFT1 S1, DFT1 1426, DFT1 C5056, DFT2 SN, DFT2 RV, DFT2 JV, and fibroblasts. As a result of the multi-batch design, not all inhibitors were screened across every cell line in this dataset. Hit identification was based on the difference in Area Under the Curve (AUC) between each tumour cell line and control fibroblasts, with an AUC difference greater than 50. Additionally, drug candidates were further evaluated by comparing the AUC of each tumour strain to the mean AUC of the fibroblasts, incorporating a standard deviation calculation (Eq. 1). Hits were defined as meeting the response threshold in at least two DFT1 and/or DFT2 cell lines. For those screened in only a single DFT1 line (due to batch design), meeting the threshold in that one line was sufficient (Dataset EV1).

$$\text{AUC}_{\text{fibroblasts}} - \text{Sd}(\text{AUC}_{\text{fibroblasts}}) > \text{AUC}_{\text{DFT}} + \text{Sd}(\text{AUC}_{\text{DFT}}) \quad (1)$$

## Cell viability and drug combination assays

Cell viability assays were performed to determine the half-maximal inhibitory concentration (IC$_{50}$) and assess synergies of selected inhibitors across various cell lines. DFT1, DFT2, and fibroblast cells were seeded in 96-well flat-bottom plates (Sarstedt) at densities of 5000 cells/well, 2000 cells/well, and 2500 cells/well, respectively. Compounds were applied in triplicate, starting at a concentration of 100 µM and serially diluted in a 1:5 ratio. For drug combination experiments, compounds of interest were applied in a matrix-like setup, including single-compound dilution series. Bortezomib (10 µM, HY-10227, MedChem Express) was included as a positive control. After 72 h of incubation at 37 °C, cell viability was assessed using CellTiter-Blue® reagent (Promega), following the manufacturer's protocol. Fluorescence was measured using a GloMax® plate reader (Promega). IC$_{50}$ values were calculated via nonlinear regression analysis of dose–response curves using GraphPad Prism (version 8.4.3, GraphPad Software). Drug combination effects were evaluated using SynergyFinder 3.0 software (Ianevski et al, 2022). The synergism between compounds was quantified using the Zero Interaction Potency (ZIP) model, focusing on the most synergistic area (MSA) score.

## RT-qPCR

Total RNA was extracted from harvested cell pellets (~5–10 × 10⁶ cells) using TRIzol® reagent (Ambion) according to the manufacturer's protocol. RNA concentration and purity were assessed using a NanoDrop 2000 spectrophotometer (Thermo Scientific). Complementary DNA (cDNA) was synthesized from RNA using the RevertAid First Strand cDNA Synthesis Kit (Thermo Scientific™) with oligo(dT)$_{18}$ primers, following the manufacturer's instructions. Primers (Appendix Table S3) were designed using the NCBI Primer-BLAST tool, targeting exon-exon junctions to minimize genomic DNA amplification. Quantitative PCR reactions were performed in triplicate using 10 µM forward and reverse primers, GoTaq® qPCR Master Mix (Promega), and 12.5 ng/µL cDNA in a final reaction volume of 10 µL. Amplifications were conducted on a Bio-Rad CFX96 Touch Real-Time PCR System (Bio-Rad Laboratories) under the following conditions: initial denaturation at 95 °C for 3 min, followed by 40 cycles of 95 °C for 10 s, 60 °C for 30 s, and 72 °C for 30 s. Specificity of the amplicons was confirmed through melt curve analysis. Gene expression levels were calculated using the comparative Cq method ($2^{-\Delta\Delta Cq}$), with RPL13A as the reference gene. Results were presented as relative fold changes. Statistical analyses, including two-tailed $t$ tests or ANOVA where applicable, were performed using GraphPad Prism (version 8.4.3, GraphPad Software), with significance set at $P < 0.05$.

## Western blot analysis

Cell pellets (~5–10 × 10⁶ cells) were harvested and washed three times with cold PBS. Pellets were resuspended in whole cell extract (WCE) buffer, snap-frozen in liquid nitrogen, and stored at −80 °C until analysis. Protein concentrations were measured using the BCA protein assay (Bio-Rad Laboratories). Equal amounts of protein were resolved on 6% or 8% SDS-PAGE gels and transferred

onto 0.45 μm nitrocellulose membranes (Amersham Protran 10600002, GE Healthcare, UK). Membranes were blocked in 5% Bovine Serum Albumin (BSA, Roth) prepared in TBST for 1 h at room temperature and then incubated overnight at 4 °C with primary antibodies (Appendix Table S4). After washing, membranes were incubated with IRDye® secondary antibodies (1:10,000; IRDye® 800CW Goat Anti-Mouse, IRDye® 680RD Goat Anti-Rabbit, LI-COR Biosciences) for 2 h at room temperature. Protein bands were visualized using a LI-COR imaging system, and band intensities were quantified with ImageJ. Normalisation was performed against the loading control, HSC70, and results were expressed as relative fold changes compared to control conditions. Statistical significance was assessed using a two-tailed $t$ test or ANOVA, where appropriate, with $P < 0.05$ considered significant. All statistical analyses were conducted in GraphPad Prism (version 8.4.3, GraphPad Software).

## Histological and immunohistochemical analysis

DFT tissues from diseased animals and controls (T588, T328, T603, T607, T594, T609, Nerve 1) (Appendix Table S2) and mouse xenograft tumours were fixed in 10% neutral buffered formalin for 24 h at room temperature. After fixation, tissues were paraffin-embedded, and consecutive 2-μm sections were cut using a microtome. Sections were stained with Hematoxylin (Merck) and Eosin G (Carl Roth) for histological analysis. For immuno-histochemistry (IHC), heat-mediated antigen retrieval was performed to unmask epitopes. Depending on the antibody requirements, one of the following antigen retrieval buffers was used: citrate buffer (pH 6.0; S1699; Dako, Agilent, Santa Clara, CA, USA), EDTA buffer (pH 8.0), or TE buffer (pH 9.0). Sections were stained with antibodies listed in Appendix Table S4 using standard protocols. Tissue sections were scanned using an automated Evident VS200 slide scanner (Olympus). Quantifica-tion of immunohistochemical staining was performed using HistoQuest TM software (TissueGnostics). Statistical significance was determined using a two-tailed $t$ test or ANOVA, where appropriate, with $P < 0.05$ considered significant. Statistical analysis was performed using GraphPad Prism (version 8.4.3, GraphPad Software).

## Protein structure prediction, alignment, and docking

The 3-dimensional structure of *Sarcophilus harrisii* (sPDGFRA) was obtained from the Alphafold database (UniProt ID: G3WJ82) and that of ERBB3 (sERBB3) was predicted using Alphafold on Google Colab (Mirdita et al, 2022) and retrieved protein sequence (NCBI: NC_045430.1). Protein identifiers are prefixed with "s" to indicate *Sarcophilus harrisii* (Tasmanian devil) origin, and with "h" to indicate the corresponding human proteins. The protein structures were manually trimmed to match the length of that of their human counterpart kinase domains aligned using PyMol (Schrödinger, LLC 2015). Human ERBB3 (hERBB3) kinase domain (PDB ID: 6JOL) and the co-crystal structure and structure factor map of human PDGFRA (hPDGFRA) kinase domain bound to imatinib (PDB ID: 4RIX) were all retrieved from the RCSB database (Berman, 2000). Sequence similarity computations were performed

using BLOSUM62 (Trivedi and Nagarajaram, 2019). Modeling of sPDGFRA into the electron density map of imatinib-bound hPGDFRA was performed using Phenix (Liebschner et al, 2019) and Coot (Emsley et al, 2010). For ERBB3 experiments, induced-fit docking was performed on the ATP binding site with sapitinib using Schrödinger's Maestro IFD (Sherman et al, 2006) software. All structural images were generated using Maestro.

## Statistical analysis

Statistical analyses were performed using GraphPad Prism (v8.4.3) and R (v4.4.0), unless indicated otherwise. Comparisons between two groups were conducted with two-tailed unpaired Student's $t$ tests, and multiple group comparisons with one-way or two-way ANOVA. Sequencing statistics are included in Appendix Table S5. For whole-genome sequencing, allele frequencies were derived from VCF read counts using custom Python scripts (pysam v0.22.1, SciPy v1.15.2); differences between parental and resistant lines were tested by Fisher's exact test, with binomial tests applied to assess allele loss, and $P$ values adjusted for multiple testing using the FDR method in R. CNVs were analysed with Control-FREEC and ploidy inferred with the R package nQuack. Phosphoproteomics data were analysed in Perseus using Student's $t$ tests with FDR < 0.05 and s0 = 0.1. For cell viability assays, $IC_{50}$ values were obtained from nonlinear regression fits of dose–response curves, and drug combination effects were quantified with SynergyFinder (v2) using the ZIP model. Western blot and immunohistochemistry quantifications were normalised to controls and assessed with $t$ tests or ANOVA as appropriate. Unless otherwise indicated, data are presented as mean ± SD, and significance was defined as $P < 0.05$.

# Data availability

All raw whole genome sequencing data generated in this study have been deposited in the European Nucleotide Archive (ENA) under accession number PRJEB93974. The mass spectrometry proteomics data have been deposited to the ProteomeXchange Consortium via the PRIDE partner repository under dataset identifier PXD058352. Code used for in silico and statistical analyses of whole genome sequencing data has been uploaded to a GitHub repository, which is accessible at: https://github.com/luenling/Schoenbichler_2025. Any additional information required to reanalyse the data reported in this paper is available from the corresponding authors upon request.

The source data of this paper are collected in the following database record: biostudies:S-SCDT-10_1038-S44318-025-00603-0.

# Peer review information

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

## Acknowledgements

The authors would like to thank Safia Zahma from Animal Breeding and Genetics at the University of Veterinary Medicine Vienna for her technical support in histology. We thank APAF (Australian Proteome Analysis Facility) for their assistance in generating the original phosphopeptide dataset. We thank the Vienna Biocenter sequencing facility for their invaluable support and sequencing efforts, which greatly contributed to this work. This research was also supported by resources from the VetCore Facility (VetImaging | VetBiobank) at the University of Veterinary Medicine Vienna.

## Author contributions

**Anna Schönbichler**: Conceptualization; Formal analysis; Investigation; Visualization; Writing—original draft; Project administration; Writing—review and editing. **Anna Orlova**: Conceptualization; Investigation; Methodology. **Carmen Kreindl**: Investigation. **Lukas Endler**: Validation; Investigation; Writing—review and editing. **Richard Wilson**: Resources; Validation; Investigation; Writing—review and editing. **Lindsay Kosack**: Conceptualization; Investigation. **Anna Hofmann**: Investigation. **Csilla Viczenczova**: Investigation. **Jocelyn Darby**: Resources. **Fettah Erdogan**: Validation; Investigation. **Amanda L Patchett**: Investigation. **Anna Koren**: Resources; Investigation. **Stefan Kubicek**: Resources. **Mathias Müller**: Supervision; Writing—review and editing. **Andrew S Flies**: Supervision; Funding acquisition; Writing—review and editing. **Andreas Bergthaler**: Conceptualization; Supervision; Funding acquisition; Validation; Project administration; Writing—review and editing. **Richard Moriggl**: Conceptualization; Supervision; Funding acquisition; Validation; Project administration; Writing—review and editing.

Source data underlying figure panels in this paper may have individual authorship assigned. Where available, figure panel/source data authorship is listed in the following database record: biostudies:S-SCDT-10_1038-S44318-025-00603-0.

## Disclosure and competing interests statement

The authors declare no competing interests.

# Expanded View Figures

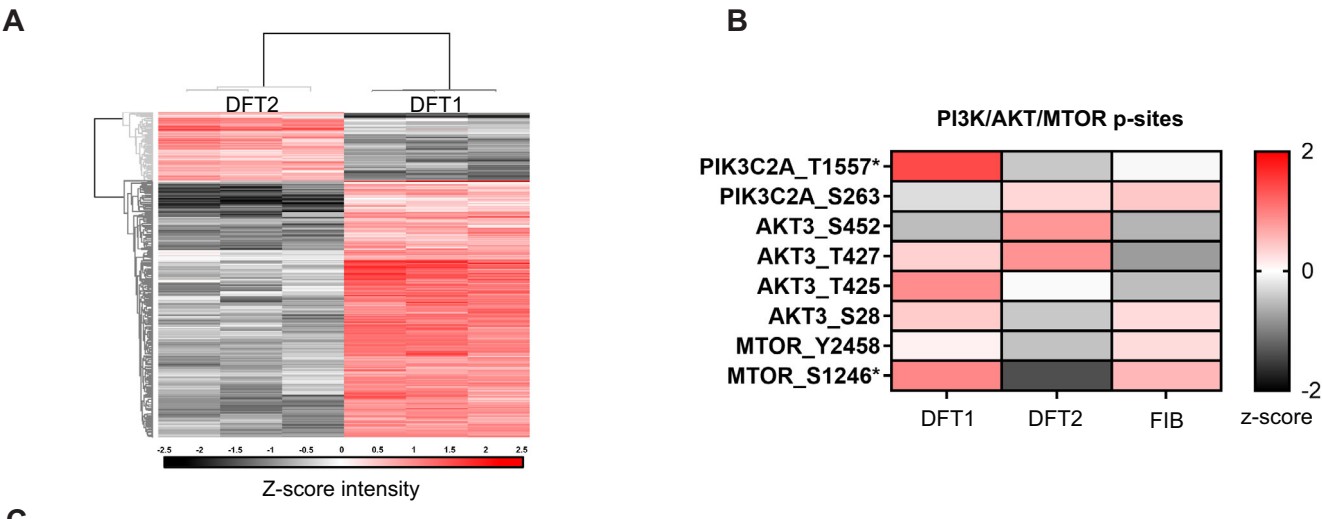

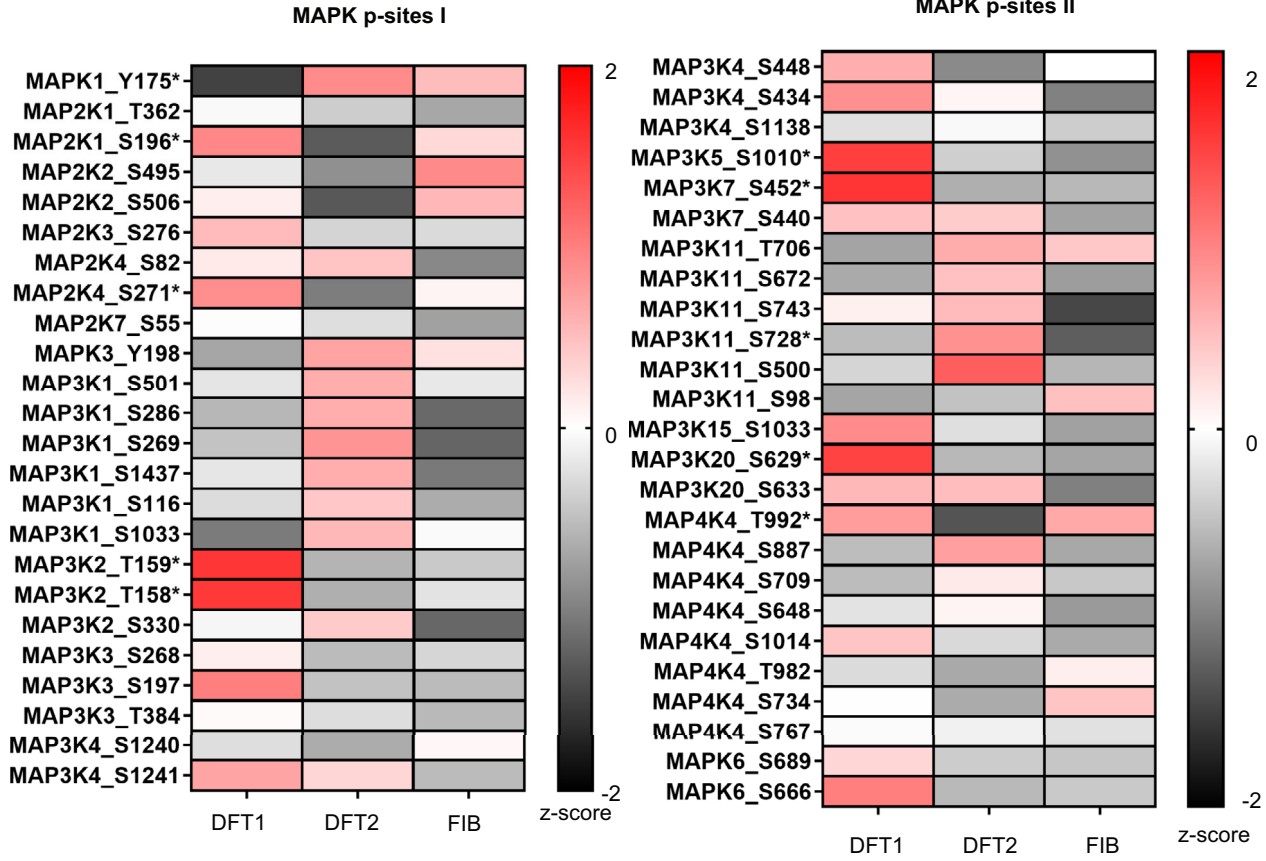

**Figure EV1.  Related to Fig. 1.**

(A) Heatmap of hierarchical clustering of z-scored log2 intensity values for significantly differentially abundant phosphopeptides (Fig. 1C). (B, C) Heatmaps of relative abundance (z-scored log2 intensity values) for peptides mapped to PI3K/AKT/mTOR pathway and MAPK pathway proteins. Stars indicate significant phosphorylation differences between DFT1 and DFT2 (*t* test, FDR < 0.05, s0 = 0.1).

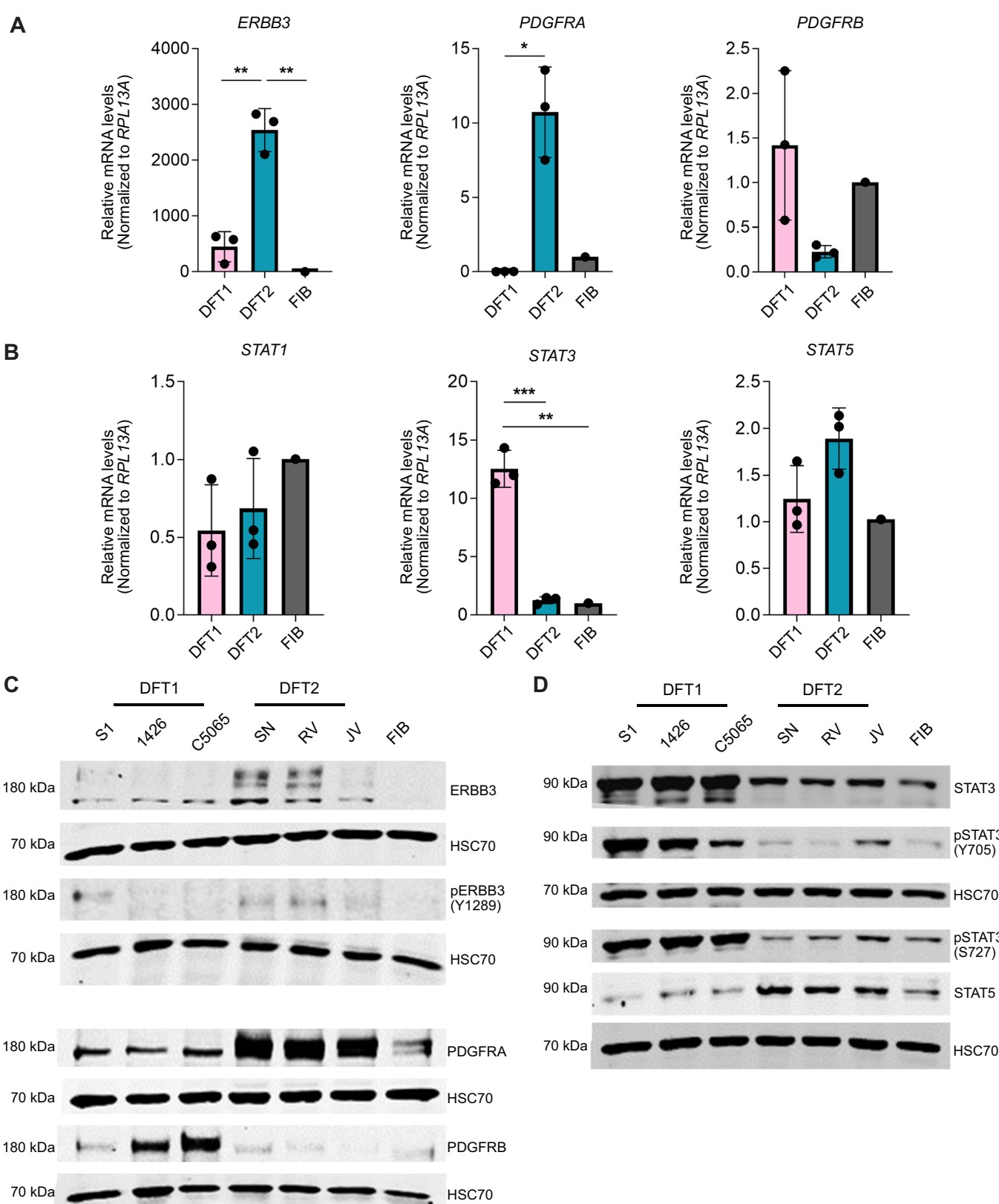

**Figure EV2.** **Related to Fig. 1.**

(A, B) Gene expression of *ERBB3, PDGFRA, PDGFRB* and downstream signalling molecules *STAT1, STAT3, STAT5\**) measured by qPCR and normalised to *RPL13A*. Data represent three DFT1 and three DFT2 cell lines (each dot corresponds to a cell line), shown as mean ± SD relative to a fibroblast control. Statistical analysis was performed using one-way ANOVA with Bonferroni's test. Statistical significance was set at $P < 0.05$ and is indicated as follows: $P < 0.0332$ (\*), $P < 0.0021$ (\*\*), $P < 0.0002$ (\*\*\*), and $P < 0.0001$ (\*\*\*\*). Exact $P$ values were as follows: *ERBB3* – DFT1 vs. DFT2: $P = 0.0045$; DFT2 vs. FIB: $P = 0.008$. *PDGFRA* – DFT1 vs. DFT2: $P = 0.0110$. *STAT3* – DFT1 vs. DFT2: $P = 0.0008$; DFT1 vs. FIB: $P = 0.0028$. Data shown are from one representative experiment of two independent replicates. (C, D) Western blot analysis of ERBB3, pERBB3 (Y1289), PDGFRA, PDGFRB, STAT3, pSTAT3 (Y705), pSTAT3 (S727) and STAT5\* in six DFT cell lines and one fibroblast control. HSC70 was the loading control. $n = 3$. \**The Tasmanian devil genome contains two genes more homologous to human STAT5B than STAT5A, suggesting the absence of STAT5A protein.* Source data are available online for this figure.

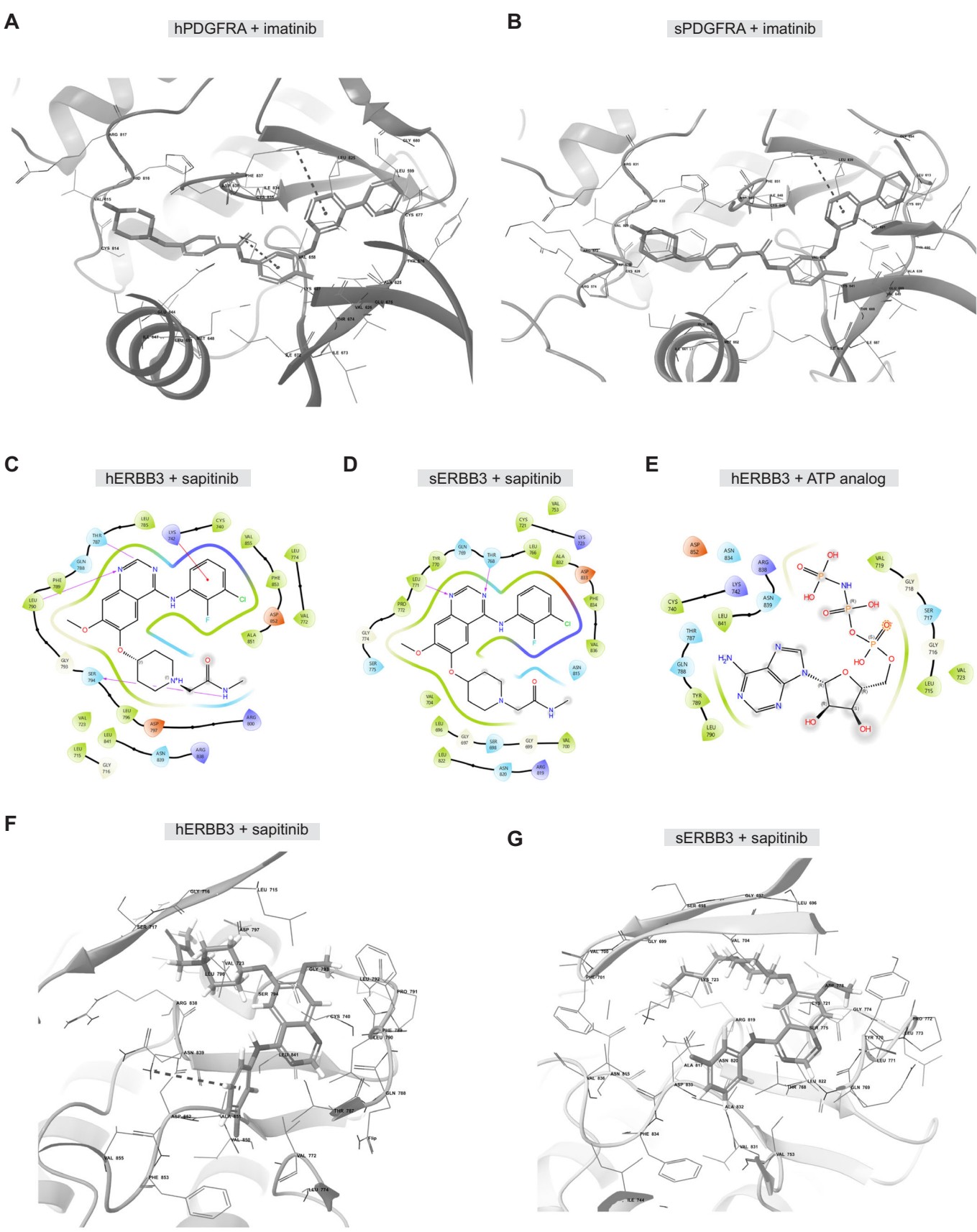

**Figure EV3. Associated with Fig. 2.**

(A, B) Structural depiction of human PDGFRA (hPDGFRA, **A**) bound to imatinib (PDB ID: 6JOL) and Tasmanian devil PDGFRA (sPDGFRA, **B**) modeled into the 6JOL structure, showing binding poses and interacting residues at the catalytic site of the kinase domain. (C–E) Residues of human ERBB3 (hERBB3, **C**) and Tasmanian devil ERBB3 (sERBB3, **D**) predicted to interact with sapitinib at the catalytic site of the kinase domain. For comparison, residues involved in binding an ATP analog ligand in the hERBB3 crystal structure (PDB ID: 4RIX) are shown (**E**). Note that residues in the 4RIX structure are truncated at the N-terminus. (F, G) Induced-fit docking of sapitinib with hERBB3 (**F**) and sERBB3 (**G**), illustrating predicted binding poses and interacting residues at the catalytic site.

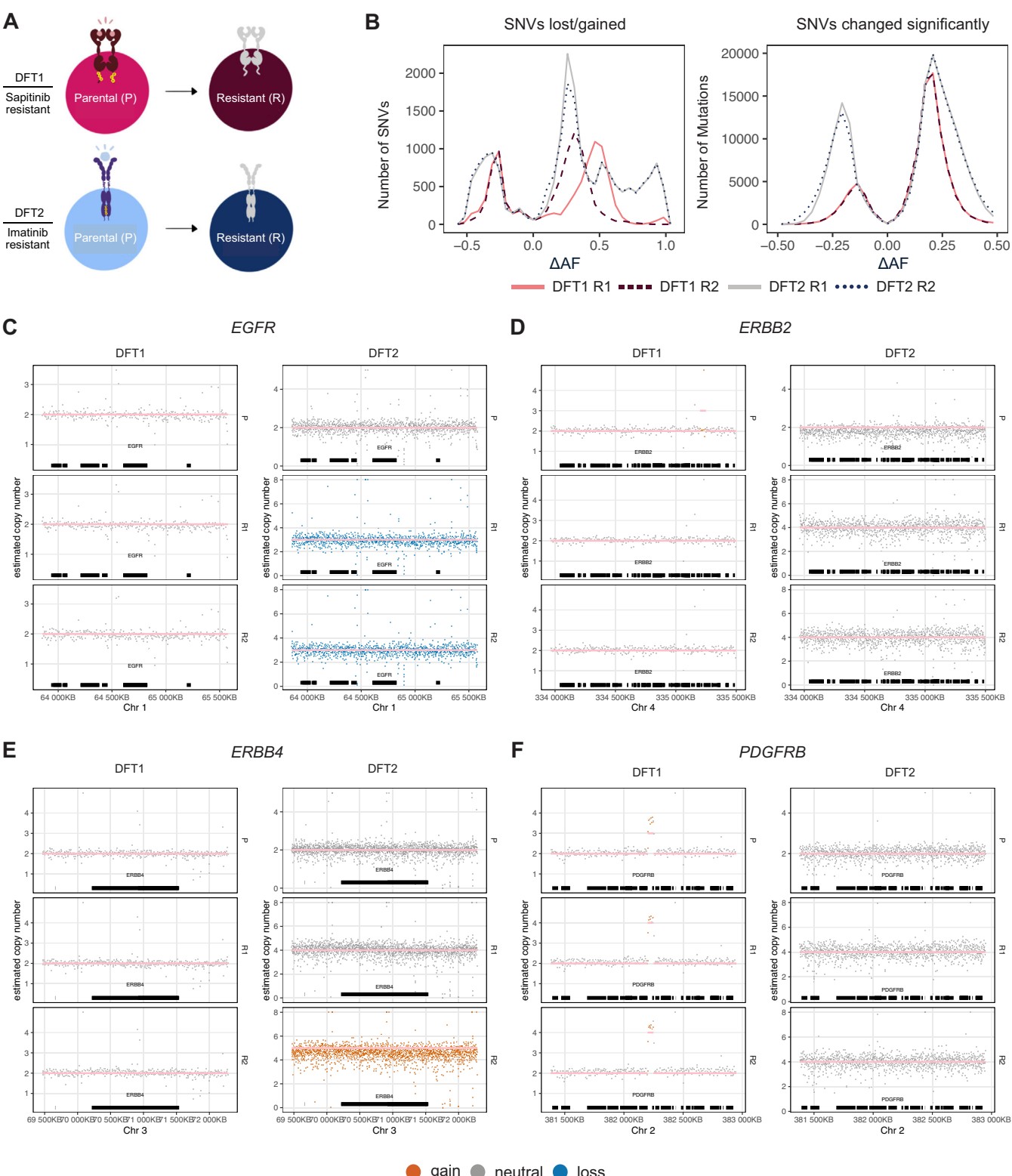

**Figure EV4.  Associated with Fig. 4.**

(**A**) Schematic showing the generation of sapitinib-resistant DFT1 and imatinib-resistant DFT2 cell lines. (**B**) Distribution of SNVs gained or lost in resistant lines. Left panel: ΔAF represents the change in minor allele frequency relative to the parental line. SNVs with ΔAF < 0 were considered lost; ΔAF > 0 indicates novel SNVs. Y-axis shows the number of mutations at each ΔAF bin. Filtering was performed using an FDR-corrected binomial test ($P < 0.1$). Right panel: SNVs showing statistically significant frequency changes between parental and resistant lines, identified using Fisher's exact test ($P < 0.05$). (**C–F**) Copy number (CN) profiles for *EGFR, ERBB2, ERBB4* and *PDGFRB. Normalised* CN rations were calculated by Control-FREEC using 5 kbp or 1 kbp genomic windows for DFT1 and DFT2 respectively and multiplied by the assumed ploidy. Genes are indicated as black bars. Copy number gains are shown in orange and losses in blue. Data are displayed in 5 kbp or 1 kbp windows. The pink line represents estimated copy number inferred by Control-FREEC.

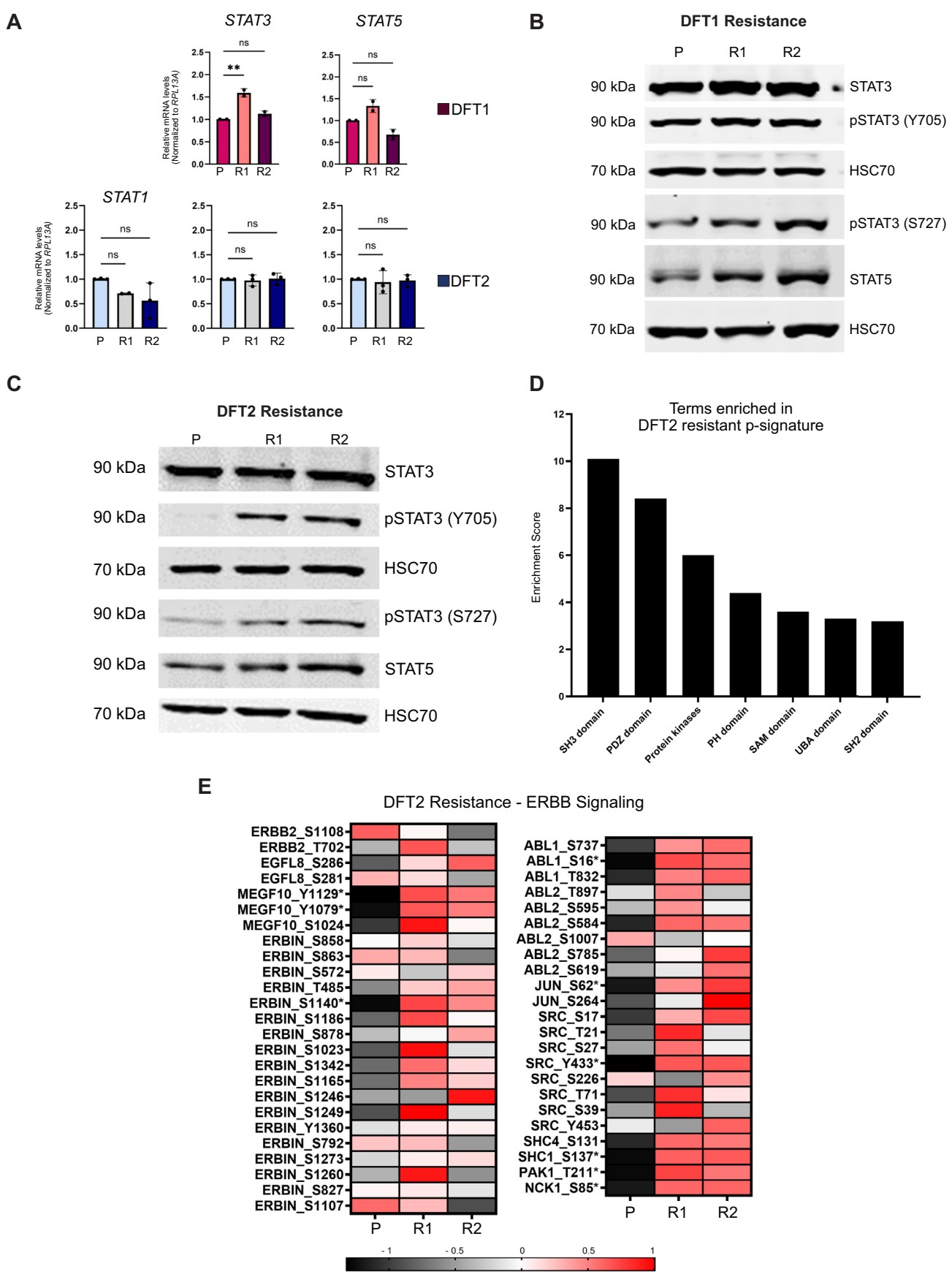

◀ **Figure EV5.  Associated with Fig. 5.**

(A) Gene expression of *STAT1, STAT3*, and *STAT5* in parental (P) and resistant (R1, R2) DFT1 (top) and DFT2 (bottom) cell lines, quantified by real-time PCR and normalised to *RPL13A*. Data are shown relative to the respective parental control and presented as mean ± SD ($n = 3$ replicates). Statistical analysis was performed using one-way ANOVA with Bonferroni's multiple comparisons test, comparing each group to the parental (P) group. Statistical significance was set at $P < 0.05$ and is indicated as follows: $P < 0.0332$ (*), $P < 0.0021$ (**), $P < 0.0002$ (***), and $P < 0.0001$ (****). STAT3 (DFT1) P vs. R1; $P = 0.0058$. (B, C) Western blot analysis of STAT3, pSTAT3 (Y705), pSTAT3 (S727), and STAT5 in parental and resistant DFT1 (B) and DFT2 (C) cell lines. HSC70 served as a loading control. $n = 2$. (D) Bar plot showing annotation terms enriched in the DFT2 resistant p-peptide signature. Terms were determined by functional annotation clustering using DAVID Bioinformatics, with selection based on enrichment score <3 and $P$ value < 0.05 (Bonferroni correction). (E) Heatmaps of the relative abundance (z-scored log2 intensity values) of ERBB pathway p-peptides in DFT2 parental and imatinib-resistant cell lines. *Significant differences in phosphorylation between parental and resistant cells lines ($P$ value < 0.05). Source data are available online for this figure.

