## [Peer Review File · The EMBO Journal]

Tyrosine kinase targeting uncovers oncogenic pathway plasticity in Tasmanian devil transmissible cancers

Anna Schönbichler, Anna Orlova, Carmen Kreindl, Lukas Endler, Richard Wilson, Lindsay Kosack, Anna Hofmann, Csilla Viczenczova, Jocelyn Darby, Fettah Erdogan, Amanda Patchett, Anna Koren, Stefan Kubicek, Mathias Müller, Andrew Flies, Andreas Bergthaler, and Richard Moriggl

Corresponding authors: Richard Moriggl (richard.moriggl@plus.ac.at) , Andreas Bergthaler (andreas.bergthaler@meduniwien.ac.at)

Review Timeline:

Submission Date:	31st Jan 25
Editorial Decision:	25th Mar 25
Revision Received:	23rd Jul 25
Editorial Decision:	3rd Sep 25
Revision Received:	17th Sep 25
Accepted:	6th Oct 25

Editor: Daniel Klimmeck

Transaction Report:

Dear Dr Moriggl,

Thank you again for the submission of your manuscript (EMBOJ-2025-120337) to The EMBO Journal, and providing us with a preliminary point-by-point response to the concerns raised by the referees. As mentioned, your study was assessed by three reviewers with expertise in cancer signaling and evolutionary biology, whose comments are enclosed below.

As you will see from their comments, the referees acknowledge the analysis and potential interest and value of your findings. However, they also express important concerns i.p. regarding completeness of the analysis and requirement of major complementation suggesting additional genomic analyses.

Given the overall interest stated and broader angle of your results, we are able to invite you to revise your manuscript experimentally to address the referees' comments, along the lines sketched in your outline. I need to stress though that we do require strong support from the referees on a revised version of the study in order to move on to publication of the work.

Please feel free to contact me if you have any questions or need further input on the referee comments.

When submitting your revised manuscript, please carefully review the instructions below.

Please feel free to approach me any time should you have additional questions related to this.

Thank you for the opportunity to consider your work for publication.

I look forward to your revision.

Kind regards,

Daniel Klimmeck

Daniel Klimmeck, PhD
Senior Editor
The EMBO Journal

Instruction for the preparation of your revised manuscript:

- 1) a .docx formatted version of the manuscript text (including legends for main figures, EV figures and tables). Please make sure that the changes are highlighted to be clearly visible.
- 2) individual production quality figure files as .eps, .tif, .jpg (one file per figure).
- 3) a .docx formatted letter INCLUDING the reviewers' reports and your detailed point-by-point response to their comments. As

part of the EMBO Press transparent editorial process, the point-by-point response is part of the Review Process File (RPF), which will be published alongside your paper.

4) a complete author checklist, which you can download from our author guidelines ([https://wol-prod-cdn.literatumonline.com/pb-assets/embo-site/Author Checklist%20-%20EMBO%20J-1561436015657.xlsx](https://wol-prod-cdn.literatumonline.com/pb-assets/embo-site/Author%20Checklist%20-%20EMBO%20J-1561436015657.xlsx)). Please insert information in the checklist that is also reflected in the manuscript. The completed author checklist will also be part of the RPF.

6) It is mandatory to include a 'Data Availability' section after the Materials and Methods. Before submitting your revision, primary datasets produced in this study need to be deposited in an appropriate public database, and the accession numbers and database listed under 'Data Availability'. Please remember to provide a reviewer password if the datasets are not yet public (see <https://www.embopress.org/page/journal/14602075/authorguide#datadeposition>).

7) Our journal encourages inclusion of *data citations in the reference list* to directly cite datasets that were re-used and obtained from public databases. Data citations in the article text are distinct from normal bibliographical citations and should directly link to the database records from which the data can be accessed. In the main text, data citations are formatted as follows: "Data ref: Smith et al, 2001" or "Data ref: NCBI Sequence Read Archive PRJNA342805, 2017". In the Reference list, data citations must be labeled with "[DATASET]". A data reference must provide the database name, accession number/identifiers and a resolvable link to the landing page from which the data can be accessed at the end of the reference. Further instructions are available at .

8) At EMBO Press we ask authors to provide source data for the main and EV figures. Our source data coordinator will contact you to discuss which figure panels we would need source data for and will also provide you with helpful tips on how to upload and organize the files.

Numerical data can be provided as individual .xls or .csv files (including a tab describing the data). For 'blots' or microscopy, uncropped images should be submitted (using a zip archive or a single pdf per main figure if multiple images need to be supplied for one panel). Additional information on source data and instruction on how to label the files are available at .

9) We replaced Supplementary Information with Expanded View (EV) Figures and Tables that are collapsible/expandable online (see examples in <https://www.embopress.org/doi/10.15252/embo.201695874>). A maximum of 5 EV Figures can be typeset. EV Figures should be cited as 'Figure EV1, Figure EV2' etc. in the text and their respective legends should be included in the main text after the legends of regular figures.

11) For data quantification: please specify the name of the statistical test used to generate error bars and P values, the number (n) of independent experiments (specify technical or biological replicates) underlying each data point and the test used to calculate p-values in each figure legend. The figure legends should contain a basic description of n, P and the test applied. Graphs must include a description of the bars and the error bars (s.d., s.e.m.).

The revision must be submitted online within 90 days; please click on the link below to submit the revision online before 23rd Jun 2025.

Referee #1:

Schönbichler et al (2025) present a thorough series of analyses on targeted kinase inhibition of Tasmanian devil facial tumour 1 (DFT1) and 2 (DFT2) cell lines.

This work is to be seen in context of two lethal, transmissible cancer epidemics which have been threatening the conservation of the Tasmanian devil in recent decades. Importantly, to this day no effective treatment or vaccine exists against these diseases, which underlines the relevance of gaining fundamental insight with respect to therapeutic potentials. Two major drug screens performed on DFT cell lines had been published before (Kosack et al., *Cancer Cell* 2019; Stammnitz et al., *Cancer Cell* 2018). These early studies highlighted the cancers' shared dependency on receptor tyrosine kinase (RTK) activity, a finding which was perhaps not too surprising in light of the two lineages' likely very similar origins from Schwann cells or precursors thereof. Notably, there appeared to be slightly different RTK signalling routes in DFT1 (via ERBB-STAT) vs DFT2 (via PDGFR).

This manuscript by Schönbichler et al provides a follow-up; it should be of value to our molecular understanding of DFTs and to future devil conservation efforts.

To my knowledge, methodological novelties of this work include:

- the first phosphoproteomics data set of DFT1 and DFT2 cells (maybe of any Tasmanian devil isolates?)
- the first generation of a DFT2 xenograft mouse model and its drug treatment
- the first generation of drug-resistant DFT1 and DFT2 cell lines.

Based on their data, the authors mainly conclude that exposure to specific RTK inhibitors renders particular phenotypes of DFT1 and DFT2 cell drug resistance.

In DFT1, there appears to be little difference between the resistant and parental cells' phosphoproteomes. Nonetheless, DFT1 resistance to Sunitinib - an ERBB2/ERBB3/EGFR receptor inhibitor - is characterised by reductions in ERBB3 levels, concomitant increases of PDGF receptor expression and thereby somewhat resemblance of the parental DFT2 phenotype. In contrast, long-term treatment of DFT2 cells with Imatinib/Gleevec - a non-specific first-generation kinase inhibitor originally developed against BCR-ABL1 fusions in CML, but here used to "target" PDGFR - caused resistance with a strongly diverging phosphoproteome to parental DFT2. Apart from upregulation of EGFR, a key finding is the reduction of MHC1 and B2M levels in Imatinib-resistant DFT2, which are also found in the parental DFT1 cell state including its low immunogenicity (e.g. see Siddle et al., *PNAS* 2013).

Overall, the authors conclude that DFT1 and DFT2 can respond to targeted RTK treatments by switching to alternative signalling pathways to maintain growth, and mutually reflecting the other DFT lineages' parental state. While it is to be expected that the cells do develop resistance over time, this inference of "plasticity" - if correct - would be new to the community and likely influence future chemotherapy attempts.

Interestingly, results from combination therapy against both ERBB and PDGFR signalling show that significant synergistic benefits are only obtained for Sunitinib-resistant DFT1 (Figure 3C) but not Imatinib-resistant DFT2 (Supplementary Figure S6C). This dichotomy and the associated challenges are important and need to be highlighted more throughout the story, especially in the abstract and final paragraph of the conclusion. Mainly together with the phosphoproteomics comparison of parental vs. resistant DFT (PC1 in Fig. 3B still captures ~30% (!) of variation), these data rather indicate more complex, nuanced scenarios than a "common resistance mechanism" by simply switching towards the other lineages' parental state. Clearly, there exist at least moderate differences between the resistant DFT lineages' signalling states and treatment vulnerabilities, and it would be desirable if the authors can dig much deeper into the involved mechanistic diversity.

There are a number of experimental avenues in which the drug resistance state of DFTs could be further explored. For example, sequencing the genome of the resistant DFT cells and pinpointing the key genetic drivers of resistance could strongly increase the impact of this story. It would allow the authors to render a more complete and definitive picture of drug resistance, especially if certain mutations appear recurrently between replicates, while complementing such insights with their phosphoproteomics, western blot and qPCR data: could there be PDGFR or ligand-associated copy number gains in DFT1? Or PDGFR inactivation and EGFR gains in DFT2? Could this maybe also inform on the remaining druggability differences between the resistant DFT1s vs DFT2s? The authors briefly mention the potential role of fusion genes in the discussion, and I believe one should address it with more data. So please, if there is any possibility, do sequence the resistant DFTs. Though if not, how about analysing full transcriptomic profiles of the resistant vs. parental DFTs - and comparing both against plenty of published normal tissue RNA-seq datasets? Or a functional screen with RNAi, CRISPR, or the like on the resistant cells, to depict and characterise relevant genes involved in the transition? Beyond any data generation, it would be very helpful if the authors can hypothesise in detail on expectations of DFT resistance evolution - "how might plasticity be genetically encoded?"

Beyond the major points on the preliminary nature of some of the results and their overall interpretation, this manuscript would benefit from a set of broad clarifications and refinements:

1.) Many kinase inhibitors lack specificity, largely because of the conservation of the kinases' active site which is the primary target of most small molecules. The here applied use of Imatinib as a PDGFR inhibitor, which is also known to act against FLT3, ABL, KIT, etc. is a prime example. More generally, how many of the "signalling target categories" listed in Figure S3A should be truly considered specific in this screen? It would be great if the authors could add a paragraph on this topic in their discussion, re-assess their main findings in light of such potential ambiguities and also cite relevant literature. Also, could the authors compare their results with the kinase inhibition data from the earlier two Cancer Cell studies? It would be helpful if the authors can point out which RTK drugs in their compound library had not been tested before, and why this particular library was chosen over other options. Another conceptual question worth discussing is why Sapatinib and Imatinib were chosen as the only inhibitors to generate resistant cell lines. What resistance phenotypes might be caused by a third, bi-specific agent with relatively high potency against both paternal DFT1 and paternal DFT2 (e.g. perhaps Afatinib)?

2.) Related to the first point, the screen here was performed on Tasmanian devil cells with fibroblasts as the only non-cancer control. There are very few - if any - other devil cell lines available which could be used as alternative controls. Though in light of that, it is crucial that the authors add some discussion with respect to key confounder risks like general small molecule toxicity (in devil cells) and devil-RTK specific effects of the used kinase inhibitors. It might help to discuss this particularly in the context of Sapatinib and Imatinib, perhaps borrowing from literature on human or mouse cells. One orthogonal idea: the authors could also make AlphaFold2 structure predictions of the devil ERBB and PDGFR sequences, compare these to human high-quality crystal structures of the same receptors (+ inhibitor) and assess if these molecules would still be likely to bind to the DFT receptors.

3.) Throughout the manuscript, the authors comment on the "speed" of the resistance obtained in vitro. E.g. "DFT1 rapidly developed resistance" and "DFT2 cells show a slower progression to resistance". But, unless I have missed this somewhere, there is no associated data - could you please add a plot to quantify and summarise these findings? Also, in terms of the discussion, which factors are likely to contribute to this difference between the two DFT lineages?

4.) Data display and statistical analyses (for the main figures only):

- Fig 1: can the authors please comment on statistical power and add such considerations to the discussion. E.g. are three replicates of DFT1, DFT2 and fibroblast cells sufficient to call significant phospho-peptide enrichment? Could there be technical reasons for why there are more significant DFT1 over DFT2 p-sites? Across analyses: why do you believe that relatively few p-sites stand out strongly?
- Fig 1C: please highlight key kinase peptides in the volcano plot, esp. JAK/STAT and PDGFR ones shown in panels E and F
- Fig 2AB: please instead show a full heatmap with IC50 (log) scores with all drugs vs. all cell lines tested; the binary categorisation of "hit" vs "no hit" in the Venn diagram does not reflect the expected differences in IC50; moreover, the colouring by ERBB vs. PDGFR inhibition is questionable with respect to inhibitor specificity (see point 1)
- (current) Fig 2B: it would be helpful to clarify if this is displayed on the log scale and which IC50 unit is meant
- Fig 2C-G: it is a bit difficult to understand why the DFT2 PDX model is merged with the DFT1 and DFT2 cell line drug screen into a single figure. While it is certainly a great new experimental system, these xenograft data don't fit much with the overall story on DFT resistance.
- Fig 3B: it is probably cleaner to flip the X and Y axes to stay consistent with the standard scheme of displaying PC1 on the X and PC2 on Y.
- Fig 3C and 3D: compared to Fig 1C, it looks like the cut-off for significant "hits" is differently calculated between the different sets of volcano plots? For consistency with the suggested Fig. 1 edits, could the authors also highlight key peptides in these volcano plots which are of relevance to panels E-G of the same figure (e.g. JAK/STAT hits)?
- Fig 3F: what is the meaning of the different categories of significance (*, **, ***, ****)?
- Fig 4B: fitting a line through these three data points and the associated R2 value do not provide additional value, could you please remove these?
- Fig 4D (like 3F): what is the meaning of the different categories of significance?
- Use of colours: this is of course a bit subjective and more of an editorial comment, but I believe that the colour scheme may be substantially improved. Currently, readers need to distinguish between: DFT1 (red), DFT2 (blue), enrichment scores, Z-scores,

IC50 values (red/red-black/pink), Sunitinib (red), Imatinib (blue), and ZIP synergy scores (red-green). Some of the former papers on DFT have used the opposite colour scheme of DFT1 (blue) vs. DFT2 (red) which adds a bit to the confusion, too.

Referee #2:

The Tasmanian devil's facial tumour (DFT) is a clonal cancer that transmits naturally as an allograft. Two DFTs are known, DFT1, and the more recently discovered DFT2. The authors previously reported that DFT1 is mainly driven by dysregulated ERBB3/STAT3 signalling whereas DFT2 is mainly driven by dysregulated PDGFRA signalling. Here, the authors performed global phosphoproteomics analyses on DFT1 and DFT2, which support and expand their previous conclusions on the oncogenic drivers. The authors then conducted a screening on DFT1 and DFT2 cell lines to assess their susceptibility to a panel of tyrosine kinase inhibitors (TKI). They found that DFT1 cells are highly susceptible to Sunitinib (EGFR inhibitor) whereas DFT2 cells are highly susceptible to Imatinib (PDGFRs inhibitor), in agreement with the predicted outcomes based on the phosphoproteomics results. Drug-resistant DFT1 and DFT2 cells were selected and mechanisms of resistance were analysed, showing that DFT1 broadly maintains the key TKI signalling pathways intact in contrast to DFT2, which drifts from PDGF to ERBB signalling.

The paper is interesting and well written, and most of the data presented are of high quality. Understanding mechanisms of DFT resistance is important for possible treatment of captive devils and the basics mechanisms of clonal cancer evolution. However, the paper also has weaknesses that should be addressed.

1. While the shift in the TKI signalling pathways is convincing for DFT2 resistant cells (DFT2-R), it is less convincing for DFT1-R cells. Suppl. Figure 5D shows variable and modest changes in DFT1-R1 and R2 relative to parental (P) and the Western blot in panel F also shows minimal differences. These results suggest that the resistance may be due to mutations in the drug target, and it is surprising that the authors have not tested this common mechanism of resistance.
2. Phosphoproteomics did not detect p-ERBB3, however the Western blot for ERBB3 in Figure 2C shows bands above ERBB3 that look like phosphorylated ERBB3 forms, although it could also be ubiquitination. In any case, it should be fairly straightforward to test this by Western blotting using phospho-specific ERBB3 antibodies.
3. It is at times difficult to match the text with the Figures because one needs constantly to jump from main to Supplemental. Reorganising the Figures to show the key results for DFT1 and DFT2 side by side, or at least in consecutive Figures would help the reader.
4. Line 209 should refer to Figure 3C.
5. Line 379, it should read "prove".

1 EMBO Journal-Referee #1:

Schönbichler et al (2025) present a thorough series of analyses on targeted kinase inhibition of Tasmanian devil facial tumour 1 (DFT1) and 2 (DFT2) cell lines.

This work is to be seen in context of two lethal, transmissible cancer epidemics which have been threatening the conservation of the Tasmanian devil in recent decades. Importantly, to this day no effective treatment or vaccine exists against these diseases, which underlines the relevance of gaining fundamental insight with respect to therapeutic potentials. Two major drug screens performed on DFT cell lines had been published before (Kosack et al., *Cancer Cell* 2019; Stammnitz et al., *Cancer Cell* 2018). These early studies highlighted the cancers' shared dependency on receptor tyrosine kinase (RTK) activity, a finding which was perhaps not too surprising in light of the two lineages' likely very similar origins from Schwann cells or precursors thereof. Notably, there appeared to be slightly different RTK signalling routes in DFT1 (via ERBB-STAT) vs DFT2 (via PDGFR).

This manuscript by Schönbichler et al provides a follow-up; it should be of value to our molecular understanding of DFTs and to future devil conservation efforts.

To my knowledge, methodological novelties of this work include:

- the first phosphoproteomics data set of DFT1 and DFT2 cells (maybe of any Tasmanian devil isolates?)
- the first generation of a DFT2 xenograft mouse model and its drug treatment
- the first generation of drug-resistant DFT1 and DFT2 cell lines.

Based on their data, the authors mainly conclude that exposure to specific RTK inhibitors renders particular phenotypes of DFT1 and DFT2 cell drug resistance.

In DFT1, there appears to be little difference between the resistant and parental cells' phosphoproteomes. Nonetheless, DFT1 resistance to Sunitinib - an ERBB2/ERBB3/EGFR receptor inhibitor - is characterised by reductions in ERBB3 levels, concomitant increases of PDGF receptor expression and thereby somewhat resemblance of the parental DFT2 phenotype. In contrast, long-term treatment of DFT2 cells with Imatinib/Gleevec - a non-specific first-generation kinase inhibitor originally developed against BCR-ABL1 fusions in CML, but here used to "target" PDGFR - caused resistance with a strongly diverging phosphoproteome to parental DFT2. Apart from upregulation of EGFR, a key finding is the reduction of MHC1 and B2M levels in Imatinib-resistant DFT2, which are also found in the parental DFT1 cell state including its low immunogenicity (e.g. see Siddle et al., *PNAS* 2013).

Overall, the authors conclude that DFT1 and DFT2 can respond to targeted RTK treatments by switching to alternative signalling pathways to maintain growth, and mutually reflecting the other DFT lineages' parental state. While it is to be expected that the cells do develop resistance over time, this inference of "plasticity" - if correct - would be new to the community and likely influence future chemotherapy attempts.

Interestingly, results from combination therapy against both ERBB and PDGFR signalling show that significant synergistic benefits are only obtained for Sunitinib-

resistant DFT1 (Figure 3C) but not Imatinib-resistant DFT2 (Supplementary Figure S6C). This dichotomy and the associated challenges are important and need to be highlighted more throughout the story, especially in the abstract and final paragraph of the conclusion. Mainly together with the phosphoproteomics comparison of parental vs. resistant DFT (PC1 in Fig. 3B still captures ~30% (!) of variation), these data rather indicate more complex, nuanced scenarios than a "common resistance mechanism" by simply switching towards the other lineages' parental state. Clearly, there exist at least moderate differences between the resistant DFT lineages' signalling states and treatment vulnerabilities, and it would be desirable if the authors can dig much deeper into the involved mechanistic diversity.

Referee #1:

Thank you for the time, advice and expertise given in evaluating our manuscript. Your thoughtful comments not only captured the key novelties of our work but also helped us improve the clarity and depth of our findings. We greatly appreciate your creative suggestions, which led to several important additions. We reframed several conclusions and strengthen our main hypothesis along critical and constructive suggestions.

To address the main points we have:

- Clarified that imatinib should be interpreted as a selective inhibitor targeting BCR-ABL, KIT, PDGFRs and FLT3, while reasoning that its effect in DFT2 is likely mediated via PDGFRA, due to its specific upregulation.
- Included structural modelling, as suggested, to show conservation of imatinib- and sapitinib-interacting residues between Tasmanian devils and humans.
- Included WGS of parental and resistant DFT1 C5065 and DFT2 SNUG cell lines and expanded the analysis of our experimental model (largely guided by the valuable Stammnitz et al. 2018 and Stammnitz et al. 2023 data compilations).

We hope that our revised manuscript provides a fundamental contribution to the field of transmissible cancer biology and please see our point for point response below.

1.1. Major comment: Whole genome sequencing of parental and resistant cell lines

There are a number of experimental avenues in which the drug resistance state of DFTs could be further explored. For example, sequencing the genome of the resistant DFT cells and pinpointing the key genetic drivers of resistance could strongly increase the impact of this story. It would allow the authors to render a more complete and definitive picture of drug resistance, especially if certain mutations appear recurrently between replicates, while complementing such insights with their phosphoproteomics, western blot and qPCR data: could there be PDGFR or ligand-associated copy number gains in DFT1? Or PDGFR inactivation and EGFR gains in DFT2? Could this maybe also inform on the remaining druggability differences between the resistant DFT1s vs DFT2s? The authors briefly mention the potential role of fusion genes in the discussion, and I believe one should address it with more data. So please, if there is any possibility, do sequence the resistant DFTs. Though if not, how about analysing full transcriptomic profiles of the resistant vs. parental DFTs - and comparing both against plenty of published normal tissue RNA-seq datasets? Or a functional screen with RNAi, CRISPR, or the like on the resistant cells, to depict and characterise relevant genes involved in the transition? Beyond any data generation, it would be very helpful if the authors can hypothesise in detail on expectations of DFT resistance evolution - "how might plasticity be genetically encoded?"

We agree with the reviewer and to meet the concern and to follow the suggestion, we performed Illumina whole-genome sequencing (WGS) on all six cell lines used in this study: DFT1 parental, resistant 1 (R1), and resistant 2 (R2), as well as DFT2 parental, resistant 1 (R1), and resistant 2 (R2) cell lines. The results of this analysis are now presented in a new **Figure 4**, with additional new data provided in Expanded Figures and Appendix Tables (see summary Table on page 2 of the rebuttal document for an overview).

To ensure robust and interpretable results, we performed additional sequencing runs, which improved sequencing depth and strengthened our dataset quality. While time constraints precluded certain analyses such as fusion gene and structural variant discovery, our study provides valuable insights into the genetic changes in DFT1 and DFT2 tumor model systems under drug-induced selective pressure.

In brief, although we called similar numbers of single nucleotide variants (SNVs) for both DFT1 and DFT2, resistant DFT2 cells exhibited ~60% more significantly altered SNVs and twice as many gained or lost variants, suggesting greater genomic disruption. A higher proportion of novel SNVs reached fixation in DFT2, consistent with stronger selection during resistance evolution (**See new Table 2 and new Fig. EV4B**).

Fig. EV4B: Distribution of SNVs gained or lost in resistant lines. Left panel: ΔAF represents the change in minor allele frequency relative to the parental line. SNVs with $\Delta AF < 0$ were considered lost; $\Delta AF > 0$ indicates novel SNVs. Y-axis shows the number of mutations at each ΔAF bin. Filtering was performed using an FDR-corrected binomial test ($P < 0.1$). Right panel: SNVs showing statistically significant frequency changes between parental and resistant lines, identified using Fisher's exact test ($P < 0.05$).

Copy number analysis revealed contrasts between tumour types: while DFT1-resistant lines showed only minor CNV changes, DFT2-resistant lines underwent widespread genomic reorganization (**Fig. 4C-D**). Predicted tetraploidy in DFT2 was supported by allele frequency patterns, Control-FREEC modelling, and independent validation via nQuack analysis (See **Appendix Table S8** for details).

Fig. 4C-D: Copy number (CN) as estimated by ControlFREEC across chromosomes 1–6 and X for DFT1 (C) and DFT2 (D) parental cell lines and their resistant derivatives (R1 and R2). Each dot represents the copy number ratio, that is the normalized read count within a 50 kbp genomic window, calculated as the read count divided by the genome-wide median, multiplied by the ploidy (ploidy 2: DFT1 P/R1/R2 and DFT2 P; ploidy 4: DFT2 R1 and R2). CN gains (red) and losses (blue) were inferred from the median signal across adjacent windows. The pink line represents estimated copy number inferred by ControlFREEC.

Gene-level copy number changes provide further mechanistic insight. For example, resistant DFT2 cells showed gains in ERBB3 and ERBB4, suggesting co-selection for ERBB

signalling, while PDGFRA was maintained at eight copies despite surrounding regional loss, indicating strong selective pressure to retain PDGFR signalling at high momentum during imatinib exposure (**Fig. 4E-F, Fig. EV4**). These changes were observed in both independent DFT2-resistant lines, pointing to a reproducible resistance trajectory. These new genetic data were carried out with a bioinformatics expert Dr. Lukas Endler who also is a new coauthor in the revised version of this manuscript.

Fig. 4E-F: Copy number profiles for *ERBB3* (E) and *PDGFRA* (F). Normalised CN ratios were estimated by ControlFREEEC using 5 kbp genomic bins. Genes are indicated as black bars.

In contrast, the more rapid acquisition of resistance in DFT1, accompanied by fewer genomic alterations, suggests that pre-existing pathway redundancies, rather than large-scale genomic reconfiguration, may have enabled adaptive resistance. This is consistent with the observation that DFT2 has a higher mutation rate in nature as elegantly demonstrated by Stammnitz and coworkers in 2023. While we remain cautious about drawing direct conclusions from our *in vitro* model, these experimental results do mirror the evolutionary tendencies seen in transmissible tumours *in vivo*. We view that our work might contribute to highlight the dependence on these drivers particularly under drug selection, where we can also confirm critical results made by others with their important work to identify the importance of these drivers.

We also share your interest in the possible role of fusion genes and structural variants in driving resistance, especially given their importance in DFT tumour evolution (Stammnitz et al., 2018, 2023). However, due to limited time and available resource expertise to identify fusion genes and structural variants as well as due to limited in depth sequence reads on the various models for resistance with paired genetic parental controls, we decided to place the SNV data or the fusion protein and structural variant analysis as open source to the research community. This sequencing resource will accompany the revised version of this manuscript and all interested researchers can access it. Thus, we focused our efforts on analyses where we could ensure robustness and reproducibility and where we clearly obtained new insights into the importance of cancer genetic analysis. For this reason, we did not include preliminary data on structural variants or fusion genes yet, but community efforts can be made. We have now added a sentence to the Discussion acknowledging the potential

importance of these mechanisms and we agree that this remains an exciting avenue for future work (**Lines 467-471**).

Similarly, we are enthusiastic about integrating RNA sequencing and functional screens to define drivers of resistance. These are logical next steps that would allow deeper characterization of regulatory changes and phenotypic transitions in resistant DFT cells.

Finally, in response to your suggestion, we have expanded our discussion of how plasticity might be genetically encoded (**see lines 435-446**). Although our system is simplified, it provides an opportunity to explore how different baseline tumour architectures might predispose to either pathway switching (DFT1) or large-scale genomic remodelling (DFT2). We hope this conceptual framework will stimulate further investigation, and we are open to collaboration with groups exploring related questions.

Minor comments

Beyond the major points on the preliminary nature of some of the results and their overall interpretation, this manuscript would benefit from a set of broad clarifications and refinements:

1.2. Minor comment: TKI Target Specificity and Inhibitor Promiscuity

A) Many kinase inhibitors lack specificity, largely because of the conservation of the kinases' active site which is the primary target of most small molecules. The here applied use of Imatinib as a PDGFR inhibitor, which is also known to act against FLT3, ABL, KIT, etc. is a prime example. More generally, how many of the "signalling target categories" listed in Figure S3A should be truly considered specific in this screen? It would be great if the authors could add a paragraph on this topic in their discussion, re-assess their main findings in light of such potential ambiguities and also cite relevant literature. B) Also, could the authors compare their results with the kinase inhibition data from the earlier two Cancer Cell studies? It would be helpful if the authors can point out which RTK drugs in their compound library had not been tested before, and C) why this particular library was chosen over other options. D) Another conceptual question worth discussing is why Sapatinib and Imatinib were chosen as the only inhibitors to generate resistant cell lines. G) What resistance phenotypes might be caused by a third, bi-specific agent with relatively high potency against both paternal DFT1 and paternal DFT2 (e.g. perhaps Afatinib)?

Thank you for raising the important issue of kinase inhibitor specificity and its implications for interpreting our findings.

To address this, we have now expanded the classification of target categories in the revised manuscript (**Figure 2A and Appendix Table S5**) to better reflect the complexity and overlap in kinase inhibitor activity. Furthermore, we have included a TKI target network analysis (**Appendix Figure S2**), which illustrates the breadth of known target profiles for the compounds included in our screen. These additions are now discussed also in lines **381-397** of the revised Discussion.

Fig. 2A: Venn diagram showing hits from a focused tyrosine kinase inhibitor (TKI) drug screen that met selection criteria in DFT1 and/or DFT2 cell lines compared to healthy fibroblasts. Target specificity of inhibitors is indicated using category-specific icons as shown in the Figure legend.

The TKI library used in our screen was curated in-house, as clarified in the revised Methods section (**lines 634-637**). Compound selection was guided by prior experience in our laboratory and we focused on specific agents (including imatinib which has still a quite selective TKI profile against PDGFR, KIT, SRC and ABL kinase family members compared e.g. to dasatinib that targets many more TKs) with clinical relevance or prior usage in DFT studies. This included inhibitors previously tested on DFT1 and/or DFT2, particularly those with inconclusive or context-specific effects. This rationale is now clearly outlined in the revised manuscript (**lines 185-189**). We would like to mention that our data support merit to use a broader acting TKI regimen since switching capacity under TKI in DFT1 and DFT2 model cancer cells was more prevalent in DFT1 with high plasticity and cancer switching capacity to adopt to selection pressure.

Regarding comparisons to previous work, we have annotated in the updated **Appendix Table S5** which inhibitors overlap with those tested in the Cancer Cell studies and highlighted compounds unique to our screen.

As for the choice of sapitinib and imatinib for generating resistant cell lines, we now provide additional justification in **lines 185-189**. Both drugs displayed clear lineage-specific activity, with sapitinib being more effective in DFT1 and imatinib in DFT2, making them appropriate models for exploring resistance in the context of lineage-biased TKI sensitivity.

Although afatinib was considered as a dual-active agent, its effects on both tumour cells and fibroblasts in our assays raised concerns about off-target toxicity. Moreover, afatinib did not emerge as a top hit in the compound screens performed by Stammnitz et al. 2018, reducing its prioritization for resistance modelling.

Finally, we acknowledge the broader conceptual point raised: the use of multi-target TKIs complicates interpretation of resistance phenotypes. We now explicitly discuss this in the revised Discussion. Despite this, we believe the comparative framework of our study remains valid and informative, particularly as resistance outcomes diverged significantly between DFT1 and DFT2 despite use of “broad-spectrum” imatinib as one suggestion of many we chose from a clinically widely used and still successful TKI.

1.3. Minor comment: Species-Specific Considerations

Related to the first point, the screen here was performed on Tasmanian devil cells with fibroblasts as the only non-cancer control. There are very few - if any - other devil cell lines available which could be used as alternative controls. A) Though in light of that, it is crucial that the authors add some discussion with respect to key confounder risks like general small molecule toxicity (in devil cells) and B) devil-RTK specific effects of the used kinase inhibitors. It might help to discuss this particularly in the context of Sapitinib and Imatinib, perhaps borrowing from literature on human or mouse cells. One orthogonal idea: the authors could also make AlphaFold2 structure predictions of the devil ERBB and PDGFR sequences, compare these to human high-quality crystal structures of the same receptors (+ inhibitor) and assess if these molecules would still be likely to bind to the DFT receptors.

Thank you for this creative suggestion and advice, which we followed by new analysis with an expert in structural analysis and MedChem expertise with TKI. In response, we performed structure-based modelling of the Tasmanian devil ERBB and PDGFR proteins using AlphaFold2 and various database open resource structural data with Dr. Fettah Erdogan, University of Toronto, who is also a new coauthor. We compared the predicted binding interfaces with high-resolution human crystal structures co-crystallized with sapitinib or imatinib. These interesting and as find quite nicely fitting results are now included in **Figure 2D-E and Fig. EV3** of the revised manuscript.

Fig. 2D-E: Residues of human PDGFRA (hPDGFRA, panel D) and Tasmanian devil PDGFRA (sPDGFRA, panel E) that interact with imatinib at the catalytic site of the kinase domain. Structural data and electron density maps for the hPDGFRA kinase domain bound to imatinib were obtained from the RCSB Protein Data Bank (PDB ID: 4RIX).

Our comparative analysis revealed a high degree of structural conservation at the inhibitor-binding sites between the devil and human orthologs. Key residues known to mediate binding in human ERBB and PDGFR family members are preserved in the devil sequences, supporting the likelihood that both sapitinib and imatinib engage their respective targets in a comparable fashion. This strengthens confidence that the lineage-specific responses we observed are indeed due to meaningful pharmacological interactions with these RTKs.

That said, we fully agree that species-specific off-target effects and broader small molecule sensitivities remain important considerations, especially given the limited availability of Tasmanian devil non-cancer cell lines. In our screen, fibroblasts served as the only non-transformed control, and we now explicitly discuss the limitations of this approach in the revised Discussion (**lines 398-408**). While our structural findings suggest that major target interactions are likely conserved, we emphasize that off-target effects cannot be completely excluded, but such potential risks exist for any drug-targeting research study. We also acknowledge that Tasmanian devil-specific RTK sequence differences outside the core kinase domain may affect drug responsiveness or downstream signalling fidelity. However, we want to conclude that STAT3 or also STAT5 are two SH2 domain containing molecules (from ~160 in humans) that display a remarkable conservation during evolution in devils or humans, suggesting that signalling integration from TK above is remarkably similar. STAT3 has only one amino acid missing in the last exon and there is another hydrophobic amino acid exchange in 770 amino acids of full length STAT3 that can be called highly conserved, with critical pY or pS at the exact same positions totally conserved between men and devils. The complete signals obtained from upstream TK are thus in this case superimposable on the STAT proteins from biochemical aspect compared to human cancer. Why in evolution TK signalling displays such a conserved conundrum is interesting to display and to possibly conclude. We describe an archaic tumour pathway that at the end might not be so different in some human solid cancer like colorectal cancer, lung cancer, head and neck cancer, pancreatic cancer that are all significantly driven by ERBB family members and again STAT3/5B oncoproteins in concert with PDGFR as well.

We hope this integrated structural and functional perspective addresses concern to add more valuable context for interpreting our compound screening data correctly.

1.4. Minor comment: Resistance establishment

A) Throughout the manuscript, the authors comment on the "speed" of the resistance obtained in vitro. E.g. "DFT1 rapidly developed resistance" and "DFT2 cells show a slower progression to resistance". But, unless I have missed this somewhere, there is no associated data - could you please add a plot to quantify and summarise these findings? B) Also, in terms of the discussion, which factors are likely to contribute to this difference between the two DFT lineages?

We thank you for highlighting the need to quantify the observed differences in resistance kinetics and we followed the suggestion. Thus, we have now included a resistance timeline in **Figure 4 (Panel A)**, which summarizes the duration of dose escalation and resistance acquisition in both DFT1 and DFT2 lines over time.

Fig. 4A: Resistance development over time. Two independent DFT1 cell lines were cultured with gradually increasing concentrations of sapitinib, and DFT2 cell lines with increasing concentrations of imatinib. Cells were maintained as bulk populations without clonal selection. DFT1 cells acquired resistance at 10 μ M sapitinib, while DFT2 cells reached resistance at 1 μ M imatinib.

To address the potential biological basis for these differences, we have expanded the discussion in the revised manuscript together with our discussions around the WGS data.

Minor comments on Data display and statistical analyses (for the main figures only):

- 1.5. Fig 1: can the authors please comment on statistical power and add such considerations to the discussion. A) E.g. are three replicates of DFT1, DFT2 and fibroblast cells sufficient to call significant phospho-peptide enrichment? B) Could there be technical reasons for why there are more significant DFT1 over DFT2 p-sites? C) Across analyses: why do you believe that relatively few p-sites stand out strongly?

We thank the reviewer for spotting this and for raising this important point regarding statistical power and interpretation of our phosphoproteomics data. We address the reviewer questions below and have included a dedicated paragraph in the new 'Limitations' section (**lines 461-471**) and expanded relevant details in the Materials and Methods and the accordingly newly edited Figure legends.

A) Replication and statistical power:

We performed phosphoproteomics on three independent replicates per group (DFT1, DFT2, and fibroblasts), with each replicate derived from cells harvested on different days in culture. While we acknowledge that more replicates would improve statistical power, this replication level is consistent with established standards in the field, particularly given the high cost and complexity of phosphoproteomic workflows. Statistical analyses were performed using established pipelines that incorporate replicate variability and control for false discovery rate, allowing us to identify confidently enriched phospho-peptides.

B) Differences in significant phospho-sites between DFT1 and DFT2:

We agree that some technical variability is possible but we believe the observed differences between DFT1 and DFT2 likely reflect underlying biology rather than technical bias. Sample preparation and LC-MS/MS data acquisition were performed in parallel under rigorously standardized conditions. Biologically, this result aligns with our broader findings: DFT1 appears to engage a wider spectrum of RTK signalling pathways, which could result in broader downstream phosphorylation patterns, whereas DFT2 signalling is more narrowly centered on PDGFRA activity. That said, we are cautious in our interpretation, as the phosphoproteomics was conducted on one representative line per tumour type. However, selected phosphorylation events were supported by our independent Western blot validation as well, suggesting validity of proposed finding.

C) On the relatively limited number of statistically significant p-sites:

Phosphorylation is a highly dynamic and transient process, especially in asynchronous cell populations. The relatively small number of statistically significant p-sites likely reflects biological heterogeneity between replicates rather than uniformly subtle changes. Indeed, several phospho-sites show fold changes >2 but remain statistically non-significant due to inter-replicate variability. In addition, since DFT1 and DFT2 are both derived from Schwann cells, they may share substantial overlap in baseline signalling, which could reduce the number of divergent phosphorylation events. Lastly, we applied conservative statistical thresholds to minimize false positives, which further reduces the number of reported hits but enhances confidence in those that are called by analytic measurement and data evaluation.

- 1.6. Fig 1C: please highlight key kinase peptides in the volcano plot, esp. JAK/STAT and PDGFR ones shown in panels E and F

We thank the reviewer for this helpful suggestion. In the revised **Figure 1C**, we have now highlighted the significantly enriched phosphopeptides corresponding to key downstream targets of PDGFRA and ERBB signalling, including components of the JAK/STAT, MAPK, and PI3K pathways (featured in **Figure 1E and F**). These additions will improve the interpretability of the volcano plot to strengthen its integration into the broader signalling narrative as presented in the revised manuscript.

- 1.7. Fig 2AB: A) please instead show a full heatmap with IC₅₀ (log) scores with all vs. all cell lines tested; B) the binary categorisation of "hit" vs "no hit" in the Venn diagram does not reflect the expected differences in IC₅₀; moreover, the colouring by ERBB vs. PDGFR inhibition is questionable with respect to inhibitor specificity (see point 1)

A) We now include a full heatmap displaying all IC₅₀ values (in linear scale/ μ M) across all tested cell lines in the main Figure body (**Figure 2B**). While log-scale is commonly used, we opted for a linear scale here to better reflect experimental relevance in a pharmacological context (treatment concentrations used *in vitro*).

B) While we retained the binary categorization of "hit" vs. "no hit" in the Venn diagram for clarity, we fully acknowledge its limitations, particularly in light of kinase inhibitor promiscuity and variable IC₅₀ values. To address this:

- We have now uploaded the complete IC₅₀ and AUC dataset, along with our hit selection criteria, as an Appendix Table S5 for full transparency.
- The rationale and methodology behind the screening and hit calls are now described in more detail in the Materials and Methods section.
- In addition, we revised the color coding of inhibitors in the Venn diagram to better reflect their multi-target profiles and to align with our updated kinase-target network shown in Appendix Figure S2.

We appreciate the helpful comment and advice, which led to improved transparency and clarity in this key dataset.

- 1.8. (current) Fig 2B: it would be helpful to clarify if this is displayed on the log scale and which IC₅₀ unit is meant.

Thank you for this helpful suggestion. We now explicitly indicate that the IC₅₀ values are displayed on a linear scale, and we have added the unit (μ M) in the axis label for clarity.

- 1.9. Fig 2C-G: it is a bit difficult to understand why the DFT2 PDX model is merged with the DFT1 and DFT2 cell line drug screen into a single figure. While it is certainly a

great new experimental system, these xenograft data don't fit much with the overall story on DFT resistance.

We agree that the xenograft (PDX) model represents a distinct dataset and that its inclusion alongside the *in vitro* drug screen may have diluted its interpretability. In response, we have split the original Figure into two. The *in vivo* PDX data now appears as a separate main Figure (**Figure 3**), highlighting its role as a valuable experimental resource. We believe this new model is a key contribution to the field and merits prominent placement in the manuscript as it opens avenues for translational studies and therapeutic testing in a relevant system.

1.10. Fig 3B: it is probably cleaner to flip the X and Y axes to stay consistent with the standard scheme of displaying PC1 on the X and PC2 on Y.

Thank you for catching this. We have flipped the X and Y axes in the PCA plot to display PC1 on the X-axis and PC2 on the Y-axis, following standard convention. This Figure now appears as **Figure 5B**.

1.11. Fig 3C and 3D: compared to Fig 1C, it looks like the cut-off for significant "hits" is differently calculated between the different sets of volcano plots? For consistency with the suggested Fig. 1 edits, could the authors also highlight key peptides in these volcano plots which are of relevance to panels E-G of the same figure (e.g. JAK/STAT hits)?

In the originally submitted version of the manuscript, we applied a less stringent statistical threshold to detect more subtle differences between parental and resistant cell lines. However, based on insights gained from our whole genome sequencing (WGS) data and your valuable comment, we reanalysed the phosphoproteomic data using a more stringent and consistent threshold ($FDR < 0.05$, $s_0 = 0.1$), in line with the criteria used in **Figure 1**.

This reanalysis has strengthened the consistency of our findings. Our WGS data show that DFT2, which accumulates more genomic alterations (SNVs and CNVs) also changes its phosphoproteome significantly. DFT1 displays fewer changes, particularly at the copy number level and shows no significantly altered phosphopeptides under the revised thresholds.

See a revised results section in lines **298-315**.

To improve clarity, we have now highlighted all JAK/STAT-related phosphopeptides of interest in the volcano plots, as well as all peptides included in the corresponding heatmaps, to make it easier for readers to track which features were selected for downstream interpretation.

If desired, we would be happy to include the results from the original, less stringent analysis in the appendix. However, the overall conclusions of our study remain unchanged and, in fact, the reanalysis better aligns with our complimenting data.

1.12. Fig 3F: what is the meaning of the different categories of significance (*, **, ***, ****)?

We thank the reviewer for pointing this out.

These qPCR plots appear now as **Fig. 5A** in the updated manuscript.

We have now clarified in the Figure legend that statistical significance was defined as follows:

$p < 0.0332$ (*),
 $p < 0.0021$ (**),
 $p < 0.0002$ (***),
 $p < 0.0001$ (****).

The Figure legend has been updated accordingly.

1.13. Fig 4B: fitting a line through these three data points and the associated R² value do not provide additional value, could you please remove these?

This Figure appears as Fig. 6B in the updated manuscript.

We agree that the fitted line and the associated R² value did not provide meaningful information given the limited number of data points. We have removed both from **Figure 6B** to simplify and clarify the visual presentation. We thank the reviewer for this thoughtful suggestion.

1.14. Fig 4D (like 3F): what is the meaning of the different categories of significance?

These qPCR plots appear now as **Fig. 6E-F** in the updated manuscript.

As in **Figure 5**, we have now included a clear explanation of the significance levels (*, **, ***, ****) directly in the **Figure 6E-F** legend to ensure clarity and consistency across the manuscript.

1.15. Use of colours: this is of course a bit subjective and more of an editorial comment, but I believe that the colour scheme may be substantially improved. Currently, readers need to distinguish between: DFT1 (red), DFT2 (blue), enrichment scores, Z-scores, IC50 values (red/red-black/pink), Sapitinib (red), Imatinib (blue), and ZIP synergy scores (red-green). Some of the former papers on DFT have used the opposite colour scheme of DFT1 (blue) vs. DFT2 (red) which adds a bit to the confusion, too.

We appreciate this thoughtful and constructive comment. We have carefully revised our colour scheme to improve clarity while retaining consistency throughout the manuscript. Specifically:

- We maintained DFT1 = pink and DFT2 = blue, which we used consistently across Figures.
- To avoid confusion with inhibitor colour coding, we now distinguish sapitinib and imatinib by symbols and line styles instead of colour.
- Where possible, we converted heatmaps (e.g. IC₅₀ values) to greyscale to minimize colour overload (e.g., **Fig. 2**).
- For enrichment heatmaps, we retained the red–grey scale to clearly show up- vs. downregulation.

We are of course open to further refinements and happy to work with the publisher on optimizations during production shall that be required. In summary, we hope that we have substantially updated and upgraded the data compilation and sharpened conclusions and result presentation.

2 EMBO Journal-Referee #2:

The Tasmanian devil's facial tumour (DFT) is a clonal cancer that transmits naturally as an allograft. Two DFTs are known, DFT1, and the more recently discovered DFT2. The authors previously reported that DFT1 is mainly driven by dysregulated ERBB3/STAT3 signalling whereas DFT2 is mainly driven by dysregulated PDGFRA signalling. Here, the authors performed global phosphoproteomics analyses on DFT1 and DFT2, which support and expand their previous conclusions on the oncogenic drivers. The authors then conducted a screening on DFT1 and DFT2 cell lines to assess their susceptibility to a panel of tyrosine kinase inhibitors (TKI). They found that DFT1 cells are highly susceptible to Sunitinib (EGFR inhibitor) whereas DFT2 cells are highly susceptible to Imatinib (PDGFRs inhibitor), in agreement with the predicted outcomes based on the phosphoproteomics results. Drug-resistant DFT1 and DFT2 cells were selected and mechanisms of resistance were analysed, showing that DFT1 broadly maintains the key TKI signalling pathways intact in contrast to DFT2, which drifts from PDGF to ERRB signalling.

The paper is interesting and well written, and most of the data presented are of high quality. Understanding mechanisms of DFT resistance is important for possible treatment of captive devils and the basic mechanisms of clonal cancer evolution. However, the paper also has weaknesses that should be addressed.

Dear Referee #2,

We would like to thank you for a positive view on our manuscript, as well as for your constructive feedback and thoughtful suggestions, which will help to enhance the quality and value of our work.

- 2.1. While the shift in the TKI signalling pathways is convincing for DFT2 resistant cells (DFT2-R), it is less convincing for DFT1-R cells. Suppl. Figure 5D shows variable and modest changes in DFT1-R1 and R2 relative to parental (P) and the Western blot in panel F also shows minimal differences. These results suggest that the resistance may be due to mutations in the drug target, and it is surprising that the authors have not tested this common mechanism of resistance.

We appreciate your insightful comment regarding the mechanisms underlying resistance in our DFT cell lines. Indeed, our phosphoproteomic and Western blot analyses show more modest and variable changes in signalling pathways for DFT1-R lines compared to the pronounced shifts observed in DFT2-R cells. This suggests that resistance mechanisms in DFT1 may be distinct and potentially less reliant on broad pathway rewiring.

Regarding the possibility of mutations in the drug target as a mechanism of resistance in DFT1, we agree that target mutations are a well-documented and common resistance mechanism in kinase inhibitor-treated cancers. To address this, we performed whole-genome sequencing (WGS) on our resistant cell lines, including DFT1-R1 and R2, as detailed in the updated **Figure 4** and associated data (See **Table** on page 2 of this rebuttal document for overview).

To ensure robust and interpretable results, we performed additional sequencing runs, which improved sequencing depth and strengthened our dataset quality. While time constraints precluded certain analyses such as fusion gene and structural variant discovery, our study provides valuable insights into the genetic changes in DFT1 and DFT2 tumor model systems under drug-induced selective pressure.

Changes in copy numbers affecting *EGFR*, *ERBB3* and *ERBB4* in the resistant DFT2 cells indicate that the ERBB pathway gains importance in the resistant cell lines. In DFT1 cells, however, no CNV were affecting these genes. *EGFR* and *ERBB4* show some SNVs (see Appendix Table S6) in the resistant DFT1 cells, however, with unknown consequences.

- 2.2. Phosphoproteomics did not detect p-ERBB3, however the Western blot for ERBB3 in Figure 2C shows bands above ERBB3 that look like phosphorylated ERBB3 forms, although it could also be ubiquitination. In any case, it should be fairly straightforward to test this by Western blotting using phospho-specific ERBB3 antibodies.

Thank you for this important observation. The band appearing above the main ERBB3 signal in **Figure EV2C** corresponds to phosphorylated ERBB3, as confirmed by phospho-specific ERBB3 Western blots performed during our revision. We have updated the Figure legend accordingly and briefly discuss this in the manuscript (**lines 151-153**).

Interestingly, despite detectable ERBB3 phosphorylation in DFT2 cells, our functional data indicate that ERBB inhibition is not a major vulnerability in these cells. This aligns with our phosphoproteomic and transcriptomic analyses showing relatively higher activation and expression of ERBB2 compared to ERBB3 in DFT2. Given that ERBB3 lacks intrinsic kinase activity, it may function primarily through non-catalytic or scaffold roles in devil cells, potentially differing from canonical signalling roles described in other species. However, we want to be careful with such assumptions.

Additionally, in resistant DFT2 cells, we observe upregulation of EGFR rather than ERBB3, suggesting a shift away from ERBB3-centered signalling during resistance. This is consistent with previous reports highlighting an ERBB3–STAT3 axis in DFT1, especially since DFT1 harbours an ERBB3 gene duplication as described by Stammnitz and colleagues in pioneering work.

Together, these data highlight tumour lineage–specific RTK signalling wiring in DFT1 versus DFT2 and underscore the importance of further investigations to clarify the precise role of ERBB3 in the Tasmanian devil cancer system.

Fig. EV3C: Western blot analysis of ERBB3, pERBB3 (Y1289), PDGFRA, PDGFRB

- 2.3. It is at times difficult to match the text with the Figures because one needs constantly to jump from main to Supplemental. Reorganising the Figures to show the key results for DFT1 and DFT2 side by side, or at least in consecutive Figures would help the reader.

We appreciate this valuable feedback and agree that improved Figure organization enhances readability. In response and to follow your advice, we have reorganized both main and Supplemental Figures to present DFT1 and DFT2 results more closely together, either side-by-side or in consecutive panels. This restructuring extends to Figure layouts and legend phrasing, particularly within the resistance sections, to better highlight the comparative aspects and distinct resistance mechanisms of the two tumour types. We believe these changes significantly improve the flow and accessibility of the data for readers.

- 2.4. Line 209 should refer to Figure 3C.

Thank you for pointing out this mistake, we double checked all Figure references now.

- 2.5. Line 379, it should read "prove".

Thank you for pointing out this typo, we have rewritten this part.

Dear Dr Moriggl,

Thank you for submitting your revised manuscript (EMBOJ-2025-120337R) to The EMBO Journal, as well for your patience with our feedback. Your amended study was sent back to the referees for their scientific reassessment, and we have received re-reports from both, which I enclose below. As you will see, the reviewers state that the work has been substantially enhanced by the revisions and they are now broadly in favour of publication, pending minor amendments.

Thus, we are pleased to inform you that your manuscript has been accepted in principle for publication in The EMBO Journal.

Please carefully consider the remaining minor points raised by the referees by adjusting the data presentation and discussion of the findings where appropriate.

Also, we now need you to take care of a number of issues related to formatting and data presentation as detailed below, which should be addressed at re-submission.

Please contact me at any time if you have additional questions related to below points.

Thank you for giving us the chance to consider your manuscript for The EMBO Journal. I look forward to your final revision.

Again, please contact me at any time if you need any help or have further questions.

Best regards,

Daniel Klimmeck

>> Author Contributions: Remove the author contributions information from the manuscript text. Note that CRediT has replaced the traditional author contributions section as of now because it offers a systematic machine-readable author contributions format that allows for more effective research assessment. and use the free text boxes beneath each contributing author's name to add specific details on the author's contribution.

More information is available in our guide to authors.
<https://www.embopress.org/page/journal/14602075/authorguide>

>> Correct the order of the manuscript sections as follows: Abstract / Keywords / Introduction / Results / Discussion / Methods / Data Availability / Acknowledgements / Disclosure and Competing Interests Statement / References / Main Figure Legends / Tables / Expanded View Figure Legends. The Data Availability section should be after the Methods.

>> Remove the bullet points from the manuscript and upload as synopsis, together with a summary of the findings.

>> Figure callouts: Please ensure that the figures and tables are called out in sequential order. Currently, Appendix Tables S6-9 are called out before Appendix Table S1 - 5.

>> Figures in separate files: all figures should be uploaded as individual, high-resolution figure files; main figures first, followed by EV figures.

>> Funding: please enter the funding information in the list of funders into our online system.

>>Appendix file with ToC: Please add a table of contents, including page numbers on the first page of the appendix. please add the appendix tables to the PDF file.

>> Add a separate 'Statistical Analysis' section to the Methods part, detailing the algorithms and statistical tests applied.

>> References: adjust the reference format to EMBO Journal format, 10 authors et al, and place References after the Disclosure

and competing interests statement, before figure legends. Remove dois.

>> Reagents and Tools table: Please upload as a separate file using the existing template in the Guide For Authors, listing key reagents, experimental models, software and relevant equipment.

>> Data availability section: please remove the referee token for the PRIDE dataset and make sure that the data are made publicly accessible. Add hyperlinks to the datasets.

>>Discussion: remove the explicit 'Limitation of the Study' part and integrate as running discussion.

>> Consider additional changes and comments from our production team as indicated below:

- DAS:

1. Please note that the specific URLs for PRJEB93974, PXD058352 datasets are not provided in the data availability statement.
2. Please note that reviewer access code for dataset PXD058352 is provided in the manuscript.
>>> now accessible AD 5.8.25
3. Please note that reviewer access code for PRJEB93974 dataset is not provided in the data availability statement.
4. Please define the annotated p values ****/****/**/* as well as provide the exact p-values for the same in the legend of figure EV2 A, B; 3C, E; 6E, F as appropriate.
5. Please note that the exact p values are not provided in the legends of figures 5A, G; EV5 A
6. Please indicate the statistical test used for data analysis in the legends of figures 1C, D; EV2 A, B; 5A, C, D
7. Please note that information related to n is missing in the legends of figures 1C, 5C, D
8. Please note that n=2 in figure 5G
9. Please note that the error bars are not defined in the legends of figures 3E, 6A

Please use the link below to submit your revision:

Referee #1:

Schönbichler et al (2025) have substantially revised their manuscript by addition of:

- key data, most notably whole genome sequencing (WGS) of the parental and resistant DFT1 and DFT2 cell lines
- key analyses, including structural modelling of human vs. Tasmanian devil tyrosine kinase target pockets as well as chromosomal inspections of copy number and single-nucleotide variation by use of their WGS
- key clarifications with respect to the difference in resistance evolution between DFT1 and DFT2, the differences in specificity between TK inhibitors, statistical analyses, figure colouring, etc.

I agree with the authors that the WGS data here generated represent a valuable resource. If analysed in more detail, e.g. by looking at structural variants as suggested by the authors, these will likely hold further insights. Future studies on DFT evolution and resistance will surely benefit from this benchmark.

Very minor points:

- the novel finding of convergent DFT2 (but not DFT1) tetraploidisation upon drug resistance is intriguing; I'd suggest to highlight this observation in the abstract
- could the authors please state the tumours' median whole genome coverage from their NovaSeq data?
- human proteins are determined with an "h", and devil proteins with an "s"; upon first use, please clarify that "s" stems from *Sarcophilus* (if that's correct)
- for improved reproducibility of in silico and statistical analyses, could the authors please summarise all their code in a dedicated open repository like Github?

Altogether, this represents a major improvement on the original study, and I now do support it for publication. I wish to thank and congratulate all the authors to their impressive progress, indeed within a very short time frame!

Referee #2:

The manuscript is significantly better and clearer. The genetic and transcriptional adaptations of DFT2-Resistant cells are well

documented, convincing and interesting. The mechanisms leading to DFT-1 resistance remain rather unclear and the authors did not seem to have directly addressed the question if mutations in the target protein may explain the phenotype. The rapid emergence of resistance and greater overall SNVs found in DFT1 compared to DFT2 suggest that target mutation is indeed a possibility. However, I also recognize that this might be a difficult question to address experimentally in the Tasmanian devil's cells, and the authors provide the WGS data and so this will probably be fully addressed in the future. Nonetheless, this is a limitation of the study, which should be discussed. That said, the study is important and provides valuable information not only to the transmissible cancer community, but also to the larger oncology field, by documenting and analysing complex pathways of resistance to TKIs.

EMBOJ-2025-120337 Final Submission Comments

Referee #1:

Schönbichler et al (2025) have substantially revised their manuscript by addition of:

- key data, most notably whole genome sequencing (WGS) of the parental and resistant DFT1 and DFT2 cell lines
- key analyses, including structural modelling of human vs. Tasmanian devil tyrosine kinase target pockets as well as chromosomal inspections of copy number and single-nucleotide variation by use of their WGS
- key clarifications with respect to the difference in resistance evolution between DFT1 and DFT2, the differences in specificity between TK inhibitors, statistical analyses, figure colouring, etc.

I agree with the authors that the WGS data here generated represent a valuable resource. If analysed in more detail, e.g. by looking at structural variants as suggested by the authors, these will likely hold further insights. Future studies on DFT evolution and resistance will surely benefit from this benchmark.

We sincerely thank the reviewer for positive feedback as well as for recognizing the value of the additional data and analyses included. We are pleased that the reviewer considers our WGS dataset a valuable resource to support future studies on DFT evolution and resistance.

Very minor points:

- the novel finding of convergent DFT2 (but not DFT1) tetraploidisation upon drug resistance is intriguing; I'd suggest to highlight this observation in the abstract

We thank the reviewer for carefully examining our WGS data and for highlighting this interesting observation. We also find the convergent tetraploidisation in DFT2-resistant cells intriguing. However, as this finding is currently based solely on our WGS analysis and bioinformatic analysis, and we have not yet confirmed ploidy changes using experimental approaches such as FACS or in situ hybridization, we prefer to be cautious and not emphasize this result in the abstract. We agree that this observation warrants further investigation, and we hope it could be a new focus of future work.

- could the authors please state the tumours' median whole genome coverage from their NovaSeq data?

The reviewer has noted that we should disclose the sequencing statistics, which we now include as the median whole-genome coverage information for all tumours, as now reported in Appendix Table S10.

- human proteins are determined with an "h", and devil proteins with an "s"; upon first use, please clarify that "s" stems from *Sarcophilus* (if that's correct)

We thank the reviewer to enhance clarity of sequence comparison description by species linkage with abbreviated wording more explicitly. Thus, we have now clarified in the Results section (lines 179-181) Materials and Methods section (line 724-725) that proteins annotated with "h" refer to human proteins, whereas those annotated with "s" refer to devil proteins, with "s" denoting *Sarcophilus*.

- for improved reproducibility of in silico and statistical analyses, could the authors please summarise all their code in a dedicated open repository like Github?

We agree with the reviewer request and we have now uploaded all code used for *in silico* and statistical analyses to a dedicated GitHub repository, which is accessible at: https://github.com/luenling/Schoenbichler_2025

Altogether, this represents a major improvement on the original study, and I now do support it for publication. I wish to thank and congratulate all the authors to their impressive progress, indeed within a very short time frame!

Thank you for all the advice and hard thinking! 😊

Referee #2:

The manuscript is significantly better and clearer. The genetic and transcriptional adaptations of DFT2-Resistant cells are well documented, convincing and interesting. The mechanisms leading to DFT-1 resistance remain rather unclear and the authors did not seem to have directly addressed the question if mutations in the target protein may explain the phenotype. The rapid emergence of resistance and greater overall SNVs found in DFT1 compared to DFT2 suggest that target mutation is indeed a possibility. However, I also recognize that this might be a difficult question to address experimentally in the Tasmanian devil's cells, and the authors provide the WGS data and so this will probably be fully addressed in the future. Nonetheless, this is a limitation of the study, which should be discussed. That said, the study is important and provides valuable information not only to the transmissible cancer community, but also to the larger oncology field, by documenting and analysing complex pathways of resistance to TKIs.

We thank the reviewer to review again our manuscript and to find merit in the overall conclusion and findings of the study for the oncology field. We agree that mutations in the target kinases may contribute to resistance, particularly given the rapid emergence of resistance. While a comprehensive analysis of target mutations was beyond the scope of the current study, we have now added a more detailed description to the findings to the discussion (Line 401-404) following the reviewer suggestion to better highlight this more critical as a potential limitation. Nevertheless, the reviewer was positive that our WGS data provide a valuable resource for future studies and we are happy to share the reagents, cell lines generated during drug resistance or any data originating from it with the oncology research community for further investigations.

Dear Dr Moriggl, dear Dr Bergthaler,

Thank you for submitting the revised version of your manuscript. I have now evaluated your amended manuscript and concluded that the remaining minor concerns have been sufficiently addressed.

I am thus pleased to inform you that your manuscript has been accepted for publication in the EMBO Journal.

On a different note, I would like to alert you that EMBO Press offers a format for a video-synopsis of work published with us, which essentially is a short, author-generated film explaining the core findings in hand drawings, and, as we believe, can be very useful to increase visibility of the work. Please see the following link for representative examples and their integration into the article web page:

<https://www.embopress.org/doi/full/10.1038/s44318-025-00417-0>

Finally, we have noted that the submitted version of your article is also posted on the preprint platform bioRxiv. We would appreciate if you could alert bioRxiv on the acceptance of this manuscript at The EMBO Journal in order to allow for an update of the entry status. Thank you in advance!

Best regards,

Daniel Klimmeck

Daniel Klimmeck, PhD
Senior Editor
The EMBO Journal
EMBO
Postfach 1022-40
Meyerohofstrasse 1
D-69117 Heidelberg

contact@embojournal.org
